# Ancestral allele of DNA polymerase gamma modifies antiviral tolerance

Yilin Kang[1], Jussi Hepojoki[2,3], Rocio Sartori Maldonado[1], Takayuki Mito[1], Mügen Terzioglu[1], Tuula Manninen[1], Ravi Kant[2,4,5], Sachin Singh[6], Alaa Othman[7], Rohit Verma[1], Johanna Uusimaa[8,9], Kirmo Wartiovaara[1,10], Lauri Kareinen[2,4,11], Nicola Zamboni[7], Tuula Anneli Nyman[6], Anders Paetau[1,10,12], Anja Kipar[3,4], Olli Vapalahti[2,4,10] & Anu Suomalainen[1,10,13 ✉]

Mitochondria are critical modulators of antiviral tolerance through the release of mitochondrial RNA and DNA (mtDNA and mtRNA) fragments into the cytoplasm after infection, activating virus sensors and type-I interferon (IFN-I) response[1-4]. The relevance of these mechanisms for mitochondrial diseases remains understudied. Here we investigated mitochondrial recessive ataxia syndrome (MIRAS), which is caused by a common European founder mutation in DNA polymerase gamma (POLG1)[5]. Patients homozygous for the MIRAS variant p.W748S show exceptionally variable ages of onset and symptoms[5], indicating that unknown modifying factors contribute to disease manifestation. We report that the mtDNA replicase POLG1 has a role in antiviral defence mechanisms to double-stranded DNA and positive-strand RNA virus infections (HSV-1, TBEV and SARS-CoV-2), and its p.W748S variant dampens innate immune responses. Our patient and knock-in mouse data show that p.W748S compromises mtDNA replisome stability, causing mtDNA depletion, aggravated by virus infection. Low mtDNA and mtRNA release into the cytoplasm and a slow IFN response in MIRAS offer viruses an early replicative advantage, leading to an augmented pro-inflammatory response, a subacute loss of GABAergic neurons and liver inflammation and necrosis. A population databank of around 300,000 Finnish individuals[6] demonstrates enrichment of immunodeficient traits in carriers of the POLG1 p.W748S mutation. Our evidence suggests that POLG1 defects compromise antiviral tolerance, triggering epilepsy and liver disease. The finding has important implications for the mitochondrial disease spectrum, including epilepsy, ataxia and parkinsonism.

Mitochondrial dysfunction is an important contributor to pathogenesis of neurodegenerative diseases, with a considerable range of manifestations from severe epilepsy to various forms of peripheral or central nervous system degeneration[7]. MIRAS, which is caused by genetic mutation(s) in the nuclear-encoded catalytic α-subunit of POLG1, is unusually variable in age of onset and clinical manifestations[8]. Disease symptoms in patients with MIRAS carrying identical homozygous founder mutations may even manifest differently—in early adolescence, early adulthood or middle age. The clinical spectrum varies from treatment-resistant epilepsy and valproate hepatotoxicity to ataxia–polyneuropathy with or without epilepsy, or polyneuropathy–parkinsonism without epilepsy[5,8–12]. The underlying POLG1 variant (c.2243G>C, p.W748S; coinciding with the neutral p.E1143G cis-variant; hereafter the MIRAS allele) is common in populations of European descent with a carrier frequency of 1:84 and 1:100 in Finnish and Norwegian populations, respectively[5,13]. The allele originates from a single ancestral founder individual, dated back to Viking times[5,13]. The p.W748S change affects the intrinsic processivity region of POLG1 that is involved in replisome contacts and mtDNA processivity, without altering the polymerase catalytic functions[14]. Out of the variable MIRAS phenotypes, the most severe is the acute-status epilepticus in a previously healthy teenager, manifesting a few weeks after a minor viral infection[5,15] and closely mimicking viral encephalitis[16,17]. These observations suggest that a viral infection could trigger the symptomatic MIRAS disease.

Abundant lines of research implicate mitochondria as key immune modulators in mouse models and human materials. Stress-induced mtDNA or mtRNA release to cytoplasm triggers a IFN-I response that

[1]Stem Cell and Metabolism Research Program Unit, Faculty of Medicine, University of Helsinki, Helsinki, Finland. [2]Department of Virology, Faculty of Medicine, University of Helsinki, Helsinki, Finland. [3]Laboratory for Animal Model Pathology, Institute of Veterinary Pathology, Vetsuisse Faculty, University of Zürich, Zürich, Switzerland. [4]Department of Veterinary Biosciences, Faculty of Veterinary Medicine, University of Helsinki, Helsinki, Finland. [5]Department of Tropical Parasitology, Institute of Maritime and Tropical Medicine, Medical University of Gdansk, Gdansk, Poland. [6]Department of Immunology, Institute of Clinical Medicine, University of Oslo and Rikshospitalet Oslo, Oslo, Norway. [7]Swiss Multi-Omics Center, ETH Zürich, Zürich, Switzerland. [8]Research Unit of Clinical Medicine and Medical Research Center, University of Oulu, Oulu, Finland. [9]Department of Pediatrics and Adolescent Medicine, Unit of Child Neurology, Oulu University Hospital, Oulu, Finland. [10]Helsinki University Hospital, HUS Diagnostics, Helsinki, Finland. [11]Finnish Food Safety Authority, Helsinki, Finland. [12]Department of Pathology, Faculty of Medicine, University of Helsinki, Helsinki, Finland. [13]HiLife, University of Helsinki, Helsinki, Finland. ✉e-mail: anu.wartiovaara@helsinki.fi

confers resistance to viral infection[18–21]. However, these reports suggest that chronic activation of mitochondrial-induced immune responses could contribute to degenerative disease, including neurodegeneration. The variable manifestations of MIRAS and the POLG1 mutation affecting mtDNA replication make MIRAS an excellent candidate for a disease involving a viral trigger.

## Immunity defects in MIRAS carriers

We first queried FinnGen, a Finnish population genome database with links to medical history data[6], for diagnoses that are enriched in individuals carrying the MIRAS-associated *POLG1* variant (rs113994097). Immunodeficiencies stood out as the most significant diagnosis (a sample of 309,154 Finnish individuals, $P = 2.01 \times 10^{-7}$; Fig. 1a). No similar enrichment of immunodeficient traits existed in a set of other mitochondrial and related disease gene variants (Extended Data Fig. 1a). The finding prompted us to examine the role of POLG1 and, particularly, the MIRAS allele in innate immune signalling.

## Decreased IFN-I and mtDNA/mtRNA release

The primary fibroblasts from patients with MIRAS (characteristics are shown in Extended Data Fig. 1b,c and Supplementary Table 1) showed decreased stability of POLG1 protein, with around a 50% reduction in the protein amount compared with the matched controls (Fig. 1b). No discernible changes were found in *POLG1* transcripts, POLG2 (the accessory β-subunit of POLG replisome), mitochondrial transcription factor A (TFAM), respiratory chain enzyme protein or transcript levels, or in the mtRNA or mtDNA abundance at the baseline (Extended Data Fig. 1d,e).

To examine the immune responses of these patients' cells, we challenged the fibroblasts with synthetic double-stranded DNA (dsDNA) or dsRNA (polyinosinic:polycytidylic acid, poly(I:C)). They mimic the pathogen-associated molecular patterns (PAMPs) of viruses, which are either released into the cytosol during host cell entry or produced during viral replication. They activate host cytosolic pattern recognition receptors (PRRs), including RNA receptors such as retinoic-acid-inducible gene I (RIG-I) and melanoma-differentiation-associated protein 5 (MDA5) and DNA receptors such as cyclic GMP-AMP synthase (cGAS) and RNA polymerase III, which can convert DNA into RNA intermediates, activating RIG-I. The activation triggers an immune cascade and converges on the production of IFN-I and pro-inflammatory cytokines leading to downstream auto/paracrine antiviral defence[22–24] (a schematic of the response is shown in Fig. 1c). Although the basal immune and cytokine gene expression levels were comparable between control and MIRAS cells, the latter showed a delayed and dampened initial IFNβ response to dsRNA or dsDNA challenges compared with the controls (Fig. 1d): MIRAS cells induced around a twofold decrease in *IFNB1* (encoding IFNβ) expression after 7 h of dsRNA treatment, and at 7 and 24 h after dsDNA treatment. Under these conditions, MIRAS cells also expressed reduced levels of IFN-inducible *RIGI* and IFN-stimulated genes (ISGs), including *ISG15* and IFN-induced protein with tetratricopeptide repeats 3 (*IFIT3*), while inflammatory cytokine genes (tumour necrosis factor (*TNF*), interleukin-6 (*IL6*) and *IL1B*) displayed variable induction dynamics to the two viral PAMP mimetic treatments (Fig. 1d and Extended Data Fig. 2a). The amounts of PRRs (RIG-I and MDA5), IFN-induced proteins (IFIT3 and IFIT2) and signal transducer and activator of transcription 2 (STAT2) were low after 24 h of PAMP mimetic treatment in MIRAS cells (Fig. 1e and Extended Data Fig. 2b,c; no difference in STING protein). We next tested the ability of viral-PAMP-mimetic-treated MIRAS and control cells to induce paracrine immune activation in naive cells (Extended Data Fig. 2d). Medium transferred from MIRAS cells resulted in a lower activation of the IFNβ pathway, supporting attenuated IFN-I cytokine release and paracrine immune response in MIRAS cells (Fig. 1f and Extended

Data Fig. 2e). Co-expression of constitutively active RIG-I and mitochondrial antiviral-signalling protein (MAVS) proteins in MIRAS cells enhanced IFNβ pathway activation in response to dsRNA treatment (Extended Data Fig. 3a–d). These results demonstrate that cells of patients with MIRAS mount a compromised early IFN-I response to viral PAMP mimetics.

mtDNA and mtRNA release into the cytoplasm has been reported to activate cGAS[18], RIG-I[21,25,26] and MDA5[19], and the IFN pathway. We investigated the ability of MIRAS and control fibroblasts to present mtDNA and/or mtRNA in the cytosol after exposure to viral PAMP mimetic. Both mtDNA and mtRNA amounts were decreased in the MIRAS cytosol compared with the total mtDNA or mtRNA pools (Fig. 1g and Extended Data Fig. 3e,f). These data support the conclusion that dampened mtDNA/mtRNA release from mitochondria contributes to lowered innate immunity activation in MIRAS fibroblasts.

## Overactivated pro-inflammatory response

Delayed and/or dampened early IFN-I response during viral infection can elicit a secondary aberrant activation of pro-inflammatory responses, particularly NF-κB signalling[27]. We investigated whether a prolonged viral PAMP mimetic exposure would trigger such a pro-inflammatory response in MIRAS fibroblasts. We found an increased amount of NF-κB transcription factor component (p65) and its Ser536-phosphorylated form that activates NF-κB signalling during viral infection[28] in MIRAS cells after 32 h of viral PAMP mimetic exposure (Fig. 1h and Extended Data Fig. 3g). This was accompanied by an increase amount of TNF−a pro-inflammatory cytokine that is associated with NF-κB activation. Neither IRF-3 transcription factor (which upregulates IFN-I cytokine expression), nor its activating kinase TBK-1 were induced under this treatment condition. *IFNB1* expression was modestly decreased in MIRAS cells at this prolonged treatment duration, pointing to a time-dependent cellular activation of IFN-I and inflammatory responses (Extended Data Fig. 3h). TNF-mediated pro-inflammatory signalling can activate necroptotic cell death through MLKL phosphorylation[29,30]. The phosphorylated MLKL (p-MLKL) signal was increased in MIRAS cells compared with in controls after 32 h of dsRNA and after 72 h of dsDNA treatment, before any gross changes in cell morphology (Fig. 1i and Extended Data Fig. 3g,i). Overall, MIRAS cells show a slow activation of the early IFN-I response, followed by overactivated pro-inflammatory NF-κB signalling and increased necroptotic sensitivity when challenged by viral PAMP mimetics.

## Aberrant responses to neurotropic viruses

Next, we tested the responses of MIRAS cells to bona fide viral infections. As the teenage-onset MIRAS manifestation resembles viral encephalitis, we included two neurotropic viruses: HSV-1, a dsDNA virus, and tick-borne encephalitis virus (TBEV), a positive-strand RNA flavivirus. The neuroinvasive SARS-CoV-2 virus, a positive-strand RNA virus underlying the COVID-19 pandemic, was also studied. All of these viruses share the characteristic of causing mild infections to most individuals, but severe delayed complications to a minority. The encephalitis caused by neurotropic HSV-1[31] or TBEV[32] are proposed to be a consequence of an overactivated innate immune response and/ or a cytokine storm[33]. In HSV-1-infected MIRAS cells, the intermediate−early regulatory protein of HSV-1, ICP27, showed around 1.6-fold higher expression compared with that in the similarly infected control cells at 24 and 48 h after infection, indicating decreased cellular restriction of viral replication in MIRAS (Fig. 2a–d and Extended Data Fig. 4a–d). HSV-1 infection decreased the POLG1 protein and mtDNA levels, especially in MIRAS, the latter being 40% less than in controls at 48 h after infection (Fig. 2b,d,e and Extended Data Fig. 4e). HSV-1 has evolved extensive strategies to evade and/or downregulate the host innate immune response. These include inhibiting IFN-I signalling and

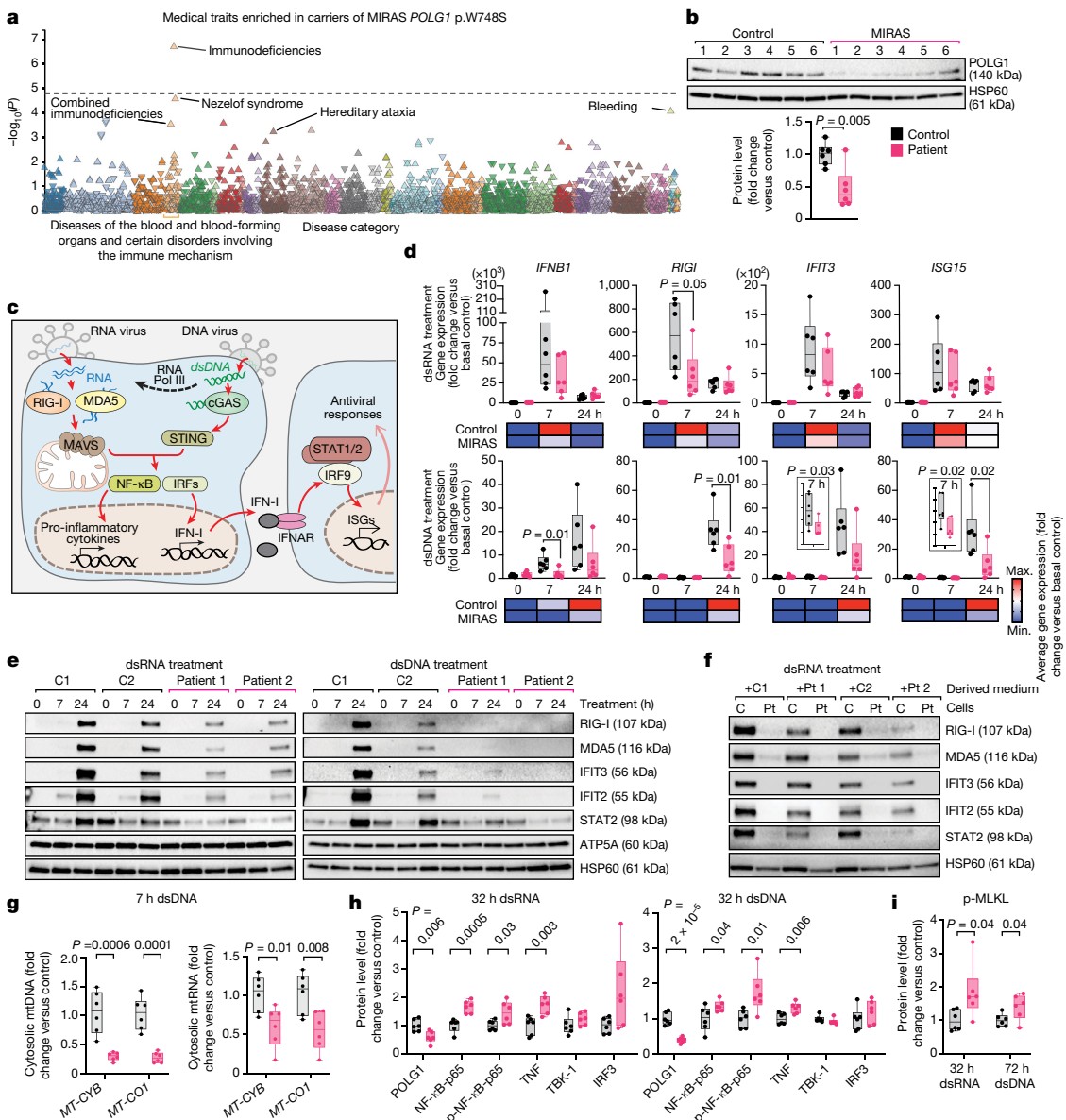

**Fig. 1 | Dysregulated immune signalling in fibroblasts of patients with MIRAS to viral PAMP mimetics. a**, The genotype–phenotype association of MIRAS *POLG1* variant (rs113994097). Significance (*P* values) and disease categories are shown. The triangles indicate diseases or traits: upward-pointing triangles show a positive association, and vice versa. The dotted line shows the cut-off for significance. Analysis was performed using SAIGE mixed model logistic regression. Data are from ref. 6. **b**, POLG1 protein levels in patients with MIRAS (patient) and control fibroblasts. Western blot and quantification. The loading control was HSP60. Fibroblasts are from six patients and six control individuals, all female. **c**, Schematic of antiviral innate immune signalling responses to viral PAMPs. **d**, IFN-I signalling pathway genes induced by viral PAMP mimetics (dsRNA/poly(I:C) or dsDNA) in patient and control fibroblasts (as in **b**). Quantitative PCR (qPCR) analysis of cDNA. The reference gene was *ACTB*. Top, box plot. Bottom, heat map showing the average gene expression per condition. **e**, IFN-I signalling pathway protein induction by viral PAMP mimetic (dsRNA (poly(I:C)) or dsDNA) in patient and control (C)

fibroblasts. Representative western blot analysis of four female control individuals and patients. The loading control was HSP60. Quantification is shown in Extended Data Fig. 2b. **f**, Paracrine immune signalling of fibroblasts in response to treatment with viral PAMP mimetic. Representative western blot of four female control individuals and patients (Pt). The loading control was HSP60. Quantification is shown in Extended Data Fig. 2e. **g**, mtDNA and mtRNA release into cytosolic extracts of fibroblasts (as in **b**; Extended Data Fig. 3e,f) after viral PAMP mimetic exposure for 7 h. Cytosolic versus whole-cell *MT-CYB* and *MT-CO1* DNA or cDNA was analysed using qPCR. **h,i**, Immune signalling (**h**) and necroptosis activation (**i**) in fibroblasts (as in **b**) after prolonged viral PAMP mimetic treatment. Quantification of the western blot is shown for the indicated treatment times (Extended Data Fig. 3g,i). The loading control was β-actin. For **b**, **d**, **g**, **h** and **i**, the box plots show minimum to maximum values (whiskers), 25th to 75th percentiles (box limits) and median (centre line). Statistical analysis was performed using two-tailed unpaired Student's *t*-tests. See also Extended Data Figs. 1–3 and Supplementary Table 1.

inducing host shut off (both are known functions of HSV-1 ICP-27) to facilitate viral gene expression and replication[34,35]. Accordingly, the host cell chaperone HSP60 showed progressive decline and IFN-I signalling protein levels in MIRAS and control cells changed after HSV-1 infection (Fig. 2b and Extended Data Fig. 4a–c). However, the infection activated pro-inflammatory NF-κB (Fig. 2b,c). At 24 h after HSV-1 infection, MIRAS

cells showed an increase in NF-κB-p65 and the Ser536 phosphorylated form (Fig. 2d and Extended Data Fig. 4b,d). This is in accordance with previous reports of HSV-1-induced persistent activation of NF-κB for efficient virus replication[36,37]. Consistent with the response induced by prolonged treatment with the PAMP mimetic, MIRAS cells also induced p-MLKL at 24 and 48 h of HSV-1 infection compared with the controls,

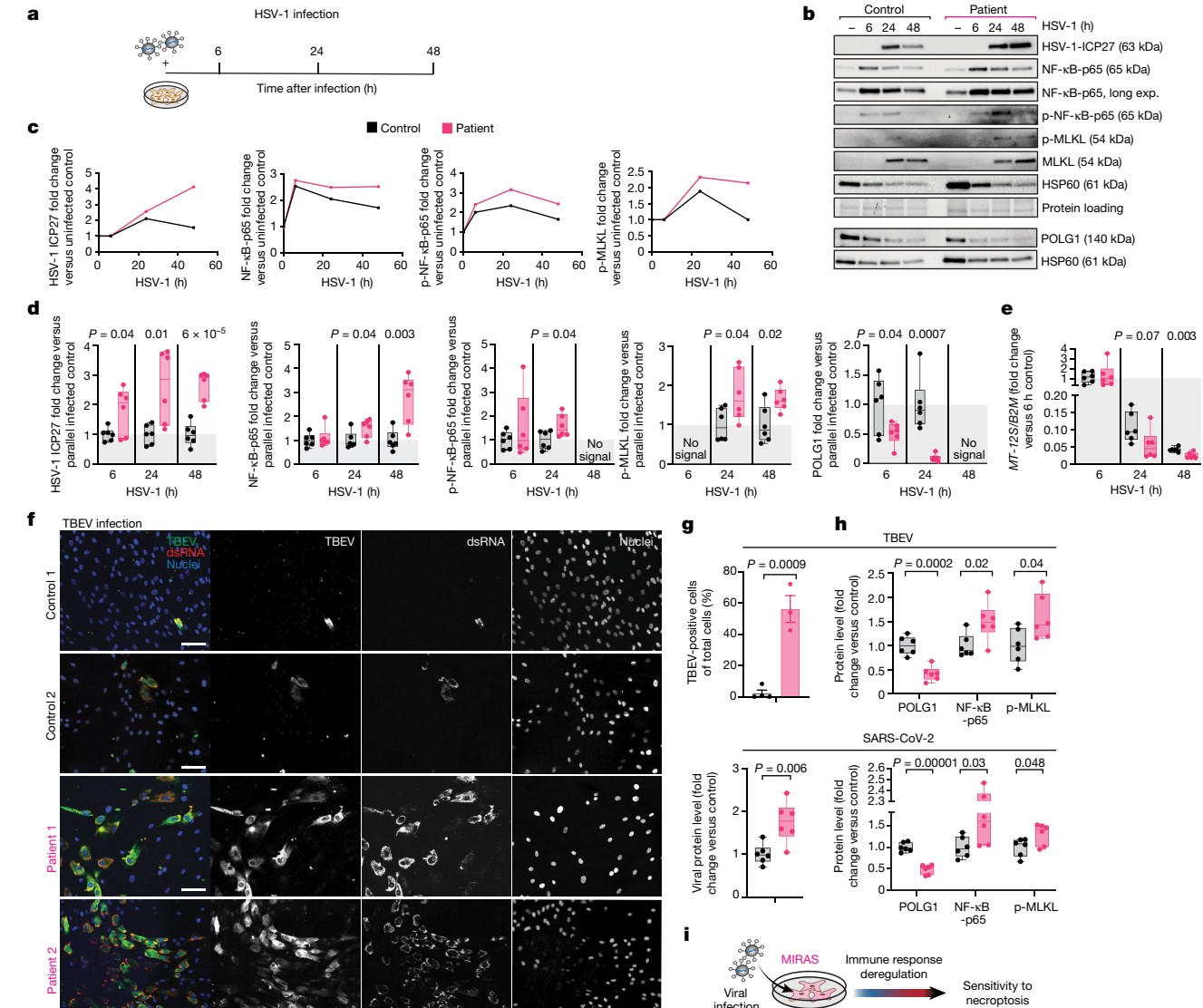

**Fig. 2 | Aberrant immune response of fibroblasts of patients with MIRAS to bona fide viral infection. a**, Schematic of HSV-1 infection of human primary fibroblasts. **b,c**, Viral load and host inflammatory protein level modulation in fibroblasts during HSV-1 infection. Western blot (**b**) and quantification (**c**) of protein amounts over time during infection in a control and patient fibroblast line. Protein loading stain was used as the loading control. Exp., exposure. **d**, Quantification of viral and host cellular protein amounts at the specific timepoint after HSV-1 infection. $n = 6$ female controls and patients. The western blot is shown in Extended Data Fig. 4a–c,e. The loading control was HSP60. **e**, mtDNA levels in fibroblasts (as in **d**) during HSV-1 infection. qPCR analysis of mtDNA (*MT-12S* (*MT-RNR1*)) relative to a nuclear gene (*B2M*). **f**, The viral load in fibroblasts after 48 h of TBEV infection. Immunofluorescence analysis of TBEV antigen (green) and dsRNA (red; detects viral RNA) with DAPI co-staining (blue) is shown. Each channel is shown in greyscale. Scale bars, 100 μm. $n = 2$ female control individuals and patients. **g**, Quantification of the viral load in

fibroblasts after 48 h of TBEV or SARS-CoV-2 infection. Top, the percentage of TBEV-positive cells. $n = 4$ control and 3 MIRAS images of 2 female control individuals and patients shown in **f**. Bottom, quantification of the western blot analysis of SARS-CoV-2 nucleocapsid protein (Extended Data Fig. 5h). The loading control was HSP60. $n = 6$ female control individuals and patients. **h**, POLG1 and inflammatory protein level in fibroblasts after 48 h of TBEV or SARS-CoV-2 infection. Quantification of the western blot is shown (Extended Data Fig. 5e–h). The loading control was HSP60. $n = 6$ female control individuals and patients. **i**, Schematic of the response of cells from patients with MIRAS to virus infection. For **d**, **e**, **g** (bottom) and **h**, the box plots show minimum to maximum values (whiskers), 25th to 75th percentiles (box limits) and median (centre line). For **g** (top), data are mean ± s.e.m. Statistical analysis was performed using two-tailed unpaired Student's *t*-tests. See also Extended Data Figs. 4 and 5.

but did not affect cellular viability at 48 h after infection (Fig. 2b–d and Extended Data Fig. 4e,f). These results corroborate the findings of viral PAMP exposure in MIRAS: a dampened early IFN response favours viral replication, resulting in overactivation of the pro-inflammatory response during prolonged infection and increased susceptibility to infection-induced necroptosis. CRISPR-correction of MIRAS *POLG1* p.W748S successfully restored POLG1 stability in induced patient fibroblasts (Extended Data Fig. 5a,b). After 48 h of HSV-1 infection, the corrected cells showed less NF-κB-p65, p-NF-κB-p65 and p-MLKL compared

with the patient cells (Extended Data Fig. 5c). Furthermore, mtDNA depletion induced by HSV-1 in the corrected MIRAS mutant lines was similar to the infected controls, indicating the causal role of *POLG1* p.W748S (Extended Data Fig. 5d).

Similar to HSV-1, TBEV and SARS-CoV-2 showed enhanced viral replication in MIRAS cells compared with in the infected controls. At 48 h of TBEV or SARS-CoV-2 infection, TBEV antigen (and dsRNA) or nucleocapsid protein (N) of SARS-CoV-2 were increased in MIRAS cells compared with in the controls (Fig. 2f,g and Extended Data Fig. 5h).

TBEV and SARS-CoV-2 infection showed severely decreased POLG1 protein in MIRAS cells. The NF-κB-p65 and necroptosis-activating p-MLKL were moderately increased (Fig. 2h and Extended Data Fig. 5e–h). At 48 h after infection, both TBEV and SARS-CoV-2 resulted in an elevated IFN response in MIRAS cells, including induced IFITs and STAT2, which were not similarly activated after HSV-1 infection, suggesting that some components of the immune overactivation in the context of MIRAS were virus specific (Extended Data Fig. 5e,f,i,j; negligible impact on fibroblast viability with the infection time frame analysed). These data collectively suggest that cells of patients with MIRAS mount an aberrant innate immune response to the three different viruses, HSV-1, TBEV and SARS-CoV-2, favouring cellular replication of viruses in the early infection phase, with a delayed, overactivated pro-inflammatory response and increased sensitivity to necroptosis (Fig. 2i).

### POLG1 and mtDNA depletion in MIRAS mice

To ascertain the physiological relevance of our findings in vivo, we generated a MIRAS mouse. These mice carry a homozygous knock-in MIRAS allele, homologous to the human MIRAS allele (p.W726S + E1121G in mice; p.W748S + E1143G in human *POLG1*) (Extended Data Fig. 6a–c). These mice are born in Mendelian proportions, and have a normal lifespan and body weight (Extended Data Fig. 6d). The mice show a 20% decrease in treadmill and 30% decrease in rotarod performance and a slightly abnormal gait compared with control mice (preliminary observation) at 12 months of age. Mitochondria isolated from the cerebral cortex, liver and spleen showed diminished amounts of POLG1, to 10–20% of the control mean. The accessory subunit POLG2 was modestly (brain) or not (liver, spleen) decreased and the amounts of TFAM and ATP5A subunit of the ATP synthase were unchanged (Fig. 3a). POLG activity was compromised—mtDNA replication activity was decreased in the brain and liver (in vivo BrdU incorporation analysis) (Fig. 3b). The mtDNA copy number was decreased by around 30% in the liver or largely unchanged in the brain compared with the controls, being surprisingly stable considering the POLG1 depletion and mtDNA replication decline (Fig. 3b). These results demonstrate the hallmarks of MIRAS disease in the MIRAS mice and validate it as a model for mtDNA maintenance disease.

### Compromised IFN-I signalling in MIRAS mice

We next examined the in vivo sensitivity of MIRAS and control mice to TBEV infection (Fig. 3c). We chose TBEV because it infects mice similar to humans, with neurotropism and nervous system manifestations[32]. At 4 days post-infection (d.p.i.), the circulatory IFNα and IFNβ levels were lower in MIRAS mice compared with in the controls (Fig. 3d). Moreover, the IFN-I pathway components in MIRAS mouse tissues reacted slowly to the infection (1 and 4 d.p.i.) (Fig. 3e and Extended Data Fig. 7a). At 4 d.p.i., the expression of IFN-I (*IFNA4* and *IFNB1*), IFN receptor (*IFNAR1*), the PRR *RIGI* and ISGs (*IFI44, IFI27, IFIT3, STAT1*) was decreased by around 30–50% in the MIRAS cerebral cortex and/or spleen (Fig. 3e), whereas pro-inflammatory NF-κB-p65 and TNF were moderately increased (Extended Data Fig. 7b). Transcriptomic analysis of the cerebral cortex at 4 d.p.i. demonstrated a weak induction of transcripts related to immune response and antiviral processes in MIRAS mice, while these were widely upregulated in controls (Fig. 3f and Supplementary Table 2). Expression of IFN regulatory factor 9 (*Irf9*), a key transcription factor of IFN-I response, was decreased by around 60% in the MIRAS cerebral cortex at 4 d.p.i. compared with the parallel-infected control mice. Functional enrichment analyses of the cerebral cortex transcriptome of TBEV-infected mice pointed to changes associated with neurodegeneration and seizure disorders in MIRAS mice (Fig. 3g). These data propose that MIRAS mouse brains are more sensitive than control mice to TBEV infection owing to a weak

ability to elicit early mechanisms for viral defence, while promoting inflammatory and neurodegenerative pathways.

### TBEV depletes nucleotide and mtDNA pools

Viruses actively reprogram host cell metabolism to capture biomolecules for their replication and for inhibiting host immune responses[38,39]. The metabolomic effects of TBEV infection in the cerebral cortex of mice at 4 d.p.i. showed a genotype-dependent metabolic fingerprint (Fig. 4a and Supplementary Table 3): (1) decreased nucleotide metabolism, especially the steady-state pools of pyrimidines (UMP, dUDP, thymine, thymidine, deoxycytidine, deoxyribose) required for cellular RNA and DNA synthesis; (2) altered methyl cycle and transsulfuration pathway driving cysteine, taurine and glutathione synthesis; and (3) amino acid metabolism (Fig. 4b–d and Extended Data Fig. 7c). Nucleotide metabolism was the most impacted process in the brain of MIRAS mice at 4 d.p.i. of TBEV, consistent with the viral-induced depletion of mtDNA in the tissue to almost 50% of controls (Fig. 4e and Extended Data Fig. 7d; no significant difference in the spleen or liver mtDNA amount). The mitochondrial pyrimidine nucleotide transporter (*Slc25a33*) was increased by approximately 40% in the brain of MIRAS mice (Fig. 4f) and glutathione metabolism, which is required for deoxynucleotide synthesis by ribonucleotide reductase and antioxidant defence, was remodelled (Fig. 4b–d and Extended Data Fig. 7c). These findings in the brains of MIRAS mice indicate severe rewiring of cellular nucleotide pools that are known to be required in viral replication[20,38,39].

### TBEV depletes GABAergic neurons in MIRAS

Previously, neuropathological autopsy studies have reported a decrease of inhibitory γ-aminobutyric-acid-producing (GABAergic) neurons in patients with *POLG1* mutations[40]. The consequent loss of inhibitory activity in the disease was proposed to underlie their seizures and ataxia[40]. Notably, increased seizure activity was predicted and GABA-related pathway transcripts were decreased in the transcriptome of TBEV-infected MIRAS mouse brains (Figs. 3g and 4g). Histological analysis of the 5 d.p.i. brain samples of MIRAS showed a reduction of GABAergic neurons, with decreased staining of glutamic acid decarboxylase 67 (GAD67, synthesis of GABA from glutamate) and GABA_A receptor (GABRB2), which binds to GABA to exert its inhibitory effect (Fig. 4h). No similar signs were present in uninfected MIRAS mice or in control mice even after infection (Extended Data Fig. 7e). These data indicate that GABAergic interneurons of MIRAS mice are highly sensitive to TBEV infection, resulting in decreased GABAergic inhibition in their brains.

### TBEV inflames the MIRAS liver

In humans, MIRAS is a disease of the liver and the brain: the patients with MIRAS with epilepsy are exceptionally sensitive to the anti-epileptic drug valproate, which causes a subacute liver necrosis in a matter of weeks[11,41]. At 4 d.p.i., the TBEV-infected MIRAS mice already developed multifocal inflammatory cell infiltrations at the hepatic portal triad areas, the arterial walls and in sinusoids with an increased number and size of inflammatory infiltrates compared with the controls (Fig. 5a,b and Extended Data Fig. 7f). The necroptosis marker p-MLKL in MIRAS mouse livers was increased compared with in the controls and POLG1 protein depletion was aggravated (Fig. 5c; no reduction in TFAM). At 9 d.p.i., both MIRAS and control mice showed marked liver damage, displaying severe steatosis, multifocal mononuclear infiltrates (dominated by CD4[+] helper T/CD8[+] cytotoxic T lymphocytes and CD68[+] macrophages). MIRAS livers showed overall increased inflammation and occasional necrotic hepatocytes (Fig. 5a,b,d and Extended Data Fig. 7g,h). Owing to the surprisingly

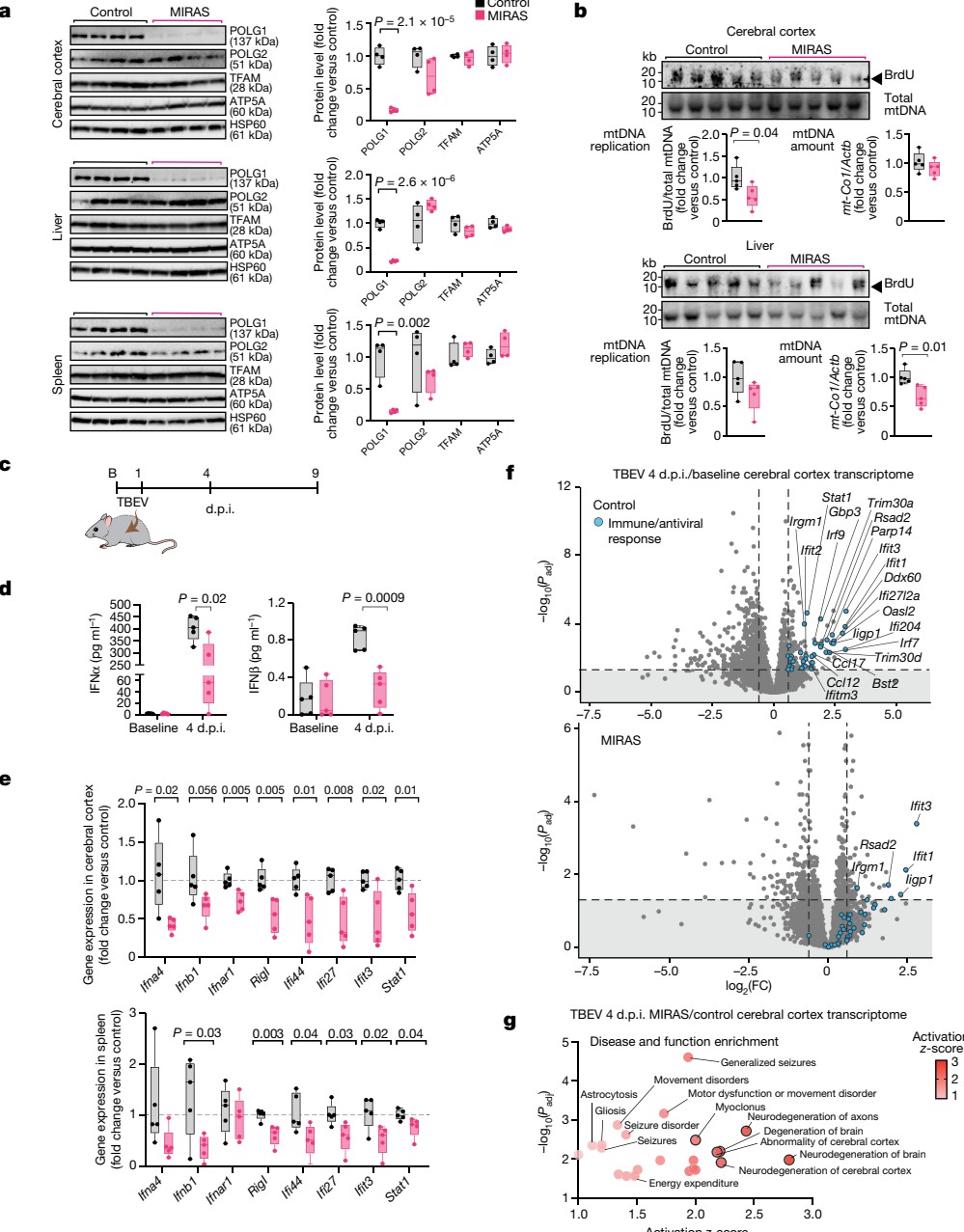

**Fig. 3 | Compromised in vivo activation of antiviral IFN-I signalling in MIRAS mice. a**, mtDNA replisome protein amount in mitochondria isolated from the mouse brain (cerebral cortex), liver and spleen. Western blot and quantification is shown. The loading control was HSP60. $n = 4$ female mice (aged 3 months) per genotype. **b**, mtDNA maintenance in the MIRAS mouse cerebral cortex and liver. mtDNA replication was analysed using south-western blotting for BrdU incorporation into mtDNA (the arrowhead indicates the band of interest for replicating mtDNA detected using anti-BrdU) relative to total mtDNA (Southern blot, mtDNA hybridization); full-length mtDNA is around 16 kb. Bottom left, quantification of BrdU-labelled mtDNA/total mtDNA. Bottom right, the mtDNA levels were assessed using qPCR analysis of mtDNA (*mt-Co1*) relative to nuclear gene (*Actb*). $n = 5$ female mice (aged 3 months) per genotype. **c**, The experimental design of TBEV infection. B, baseline uninfected. **d**, Circulatory IFN-I levels at day 4 after TBEV infection compared with uninfected mice. $n = 5$ female mice (aged 12 months) per condition. **e**, The expression of IFN-I-response components in the mouse cerebral cortex and

spleen at day 4 after TBEV infection (as in **d**). cDNA was analysed using qPCR. The reference gene was *Actb*. **f**, The transcriptome profile of MIRAS and control mouse cerebral cortex on day 4 after TBEV infection compared with the baseline uninfected state (as in **d**). The volcano plot shows significance (adjusted $P$ ($P_{adj}$), Wald test with Benjamini–Hochberg adjustment) and fold change (FC). Immune/antiviral-response-related genes are highlighted in blue. **g**, Disease and function enrichment analysis (Ingenuity pathway analysis) of the cerebral cortex transcriptome of MIRAS mice compared with parallel infected control mice on day 4 after TBEV infection (as in **d**) on transcripts with adjusted $P < 0.05$. Annotations with $P < 0.05$ (Fisher's exact test) with activation $z \geq 1$ are shown; and those with $z \geq 2$, indicating predicted significant activation, are highlighted with a black border. For **a**, **b**, **d** and **e**, the box plots show minimum to maximum values (whiskers), 25th to 75th percentiles (box limits) and median (centre line). Statistical analysis was performed using two-tailed unpaired Student's *t*-tests. See also Extended Data Figs. 6 and 7a,b and Supplementary Table 2.

severe liver inflammation, murine hepatitis virus was excluded. In humans, TBEV causes substantial encephalitis, but it can also cause mild hepatitis[42].

At 9 d.p.i., the amount of TBEV RNA in the liver was still low in all mice. At this point, viral RNA was detected in the brain, with viral antigen expression in neurons in both MIRAS and control mice (Extended

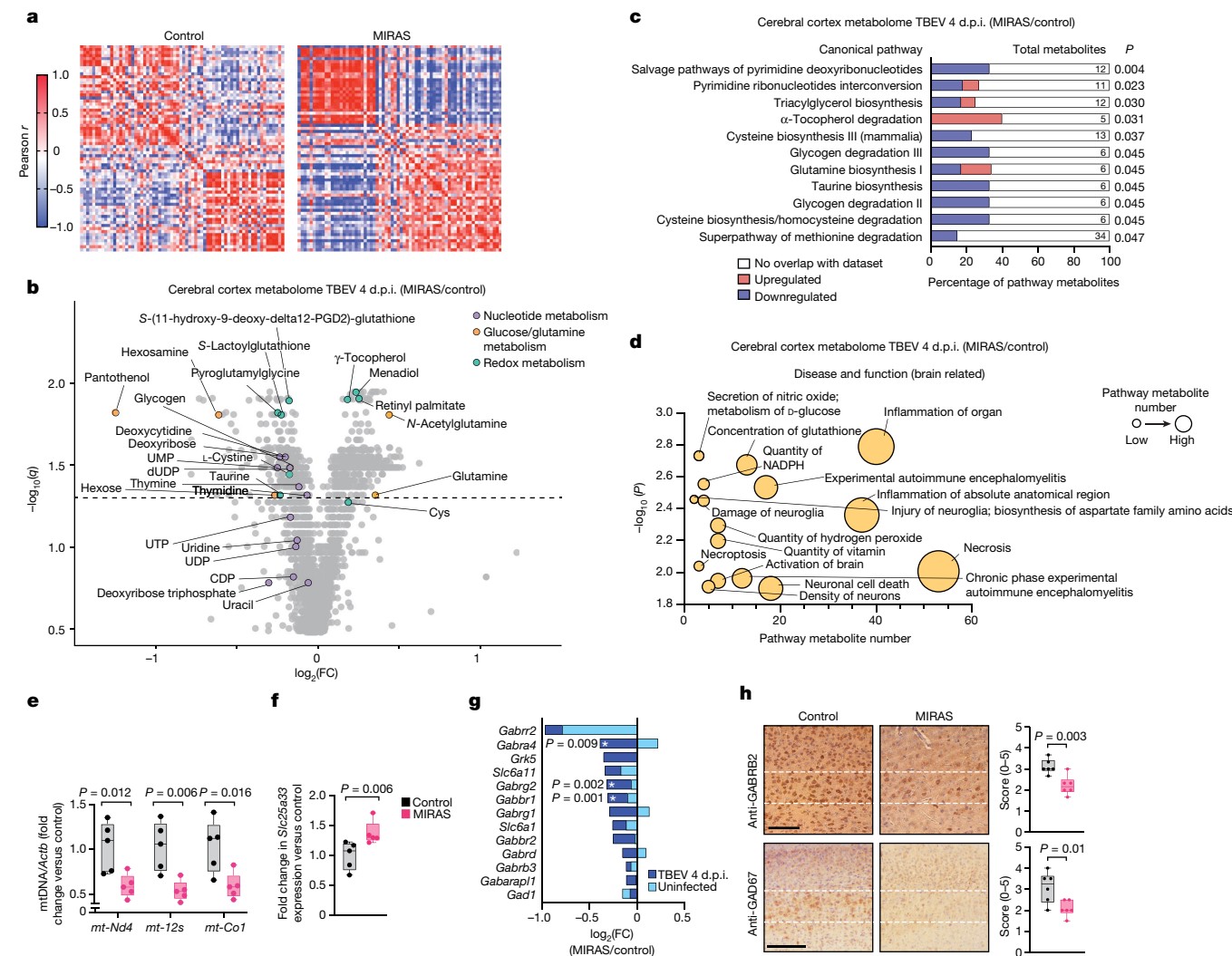

**Fig. 4 | Infection-induced metabolome alteration and acute GABAergic neuronal loss in the mouse brain. a–d**, The metabolome of MIRAS and control cerebral cortex on day 4 after TBEV infection. $n = 5$ female mice (aged 12 months). **a**, Pearson $r$ correlation of metabolites with $P_{adj} < 0.05$. $P$ values were calculated using two-sample Student's $t$-tests with Benjamini–Hochberg multiple-testing correction. **b**, Volcano plot showing significance ($q$, two-sample Student's $t$-test with Storey–Tibshirani multiple-testing correction) and metabolite fold change. The dashed line indicates $q = 0.05$. **c,d**, Canonical pathway (**c**) and brain-related disease and function (**d**) enrichment analyses (Ingenuity pathway analysis) of metabolites ($q < 0.05$ as in **b**). Statistical analysis was performed using Fisher's exact tests. **e**, mtDNA levels in the mouse cerebral cortex on day 4 after infection (as in **a**). qPCR analysis of mtDNA ($mt$-$Nd4$, $mt$-$12s$ and $mt$-$Co1$) relative to nuclear gene ($Actb$) is shown. Statistical analysis was performed using two-tailed unpaired Student's $t$-tests. **f**, RNA-seq analysis of $Slc25a33$

levels in the mouse cerebral cortex on day 4 after infection (as in **a**). Statistical analysis was performed using Wald tests. **g**, RNA-seq analysis of GABAergic-related gene expression in the mouse cerebral cortex at the baseline (uninfected) and on day 4 after TBEV infection. $n = 5$ female mice (aged 12 months) per condition. Statistical analysis was performed using Wald tests; *$P < 0.05$. **h**, GABAergic marker (GABRB2 and GAD67) staining in the mouse neocortical region on day 5 after TBEV infection. The region between the dotted lines shows interneurons in mid-cortical laminar layer 4. Representative image (left) and semiquantitative scoring (right); $n = 6$ female mice (aged 12 months) per condition. Statistical analysis was performed using two-tailed unpaired Student's $t$-tests. Scale bars, 100 μm (top) and 200 μm (bottom). For **e**, **f** and **h**, the box plots show minimum to maximum values (whiskers), 25th to 75th percentiles (box limits) and median (centre line). See also Extended Data Fig. 7c–e and Supplementary Tables 2 and 3.

Data Fig. 7i,j). Brain infection was overall widespread, as viral antigen was generally detected in numerous neurons in olfactory bulb, cortex, brain stem, hippocampus and medulla oblongata. The neurons were often accompanied by mild focal perivascular mononuclear infiltrates in the adjacent parenchyma, consistent with mild nonsuppurative encephalitis, accompanied by mild microgliosis, more pronounced in MIRAS. The infiltrating leukocytes were mainly macrophages (IBA1+), with fewer T cells (CD3+) and rare B cells (CD45R+B220+) (Extended Data Fig. 7j,k). The extent of virus antigen expression and the inflammatory response was comparable between genotypes. Together, these in vivo data show that MIRAS allele sensitizes mice to viral infection.

## High IL-6 in mice and patients with MIRAS

Given the increased liver inflammation in TBEV-infected MIRAS mice, we tested the amount of the cytokine IL-6, which belongs to the acute phase response of pro-inflammatory signalling in the liver and promotes infection-induced immunopathology[43,44]. MIRAS mice showed a moderately increased level already at the baseline, further elevated by day 9. The levels also increased in the controls, reaching those of MIRAS-baseline at 9 d.p.i. (Fig. 5e). Notably, in patients with MIRAS, IL-6 was increased by up to fourfold (samples taken 1 to 25 years after disease onset) and the pro-inflammatory cytokine TNF also was moderately elevated compared with in healthy individuals (Fig. 5f). The results

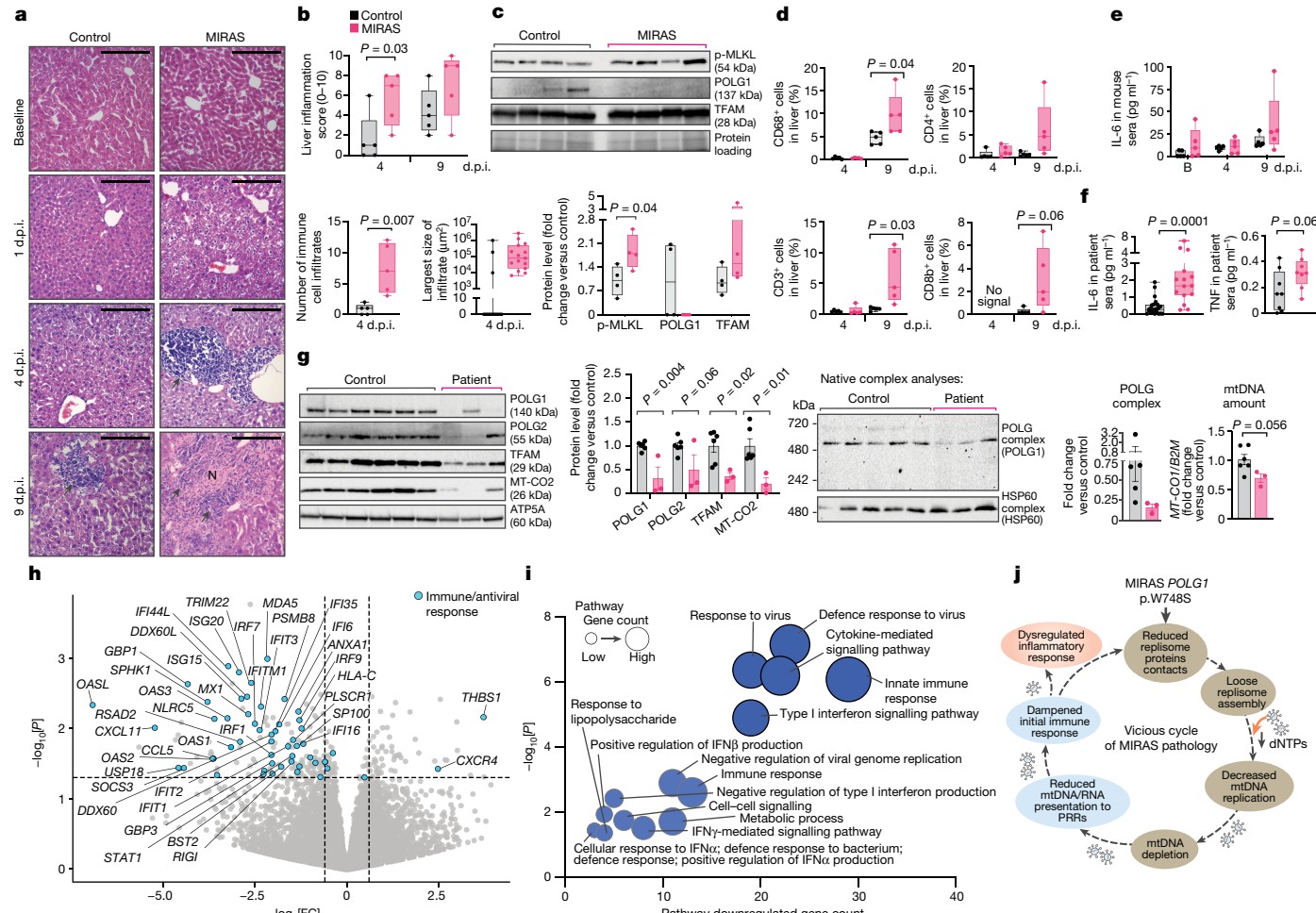

**Fig. 5 | Infection triggered exacerbated liver inflammation and pro-inflammatory circulatory cytokines in MIRAS and compromised mtDNA replisome and antiviral responses in patient brains. a,b,** Liver histopathology after TBEV infection. **a**, Representative haematoxylin and eosin staining. The arrows indicate immune cell infiltration. N, necrotic cells. Scale bars, 200 μm. **b**, Liver inflammation. Top, semiquantitative scoring of the overall severity. Bottom, the total number and size of immune cell infiltrates. $n = 3$ views per mouse, 5 female mice (aged 12 months) per condition. **c**, Liver necroptotic activation on day 4 after TBEV infection (as in **a**). Western blot analysis and quantification is shown. The loading control was protein loading stain. $n = 4$ mice. **d**, Quantification of immune cell marker staining in livers (as in **a**). Representative immunohistochemical staining is shown in Extended Data Fig. 7g. **e**, IL-6 cytokine levels in mouse sera (as in **a**). **f**, IL-6 and TNF cytokines in patient sera. $n = 15$ patients (7 male, 8 female) and 23 controls (9 male, 14 female) (IL-6); and $n = 8$ patients (4 male, 4 female) and 8 controls (3 male, 5 female) (TNF). **g**, The mtDNA replisome components and native complex levels (mitochondrial fractions), their respective quantifications and mtDNA levels (from whole tissue) were analysed in autopsy-derived samples from cerebral cortex. Left, western blot and quantification. The loading control was

ATP5A. Middle, POLG and HSP60 complex were analysed using native complex analyses. Right, mtDNA levels were analysed using qPCR (*MT-CO1* relative to nuclear gene *B2M*). $n = 3$ patients and 6 controls (5 for native complex analysis), all female. **h**, RNA-seq analysis of the transcriptome of patient and control cerebral cortex. The volcano plot shows $P$ values and the fold change (patient/control) of protein-coding transcripts. Statistical analysis was performed using Wald tests. Immune/antiviral response genes ($P < 0.05$) are shown in blue. $n = 3$ patients and 5 controls, all female. **i**, InnateDB pathway analysis of transcripts with $P < 0.05$ in the patient or control cerebral cortex transcriptome (as in **h**). Statistical analysis was performed using hypergeometric tests with Benjamini–Hochberg multiple-testing correction. Pathways predicted downregulated in cerebral cortex (pathway $P < 0.05$) are tabulated. The darker blue nodes indicate pathway $P_{adj} < 0.05$. **j**, Working model of MIRAS disease pathology. For **b–f**, the box plots show minimum to maximum values (whiskers), 25th to 75th percentiles (box limits) and median (centre line). For **g**, data are mean ± s.e.m. Statistical analysis for **b–g** was performed using two-tailed unpaired Student's *t*-tests. See also Extended Data Figs. 7f–j and 8–10 and Supplementary Tables 1, 4 and 5.

support chronic dysregulation of circulating IL-6 and sustained overactive inflammatory responses as a contributor to MIRAS disease manifestation. We propose that the weak early immune response to viruses such as TBEV leads to tissue damage and chronic pro-inflammatory activity, contributing to the progression of MIRAS disease.

## Aberrant patient brain immune pathways

Next, we examined whether any findings similar to MIRAS mice were present in the autopsy-derived brain samples of patients with MIRAS,

compared with in matched control individuals (non-neurological cause of death). The patient brains showed low POLG1 protein (less than 40% of the control mean) and native mtDNA replisome complex amounts in isolated mitochondrial preparations from the cerebral cortex and cerebellum of the patients with MIRAS. mtDNA was depleted by 31% and 46% in the patient cerebral cortex and cerebella, respectively (Fig. 5g and Extended Data Fig. 8a). The cerebral cortex transcriptome of three patients with MIRAS revealed wide downregulation of transcripts encoding immunomodulatory proteins. These included PRRs (*RIGI* and *MDA5*), DExD/H-box RNA helicases (*DDX60*) involved in RIG-I

signalling, 2′,5′-oligoadenylate synthase (OASs; which activates RNase L to degrade viral RNA), antigen-presenting human leukocyte antigens (HLAs), IFN regulatory transcription factors (IRFs), *STAT1* and ISGs (ISGs, IFITs, IFITMs, MXs and TRIMs) (Fig. 5h, Extended Data Fig. 8b and Supplementary Table 4). Only a few inflammatory or cytokine response activating genes were elevated (thrombospondin 1 (*THBS1*), C-X-C motif chemokine receptor 4 (*CXCR4*)) (Fig. 5h). Pathway enrichment analyses of transcriptomes implicated compromised antiviral responses, particularly dampened IFN-I signalling pathway and other anti-pathogen defence pathways, in MIRAS (Fig. 5i and Extended Data Fig. 8c–g). Proteomic analyses also pointed to dysregulated immune signalling and viral pathogenesis pathways in MIRAS (Extended Data Fig. 9a,b and Supplementary Table 5).

The transcriptional signature of dampened innate immune pathways of the patient brains showed a notable overlap with those activated by TBEV infection in the cerebral cortex of wild-type mice, including IFN-I-related and antiviral pathways. However, the conserved antiviral immune defence pathways that TBEV infection typically activates in the host were chronically dampened in patients with MIRAS (Extended Data Fig. 9c and Supplementary Tables 2 and 4). A caveat is that part of the cells relevant for pathogenesis are probably not present anymore in the terminal disease stage.

Taken together, we report here that p.W748S-carrying POLG1 protein results in a decreased amount of mtDNA replisome complex and reduced de novo mtDNA replication in vivo, with consequent mtDNA depletion and lesser presentation of mtDNA/RNA fragments to the cytoplasm as part of the antiviral immune defence. In the mouse brain, viral infection depletes subacutely GABAergic neurons and challenges nucleotide pools, further compromising mtDNA replication and reducing the activation of PRRs and antiviral responses in the early infection. Our evidence proposes that these together with a delayed overactivated pro-inflammatory response contribute to a vicious cycle of MIRAS pathology (Fig. 5j and Extended Data Fig. 10).

## Discussion

Our integrative data from patient materials and mice present a strong connection between innate immunity and the mtDNA replicase POLG. We show that the globally spread MIRAS founder mutation in *POLG* causes a reduced ability to induce a IFN-I-type innate immune response against TBEV, HSV1 and SARS-CoV-2. In the general population, all three viruses cause typically mild symptoms, but some individuals experience delayed severe complications characterized by an overactivated immune response[45]. Such biphasic sequences of infection and severe immunological manifestations mimic the disease onset of POLG epilepsy[15,46]. We propose that aberrant innate immune responses trigger the acute severe epileptic form of MIRAS, clinically mimicking primary HSV-1/TBEV encephalitis. The mechanism involves a decreased release of mitochondrial nucleic acids to cytoplasm, a slowed activation of RIG-I viral sensor, an increased early amplification of the virus and a delayed overactivated inflammatory response. A recent report showed that mtDNA breaks activated RIG-I through mtRNA release[21]. These findings highlight the importance of mtRNA release for innate immunity, which aligns well with our findings in MIRAS. Indeed, we show that, even in the presence of poly(I:C) mimicking viral RNA, MIRAS cells trigger a lowered IFN-I response. These data suggest that mitochondrial nucleic acid release is necessary for full activation of the early-stage antiviral response and that mtDNA replisome is an active component of innate immune responses in vivo.

We identified that the MIRAS allele sensitizes GABAergic neurons to TBEV infection. Selective loss of GABAergic inhibitory interneurons has been reported previously in autopsy samples of patients with *POLG* mutation[40,47]. GABA metabolism and mtDNA maintenance are known to be linked[48], which raises the possibility that the TBEV-induced nucleotide pool imbalance in MIRAS mice shifts GABA homeostasis in their interneurons, reducing inhibitory activity and triggering epilepsy. In addition to their sudden onset of epilepsy, young patients with MIRAS are also extremely sensitive to the common anti-epileptic drug valproate, which triggers a subacute fulminant liver failure typically requiring liver transplantation[5,11,12,41]. We found that MIRAS mice are also sensitized to manifest an acute-onset liver disease. TBEV-infection caused severe hepatic inflammation and activated the MLKL pathway, a known valproate-induced death mechanism[49]. Taken together, we propose a mechanistic sequence for the acute-onset epileptic MIRAS form with valproate hepatotoxicity: (1) virus infection and the MIRAS-related aberrant immune response elicit an epileptic status through subacute loss of GABAergic cortical neurons; (2) the virus causes a subclinical inflammation also in the liver, priming the cells to necroptosis through increased MLKL phosphorylation. The POLG-deficient, mtDNA-depleted mitochondria in the inflamed liver do not oxidize valproate, a fatty acid, triggering toxic inflammatory liver failure through necroptosis (an overview is shown in Extended Data Fig. 10).

In conclusion, our evidence highlights the notable cross-talk of viruses and mtDNA maintenance and shows that these extrinsic factors contribute to the exceptional clinical variability of mitochondrial disease manifestations. Our data present innate immunity mechanisms as therapeutic targets for POLG disorders, relevant also for primary and secondary mitochondrial diseases with similar clinical manifestations.

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

## Methods

### Ethical aspects

Human samples were collected and used with informed consent, according to the Helsinki Declaration and approved by the Ethical Review Board of Kuopio University Central Hospital (410/2019). Animal experimental procedures were approved by the Animal Experimental Board of Finland (ESAVI/689/4.10.07/2015 and ESAVI/3686/2021). Patient and control materials included fibroblasts (established from skin biopsies from individuals' forearms), blood and autopsy-derived brain samples. Control samples were from voluntary healthy individuals (fibroblasts and sera) and, for brains, from people who died acutely with a non-central-nervous-system-disease cause. Autopsy sample collection was approved by the governmental office for social topics and health.

### Antibodies, antisera and kits

Information of the antibodies and oligonucleotide sequences is provided in Supplementary Table 6. Enzyme-linked immunosorbent assay (ELISA) kits for mouse IFNα all subtypes (42115-1), mouse IFNβ (42410-1), mouse IL-6 (BMS603HS), human IL-6 (BMS213HS) and human TNF (HSTA00E) and the CellTiter-Glo Luminescent Cell Viability Assay kit (Promega) were commercially purchased, and assays were performed according to the manufacturer's instructions.

### MIRAS mouse generation

MIRAS knock-in mice were generated and maintained in the C57BL/6JOlaHsd background carrying two variants homologous to mutations of patients with MIRAS on mouse chromosome 7 (NCBI Reference Sequence: NC_000073.7): c.2177G>C into exon 13 (p.W726S); c.3362A>G into exon 21 (p.E1121G). In brief, the pL253 construct carrying exons 4–22 of the *Polg1* genomic region carrying the MIRAS variants was transfected into embryonic stem (ES) cells by electroporation and homologous recombination introduced to the endogenous gene. ES clones with successful recombination were selected based on neomycin resistance. The mutations were confirmed using Southern blot hybridization, PCR and DNA sequencing (DNA-seq). Correct ES clones were injected into blastocysts and implanted into pseudopregnant female mice. Lines with verified germ-line transmission were crossed with mice expressing FLP recombinase to remove the neomycin cassette. The correct genotypes of MIRAS mice were confirmed by DNA-seq. The genotypes were born in Mendelian frequencies, with no gross phenotypic differences between the groups. Mice were housed in controlled rooms at 22 °C under a 12 h–12 h light–dark cycle and with ad libitum access to food and water, and were regularly monitored for weight and food consumption. Further details are provided in Extended Data Fig. 6.

### Cell culture and transfection

Human primary dermal fibroblasts (of the first 8 passages; ±2 passage difference across cell lines of different individuals) that were genetically screened for MIRAS point mutations (by DNA-seq) were used for analyses. Fibroblasts were cultured in DMEM (Lonza; with 4.5 g l$^{-1}$ glucose) supplemented with 10% (v/v) heat-inactivated FBS (Lonza), 50 U ml$^{-1}$ penicillin–streptomycin (Gibco), 0.05 mg ml$^{-1}$ uridine (Calbiochem) and 2 mM GlutaMAX (Gibco) at 37 °C under 5% CO$_2$, with fresh medium replaced every 2 days, and were tested negative for mycoplasma. Transfection of synthetic dsDNA[50] and dsRNA (poly(I:C), Sigma-Aldrich) was performed using FuGENE HD transfection reagent (Promega). In brief, around $2 \times 10^5$ cells were plated onto six-well dishes the day before transfection and transfected with 2.5 µg of dsDNA or dsRNA per well with a 1:2 ratio of nucleic acid:transfection reagent, according to the manufacturer's instructions (sequence details are provided in Supplementary Table 6). For expression of RIG-I or MAVS, fibroblasts were transfected with pcDNA3.1(+)-Flag containing RIG-I (N) or MAVS[51] before poly(I:C) transfection 24 h later and incubated for another 7 h before collection.

### Patient genetic mutation correction in iPSCs

For MIRAS *POLG1* genetic correction, electroporation with CRISPR–Cas9 system components was performed as previously described[52]. We used high-efficiency gRNA and a dsDNA donor template including the desired correction along with a novel restriction site for SalI (GˆTCGAC). A total of 55 monoclonal colonies was individually screened by SalI digestion and successful correction was validated by Sanger sequencing. The chromosomal integrity was confirmed by G-banding performed by Anàlisis Mèdiques Barcelona. A list of the gRNA, donor template and primers for top-six off-target Sanger sequencing (CRISPOR, https://benchling.com) is provided in Supplementary Table 6.

### Differentiation of iPSCs into iFLCs

Induced pluripotent stem cells (iPSCs) were cultured on Matrigel-coated (Corning) plates in E8 medium (Thermo Fisher Scientific) until 90–100% confluency, then split and plated in suspension in ultra-low attachment plates containing hES medium without basic fibroblast growth factor (bFGF) and supplemented with 5 µM ROCK inhibitor (Y-27632, Selleckchem). The medium without ROCK inhibitor was refreshed every other day until day 14, when the aggregates were plated onto gelatin-coated plates containing DMEM/F12 + 20% FBS (Thermo Fisher Scientific) to allow for expansion. The cells were kept for at least 5 passages to obtain induced fibroblast-like cells (iFLCs).

### qPCR

RNA from cells was extracted using the RNeasy kit (Qiagen) according to the manufacturer's instructions. For tissues, homogenization was first performed with ceramic beads using Precellys 24 homogenizer (Precellys) before RNA extraction using the Trizol/chloroform method followed by purification using the RNeasy kit. DNase-treated RNA (normalized across samples) was used for cDNA synthesis using the Maxima first-strand cDNA synthesis kit (Thermo Fisher Scientific) before qPCR using SensiFAST SYBR No-ROX kit (Bioline) and primers (details in Supplementary Table 6) according to the manufacturer's instructions. The amplification level of the assayed gene (2–4 technical replicates per controls and patients) was normalized to *ACTB* and analysed using the $2^{-\Delta\Delta C_t}$ method. mtDNA qPCR was performed on DNA extracted using the DNeasy blood and tissue kit (Qiagen) as described above and previously[53] and normalized to nuclear *ACTB* or *B2M*. For viral RNA analyses, TBEV NS5 RNA[54] or murine hepatitis virus[55] RNA amount was detected using primers and Taqman probes against the targeted viral genome, using the TaqMan Fast Virus 1-Step Master Mix (Thermo Fisher Scientific) according to the manufacturer's instructions. The copy number for TBEV NS5 RNA was determined using a standard curve generated by serial dilution of TBEV-isolated NS5 RNA. Details of the primers are provided in Supplementary Table 6.

### Cytosolic extraction and detection of cytosolic mtRNA/mtDNA

Pelleted cells were resuspended in isolation buffer (20 mM HEPES-KOH pH 7.6, 220 mM mannitol, 70 mM sucrose, 1 mM EDTA, 1× protease inhibitor (Thermo Fisher Scientific)) and divided into two equal fractions: fraction 1, purify total cellular RNA or DNA; and fraction 2, subcellular fractionation to isolate cytosolic RNA or DNA. In brief, fraction 2 was homogenized in around 900 µl of suspension buffer in a handheld Dounce tissue homogenizer with glass pestle (-15 strokes). The homogenate was centrifuged at 800 g for 5 min at 4 °C and the resulting supernatant was centrifuged at 12,000 g for 10 min at 4 °C. The supernatants were collected and centrifuged at 17,000 g for 15 min at 4 °C to purify the cytosolic fraction. The whole-cell (fraction 1) and cytosolic (of fraction 2) fractions were subjected to DNA or RNA purification using the RNeasy Kit or DNAeasy Blood and Tissue Kit (Qiagen) and eluted into

an equal volume of water. RNA eluate was treated with DNase before cDNA production. Equal volume of cDNA or DNA eluate were used for qPCR using nuclear gene primers (*ACTB* or *B2M*) or mitochondrial genome-specific primers (*MT-CYB* and *MT-CO1*). mtDNA/RNA abundance in whole cells served as normalization controls for their values obtained from cytosolic fractions[18]. The purity of cytosolic fraction was examined by western blotting.

### In vivo BrdU labelling and south-western analyses
Mice receiving an intraperitoneal injection of 300 µg of BrdU (BD Biosiences) per gram of mouse weight were euthanized 24 h after injection. DNA was isolated by routine phenol–chloroform extraction. XhoI-digested DNA was separated using agarose gel electrophoresis and blotted onto Hybond N+ membranes (Amersham) as described previously[53]. Immunodetection was performed using anti-BrdU antibodies, and total mtDNA was detected using Southern hybridization as described previously[56].

### Viral stocks and infections of fibroblasts
The European subtype of TBEV was isolated from human neuroblastoma cells (SK-N-SH; passage 1) infected with tick collected in Finland[57]; SARS-CoV-2 was isolated from a patient with COVID-19 on human non-small cell lung cancer (Calu-1) cells[58], passaged on African green monkey kidney (Vero E6) cells expressing type II membrane serine protease 2 (TMRSS2) via lentivirus transduction[59]; the KOS strain of herpes simplex virus 1, HSV-1[60], was passaged on Vero cells. SK-N-SH (https://www.atcc.org/products/htb-11), Calu-1 (https://www.atcc.org/products/htb-54), Vero E6 (https://www.atcc.org/products/crl-1586) and Vero (https://www.atcc.org/products/ccl-81) cells were purchased from ATCC. The virus work was performed under bio-safety level 3 (BSL-3) conditions for TBEV and SARS-CoV-2 and under BSL-2 conditions for HSV-1. The ability of viruses to infect fibroblasts was tested by inoculating cells grown on a 96-well plate with serially tenfold diluted virus stocks and the optimal viral dilution was selected based on the dilution showing the most prominent difference in infected cells number between wild-type control and MIRAS cells using immunofluorescence.

For fibroblast infection, around $2 \times 10^5$ fibroblast cells were grown on six-well plates the day before (or ~$1 \times 10^5$ iFLCs 2 days before) being inoculated with 500 µl of 1:20 diluted TBEV, 1:10 diluted SARS-CoV-2 or 1:5,000 diluted HSV-1 (multiplicity of infection (MOI) of ~0.1–1). After 1 h (at 37 °C, 5% $CO_2$), the inocula were removed, the cells were washed twice with conditioned medium, 3 ml of fresh medium was added to each well and the plates were incubated at 37 °C under 5% $CO_2$ for 6, 24 or 48 h. Non-treated cells that were plated simultaneously alongside those subjected to viral infection were used as the uninfected control. At the end of incubation, the cells were washed twice with PBS and were lysed in RIPA buffer (50 mM Tris, 150 mM NaCl, 1% Triton X-100, 0.1% SDS, 0.5% sodium deoxycholate, pH 8.0) supplemented with EDTA-free protease inhibitor cocktail (Roche), at 150 µl per well for western blotting analyses. For DNA/RNA analyses, 60 µl of RIPA lysate was mixed with TRIzol Reagent (Thermo Fisher Scientific) before DNA or RNA extraction and RT–qPCR or qPCR as described in relevant Methods section. For the immunofluorescence assay, infected cells were fixed with 4% paraformaldehyde (PFA, in PBS) and incubated for 15 min at room temperature. The cells were washed once with PBS, permeabilized for 5 min at room temperature with Tris-buffered saline, pH 7.4 supplemented with 0.25% Triton X-100 and 3% (w/v) of bovine serum albumin, and replaced with PBS. Virus inactivation was confirmed by UV-inactivation with a dose of 500 mJ cm$^{-2}$ before incubation with primary antibodies and processed as described below.

### Immunofluorescence microscopy
The PFA-fixed viral-infected cells were stained with primary antibodies (Supplementary Table 6) overnight at 4 °C and for 1 h at room temperature with secondary antibodies. Three washes with PBS were included between each step. Coverslips were mounted with VECTASH-IELD anti-fade mounting medium containing DAPI (Vector Laboratories). Images were acquired using the Zeiss AxioImager epifluorescence microscope. Quantification of the immunofluorescence signal was performed using CellProfiler (v.4.2.6)[61].

### Gel electrophoresis and western blotting
Cells lysed in RIPA buffer (150 mM NaCl, 1% Triton X-100, 0.5% sodium deoxycholate, 0.1% SDS, 50 mM Tris-Cl, pH 8.0) were measured for protein concentration using the BCA assay (Pierce) and equal amounts of protein samples were resuspended into SDS–PAGE loading dye (50 mM Tris-Cl, pH 6.8, 100 mM dithiothreitol, 2% (w/v) sodium dodecyl sulphate, 10% (w/v) glycerol, 0.1% (w/v) bromophenol blue), boiled for 5–10 min at 95 °C before SDS–PAGE analysis using the 4–20% gradient gel (Bio-Rad) according to the manufacturer's instructions.

For mitochondrial protein analyses, mitochondria were isolated from tissue using differential centrifugation as described previously[62]. The clarified mitochondrial pellets were resuspended into buffer (20 mM HEPES-KOH pH 7.6, 220 mM mannitol, 70 mM sucrose, 1 mM EDTA) and analysed using SDS–PAGE, or solubilized using 1% (w/v) n-dodecyl-β-ᴅ-maltoside (DDM) in 1.5 M α-amino *n*-caproic acid for 30 min on ice for blue-native (BN) electrophoresis analysis. DDM-solubilized samples were centrifuged at 20,000$g$ for 20 min at 4 °C. The clarified supernatants were measured for protein concentration using the BCA assay and equal amounts of protein samples were mixed with BN loading dye (0.25% (w/v) Coomassie blue G250 (MP Biomedicals), 75 mM α-amino *n*-caproic acid) before BN electrophoresis using cathode buffer (50 mM tricine, 15 mM Bis-Tris, pH 7.0, 0.02% (w/v) Coomassie blue G250) and anode buffer (50 mM Bis-Tris, pH 7.0) on self-casted 1-mm-thick 5–12% gradient polyacrylamide gels. Separation part of the gel was prepared by mixing solution of 5 and 12% acrylamide (acrylamide:bisacrylamide 37.5:1) in 0.5 M α-amino n-caproic acid, 50 mM Bis-Tris (pH 7.0), 11 or 20% (w/v) glycerol, 0.027% ammonium persulfate, 0.1% TEMED. Separation gel was overlaid with a 4% acrylamide stacking gel solution as described above (no glycerol; but 0.084% ammonium persulfate, 0.17% TEMED).

After electrophoresis, gels were transferred onto 0.45 µm PVDF membranes using a semidry transfer (SDS–PAGE) or wet transfer (BN-PAGE) apparatus (Bio-Rad) before western blotting using the desired antibodies (details are provided in Supplementary Table 6). Images were obtained using ChemiDoc XRS+ imaging machine (Bio-Rad) and signals were quantified using Image Lab (v.6.1.0 build 7; Bio-Rad) according to the manufacturer's instructions. The protein-of-interest signal was normalized to the loading control signal in the sample.

### Mouse behavioural analyses
**Treadmill.** An Exer-6M treadmill (Columbus Instrument) was used as described previously[63]. The tests were completed as a set of five independent trials over 1 h. The running time was counted when the mouse stopped for five continuous seconds and did not continue.

**Rotarod.** The rotating rod system (Rota-Rod; Ugo Basile, 47600) with a PVC drum (diameter of 44 mm) was used as described previously[64]. The animals were trained for three consecutive days before the test.

**Footprint analyses to detect ataxia.** Mouse feet were painted with non-toxic washable paint (separate colours for hind- and forelimbs) and the mouse was allowed to walk through a tunnel on paper. The stride length and width were measured. Scoring data were obtained using at least two consecutive steps from each foot.

**Infection of mice, histology and immunohistochemistry.** Mice were transported to the BSL-3 facility and acclimatized to individually

ventilated biocontainment cages (ISOcage; Scanbur) for 7 days before being inoculated intraperitoneally with 1,000 plaque-forming units of TBEV. Mice were euthanized at the indicated days after infection and sera were collected for cytokine analyses using commercially purchased ELISA kits (see the 'Antibodies, antisera and kits' section). For DNA, RNA or protein analyses (see the relevant Methods section), tissues were collected into TRIzol Reagent (Thermo Fisher Scientific). For histology, liver samples were fixed in cold 4% (v/v) PFA in PBS and incubated in PBS supplemented with 30% (w/v) sucrose at 4 °C for 3 days before routine embedding in OCT compound and trimmed into sections with a thickness of 6–8 μm for haematoxylin and eosin or ORO staining according to the standard protocol[65]. For immunohistochemical staining, liver sections were stained with the following antibodies: CD3 (T cell marker), CD4 (helper T cell marker), CD8b (cytotoxic T cell marker) or CD68 (macrophage marker) using the ImmPRESS HRP goat anti-rat IgG (Mouse Adsorbed) Polymer Kit (Vector Laboratories, MP-7444), and with haematoxylin counterstaining according to the manufacturer's instructions. Liver inflammation severity was semi-quantitatively scored and the total number of immune cell infiltrations was quantified from three unique visual fields at ×5 magnification (15,370,559 μm$^2$ per view) per mouse liver section. The area (μm$^2$) of the largest infiltrate detected per view was measured using ImageJ (2.0.0-rc-69/1.52n; https://imagej.net/ij/). Liver ORO and CD protein signal was quantified using CellProfiler (v.4.2.6)[61] after pixel classification using ilastik (v.1.3.3)[66].

For brain histology, brain halves (cut in midline) were fixed in PFA for 48 h, then stored in 70% (v/v) alcohol until processing. They were trimmed and routinely paraffin-wax embedded. Consecutive sections (3–4 μm) were prepared and stained with haematoxylin and eosin or subjected to immunohistochemical staining for TBEV antigen, CD3 (T cell marker), CD45R/B220 (B cell marker) and IBA1 (marker of macrophages and microglial cells), according to previously published protocols[67,68]. Mouse brain GABAergic marker staining was performed using GAD67 and GABRB2 antibodies followed by blinded semi-quantitative scoring by A.P. (neuropathologist). Details of the antibodies are provided in Supplementary Table 6.

**Bulk RNA-seq analysis.** RNA-seq was performed at the Biomedicum Functional Genomics Unit (University of Helsinki) according to the Drop-seq protocol as described previously[69,70]. A total of 10 ng of extracted RNA was used as the starting material. The quality of the sequencing libraries was assessed using the TapeStation DNA High Sensitivity Assay (Agilent). The libraries were sequenced on the Illumina NextSeq 500 system[70]. For read alignment and generation of digital expression data, raw sequencing data were inspected using FastQC and multiQC[71,72]. Subsequently, reads were filtered to remove low-quality reads and reads shorter than 20 bp using Trimmomatic[73]. Reads passing the filter were then processed further using Drop-seq tools according to the pipeline described[69] (v.2.3.0). In brief, the raw, filtered read libraries were converted to sorted BAM files using Picard tools (http://broadinstitute.github.io/picard). This was followed by tagging reads with sample specific barcodes and unique molecular identifiers (UMIs). Tagged reads were then trimmed for 5′ adapters and 3′ poly A tails. Alignment ready reads were converted from BAM-formatted files to fastq files that were used as an input for STAR aligner[74]. Alignments were performed using the GRCm38 (mouse) reference genome and GENCODE mouse release 28 or the GRCh38 (human) reference genome and GENCODE human release 33 comprehensive gene annotation files[75] with default STAR settings. After the alignment, the uniquely aligned reads were sorted and merged with the previous unaligned tagged BAM file to regain barcodes and UMIs that were lost during the alignment step. Next, annotation tags were added to the aligned and barcode-tagged BAM files to complete the alignment process. Finally, Drop-seq tools were used to detect and correct systematic synthesis errors present in sample barcode sequences. Digital expression matrices were then created by counting the total number of unique UMI sequences (UMI sequences that differ by only a single base were merged together) for each transcript. Differential expression analysis was performed with DESeq2 (using the default settings) in the R environment[76].

**Untargeted metabolomics.** Metabolites were extracted from 20 mg of mouse cerebral cortex in hot ethanol. In brief, frozen samples were homogenized in 0.5 ml 70% (v/v) ethanol with ceramic beads using a Precellys 24 homogenizer (Precellys). Before and after homogenization, the samples were kept frozen (at ≤−20 °C). The samples were transferred to a 15 ml tube with washing using 0.5 ml of 70% (v/v) ethanol. To each tube, we added 7 ml of 70% (v/v) ethanol that was preheated at 75 °C, immediately vortexed and placed the sample into a water bath at 75 °C for 1 min followed vortexing once. The content was cooled down in cold bath at −20 °C before being centrifuged for 10 min (4 °C). The clear supernatant was transferred to a new tube and stored at −80 °C until analysis using mass spectrometry (MS).

Untargeted metabolite profiling was performed using flow injection analysis on the Agilent 6550 QTOF instrument (Agilent) using negative ionization, 4 GHz high-resolution acquisition and scanning in MS1 mode between $m/z$ 50 and $m/z$ 1,000 at 1.4 Hz. The solvent was 60:40 isopropanol:water supplemented with 1 mM NH4F at pH 9.0, as well as 10 nM hexakis(1H,1H,3H-tetrafluoropropoxy)phosphazine and 80 nM taurochloric acid for online mass calibration. The seven batches were analysed sequentially. Within each batch, the injection sequence was randomized. Data were acquired in profile mode, centroided and analysed using MATLAB (MathWorks). Missing values were filled by recursion in the raw data. After identification of consensus centroids across all of the samples, ions were putatively annotated by accurate mass and isotopic patterns. Starting from the HMDB v.4.0 database, we generated a list of expected ions including deprotonated, fluorinated and all major adducts found under these conditions. All formulas matching the measured mass within a mass tolerance of 0.001 Da were enumerated. As this method does not use chromatographic separation or in-depth MS2 characterization, it is not possible to distinguish between compounds with an identical molecular formula. The confidence of annotation reflects level 4 but, in practice, in the case of intermediates of primary metabolism, it is higher because they are the most abundant metabolites in cells. The resulting data matrix included 1,943 ions that could be matched to deprotonated metabolites listed in HMDB v.3.0.

**Proteomics.** Protein was extracted from 50 mg of frozen brain autopsy samples using TRIzol Reagent (Thermo Fisher Scientific) according to the manufacturer's instructions. Extracted protein pellets were resuspended into 100 μl of buffer containing 6 M urea, 50 mM ammonium bicarbonate, pH 8 and boiled for 5–10 min at 95 °C. The protein concentration was estimated using the BCA assay (Pierce) and equal amounts of protein samples were aggregated on amine beads[77]. For on-bead digestion, 50 mm ammonium bicarbonate buffer was added to the beads. Proteins were reduced with 10 mM DTT for 30 min at 37 °C and alkylated with 20 mM IAA for 30 min at room temperature in dark, after which 0.5 μg of trypsin was added, and trypsin digestion was performed overnight at 37 °C. Beads were separated using a magnet, the supernatant was transferred to new tube and acidified, and the tryptic peptides were desalted using C$_{18}$ StageTips for MS analysis. Liquid chromatography coupled with tandem MS (LC–MS/MS) analysis of the resulting peptides was performed using the Easy nLC1000 liquid chromatography system (Thermo Electron) coupled to a QExactive HF Hybrid Quadrupole-Orbitrap mass spectrometer (Thermo Electron) with a nanoelectrospray ion source (EasySpray, Thermo Electron). The LC separation of peptides was performed using the EasySpray C18 analytical column (2 μm particle size, 100 Å, 75 μm inner diameter and 25 cm length; Thermo Fisher Scientific). Peptides were separated over a 90 min gradient from 2% to 30% (v/v) acetonitrile in 0.1% (v/v) formic

acid, after which the column was washed using 90% (v/v) acetonitrile in 0.1% (v/v) formic acid for 20 min (flow rate 0.3 µl min⁻¹). All LC–MS/MS analyses were operated in data-dependent mode where the most intense peptides were automatically selected for fragmentation by high-energy collision-induced dissociation. For data analysis, raw files from LC–MS/MS analyses were submitted to MaxQuant (v.1.6.7.0)[78] for peptide/protein identification and label-free quantification. Parameters were as follows: carbamidomethyl (C) was set as a fixed modification; protein *N*-acetylation and methionine oxidation as variable modifications; first search error window of 20 ppm and main search error of 6 ppm; the trypsin without proline restriction enzyme option was used, with two allowed miscleavages; minimal unique peptides was set to one; and the FDR allowed was 0.01 (1%) for peptide and protein identification. The UniProt human database (September 2018) was used for the database searches. MaxQuant output files (proteinGroups.txt) were loaded into Perseus (v.1.6.1.3)[79] for further data filtering and statistical analysis. Identifications from potential contaminants and reversed sequences were removed, and normalized intensities (LFQ) were log₁₀-transformed. Next, a criteria of at least 50% valid values in at least one group was used to filter the results. All zero intensity values were replaced using noise values of the normal distribution of each sample. Protein abundances were compared using a two-sample Student's *t*-test with $P < 0.05$ as the criteria for a statistically significant difference between the two groups.

**Functional and pathway enrichment analyses.** Qiagen Ingenuity Pathway Analyses (Qiagen; https://digitalinsights.qiagen.com/IPA), g:Profiler[80] (https://biit.cs.ut.ee/gprofiler) toolset and KEGG database[81] were used for the analyses of transcriptome, metabolome and/or proteome datasets. For immune pathway analyses, we further used the manually curated InnateDB database[82] (https://www.innatedb.com/index.jsp).

**Genotype–phenotype association analyses.** Analyses were performed on the data from the FinnGen study, a large-scale genomics initiative that has analysed Finnish Biobank samples and correlated genetic variation with health data to understand disease mechanisms and predispositions[6]. The mixed-model logistic regression method SAIGE (R package developed with Rcpp for genome-wide association tests in large-scale datasets and biobanks) was used for association analysis and included the following covariates in the model: sex, age, genotyping batch and ten principle components. These results are from 3,095 end points, 16,962,023 variants and 309,154 individuals in data freeze 7 (https://r7.finngen.fi/).

**Statistical analyses.** Statistical analyses as described in the figure legends were performed either using Microsoft Excel v.16.80, GraphPad Prism v.10.1.1 for macOS (GraphPad, www.graphpad.com) or using toolsets as indicated in the respective figure legends and in relevant method sections. GraphPad Prism v.10.1.1 as described above was used to create box and whisker plots using the standard five-number summary: minimum, lower quartile (25th percentile), median (50th percentile), upper quartile (75th percentile) and maximum, with whiskers extending down to the minimum and up to the maximum value; bar charts show the mean ± s.e.m. The datapoints for each value are superimposed on the plot. No statistical methods were used to predetermine the sample size. Sample sizes were chosen to ensure adequate power and to account for potential interindividual/animal, gender and age variance (age- and sex-matched samples were used as controls). The number of biologically independent mouse or human samples is described in the respective figure legends.

**Reporting summary**
Further information on research design is available in the Nature Portfolio Reporting Summary linked to this article.

## Data availability

The mouse RNA-seq data generated in this study have been deposited at the NCBI Gene Expression Omnibus (GEO) and are accessible through GEO series accession number GSE249432. Metabolomic data have been deposited at the MassIVE database and are accessible through MSV000093634. Human omics data sharing is restricted owing to European general data protection regulations (GDPR) laws. Individual enquiries about expression changes of specific genes/proteins can be directed to the corresponding author. Numerical source data giving rise to graphical representation and statistical description in Figs. 1–5 and Extended Data Figs. 1–9 are provided as source data and in Supplementary Tables 2–5. Characteristics of human research participants and relevant materials used in this study are provided in Supplementary Tables 1 and 6, respectively. Uncropped images of Southern blots and immunoblots presented in the figures are included in Supplementary Fig. 1. Source data are provided with this paper.

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

**Acknowledgements** We thank M. Innilä for coordination of patient and control sampling, for expertise in histology and for collecting autopsy controls; L. Euro for expert advice on native gel analyses; J. Arola for expert consultation of liver histology; B. Hollmann, S. Malinen, S. M. Jansson, A. Zhaivoron, V. Venkat and K. J. Aaltonen for technical help; T. Sironen for advice on mouse infection experiments; T. Rolova and J. Koistinaho for technical expertise; M. James and J. Väänänen for RNA-seq and bioinformatic analyses; P. Tienari for comments on the manuscript; H. Kallankari and A. M. Saukkonen for recruitment of patients with MIRAS; the members of the Laboratory Animal Center in the University of Helsinki and Transgenic Core Facility in the University of Oulu for animal husbandry and experimental support; the staff at the Biomedicum Functional Genomics Unit at the Helsinki Institute of Life Science and Biocenter Finland at the University of Helsinki for RNA-seq, bioinformatic and genomic services; the members of the Biomedicum Imaging Unit for imaging equipment and expertise; and the technical staff at the Histology Laboratory, Institute of Veterinary Pathology, Vetsuisse Faculty, University of Zurich for technical support. MS-based proteomic analyses were performed by the Proteomics Core Facility, Department of Immunology, University of Oslo/Oslo University Hospital, which is supported by the Core Facilities program of the South-Eastern Norway Regional Health Authority. This core facility is also a member of the National Network of Advanced Proteomics Infrastructure (NAPI), which is funded by the Research Council of Norway INFRASTRUKTUR-program (project number, 295910). Metabolomics analyses at the Swiss Multi-Omics Center were supported by the Swiss initiative on Personalized Health and Related Technology by the ETH Domain. We acknowledge the participants and investigators of the FinnGen study. This work was supported by the Jane and Aatos Erkko Foundation (to O.V. and A.S.), the Academy of Finland (to A.S., Y.K., J.H., O.V. and J.U.), the Sigrid Jusélius Foundation, University of Helsinki and Helsinki University Hospital (to O.V. and A.S.); European Molecular Biology Organization's postdoctoral fellowship (to Y.K.); and the POLG foundation and Business Finland (to A.S.).

**Author contributions** Y.K. developed the concept, tested hypotheses, designed and performed the experiments, analysed data, interpreted results, and wrote and edited the manuscript. J.H. supervised and advised the experimental designs for viral infection of cultured cells, performed the viral infection procedure and cell viability assay, and advised on data interpretation. R.S.M. performed experiments on gene editing for *POLG* disease mutation correction, and validated iPSCs and their differentiation. T. Mito performed quantification of microscopy data and contributed to analysis. M.T. contributed to mouse phenotyping, including behavioural tests and mitochondrial functions. T. Manninen generated the MIRAS mouse model, performed genetic verification, phenotyping and analysed responses. R.K. and L.K. designed, supervised and performed the mouse infection procedure. R.V. and J.U. contributed to patient recruitment plan, examination, sampling (fibroblasts, sera) and data analysis. T.A.N. and S.S. performed the MS-based proteomics analysis. K.W. supervised cell culture work, and wrote and commented on the manuscript. A.O. and N.Z. contributed the metabolomics analysis and supported in the interpretation. A.P. supervised, analysed and interpreted GABAergic pathology in mouse brain and co-analysed mouse liver histology. A.K. analysed and interpreted data on mouse brain histology with viral infection. O.V. supervised the virus experimentation, provided viral stocks, reagents and advice on viral infection and data interpretation. A.S. conceived and developed the concept, supervised the overall project and experimental design, interpreted results, and wrote and edited the manuscript.

**Competing interests** The authors declare no competing interests.

**Additional information**
**Correspondence and requests for materials** should be addressed to Anu Suomalainen.

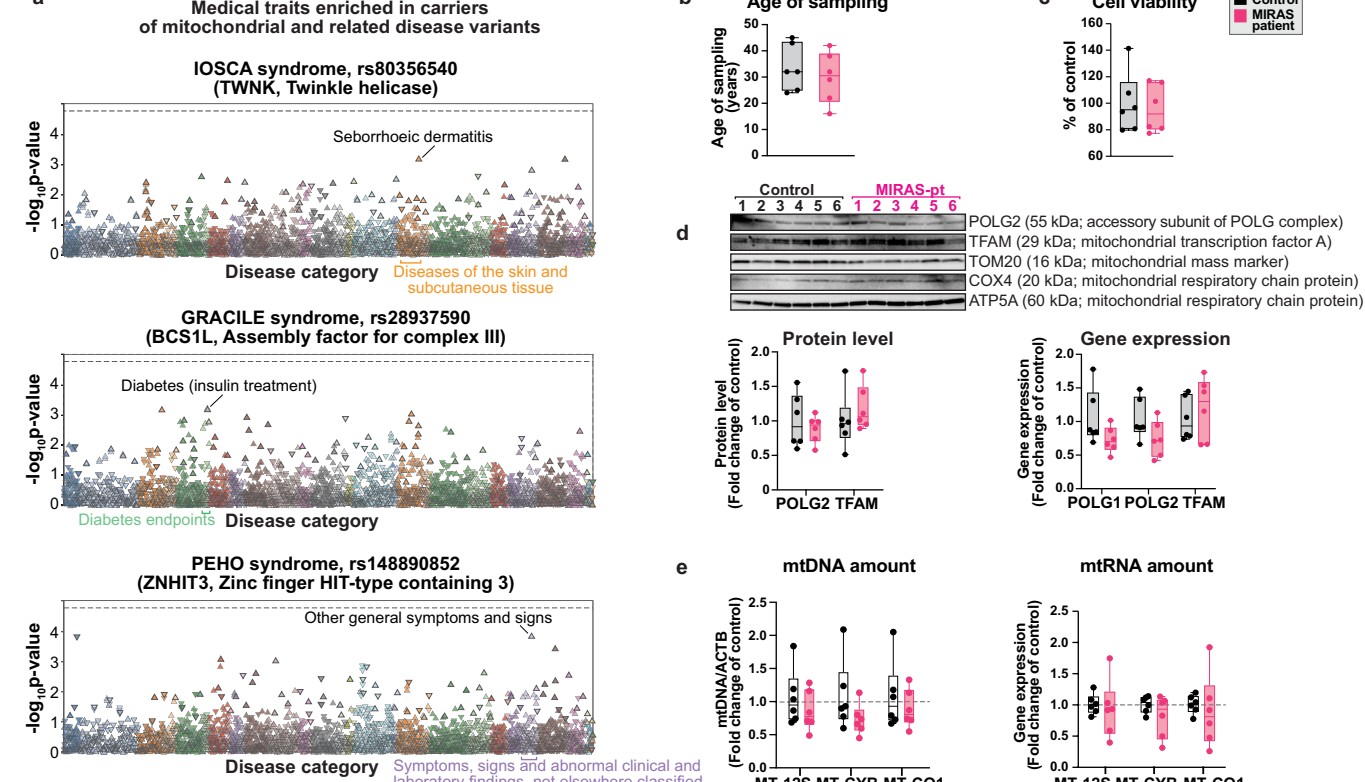

**Extended Data Fig. 1 | FinnGen data and characterization of MIRAS patient fibroblasts.** (a) Genotype-phenotype association; FinnGen database. Clinical phenotype enriched in carriers of mitochondrial and related disease gene variants: rs80356540, c.1523A>G, p.Tyr508Cys in TWNK (Twinkle helicase) causing IOSCA (infantile onset spinocerebellar ataxia); rs28937590, c.232A>G, p.Ser78Gly in BCS1L (assembly factor for complex III) causing GRACILE syndrome (complex III deficiency); and rs148890852, c.92C>T, p.Ser31Leu in ZNHIT3 (Zinc finger HIT-type containing 3) causing PEHO (Progressive encephalopathy with oedema, hypsarrhythmia, and optic atrophy). Significance (p-value) and disease categories tabulated. Triangles: diseases or traits. Upward-pointing triangle: positive association and vice versa. Dotted line: cut-off for significance; Mixed model logistic regression method SAIGE. Data are from ref. 6. (b) Primary fibroblast characteristics: Age-of-sampling, MIRAS patients and control individuals. N = 6 for both patients (all homozygous for p.W748S + E1143G, females) and age- and gender-matched healthy control. Mean age of control group: 33 years, MIRAS patients: 30 years. Refer to Supplementary Table 1 for patient sample details. (c) Cell viability of control and MIRAS patient fibroblasts. N = 6 female controls and patients. (d) mtDNA replisome protein and gene expression in control and MIRAS patient fibroblasts. Western blot and quantification, loading control HSP60 (See Fig. 1b for POLG1 and HSP60 data); Gene expression, qPCR of cDNA, reference gene *ACTB*. N = 6 female controls and patients. (e) mtDNA and mtRNA amount in control and MIRAS patient fibroblasts. qPCR of DNA or cDNA (*MT-12S* (or *MT-RNR1*), *MT-CYB* and *MT-CO1* relative to nuclear/reference gene *ACTB*). N = 6 female controls and patients. In **b**, **c**, **d** and **e**, box plots show minimum-25th-50th(median)-75th percentile-maximum (whiskers extend to the smallest and largest value); two-tailed unpaired student's t-test. *Abbreviations: Pt, patient.*

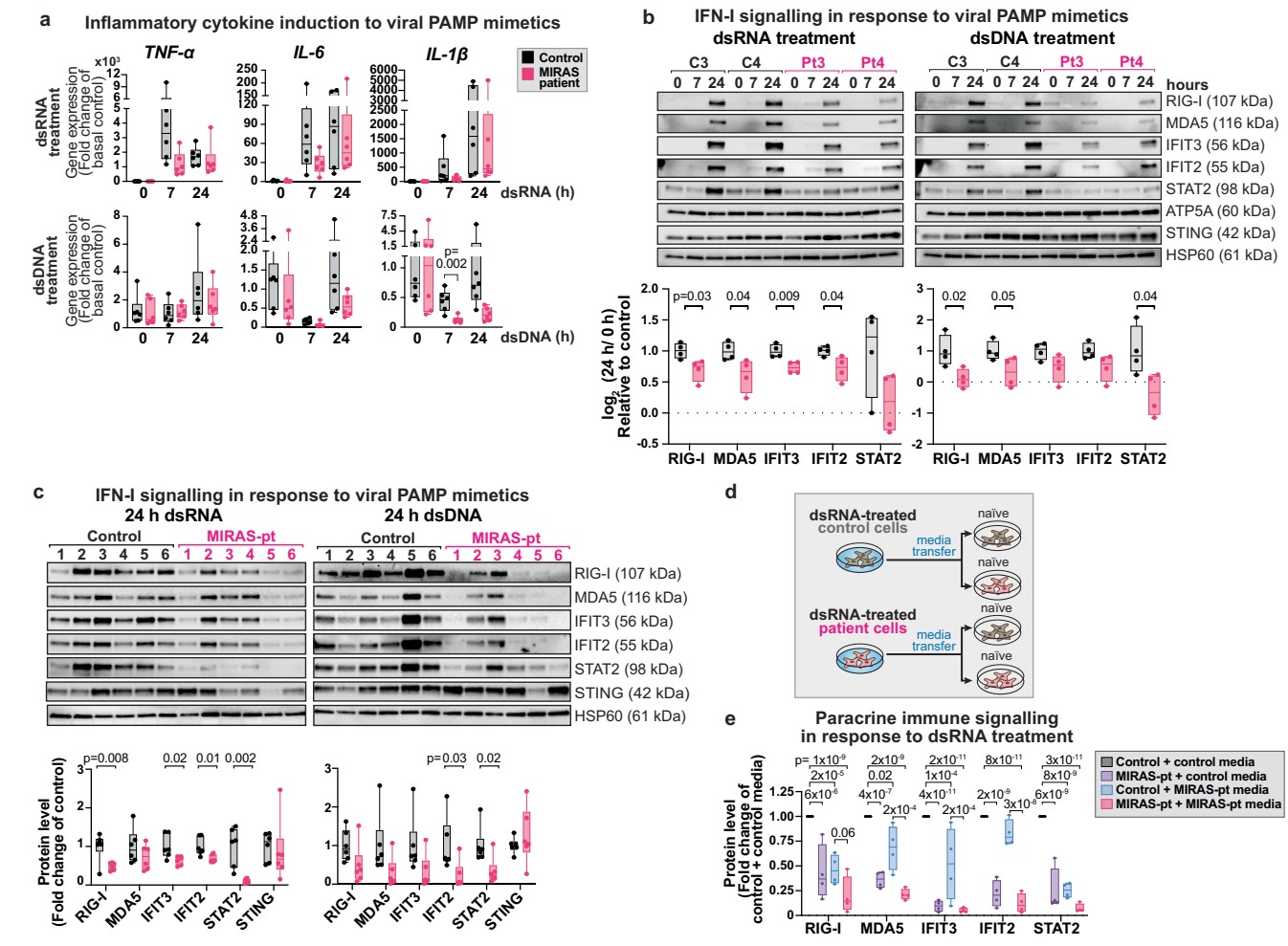

**Extended Data Fig. 2 | Intracellular and paracrine IFN-I signalling in MIRAS fibroblasts exposed to viral PAMP mimetics.** (a) Inflammatory cytokine gene induction by viral PAMP mimetics (dsRNA/poly(I:C) or dsDNA) in fibroblasts. qPCR of cDNA, reference gene *ACTB*. N = 6 female controls and patients. (b) IFN-I signalling pathway proteins; fibroblasts exposed to viral PAMP mimetics (dsRNA/poly(I:C) or dsDNA). Western blot of two controls and patients in addition to the two controls and patients shown in Fig. 1e. Quantification of 24 h treatment protein level relative to basal condition of each cell line (n = 4 female controls and patients), loading control HSP60. (c) IFN-I signalling pathway proteins in fibroblasts at 24 h post dsRNA/poly(I:C) or dsDNA treatment. Western blot, quantification, loading control HSP60.

N = 6 female controls and patients. (d) Experimental setup; transfer of cell culture media from 24 h dsRNA/poly(I:C)-treated control or patient cells to naïve cells for another 24 h incubation to investigate paracrine immune signalling. (e) Paracrine immune signalling of fibroblasts as illustrated in **d**. Quantification of western blot, loading control HSP60. N = 4 female controls and patients. Refers to representative western blot in Fig. 1f. In **a**, **b**, **c** and **e**, box plots show minimum-25th-50th(median)-75th percentile-maximum (whiskers extend to the smallest and largest value); two-tailed unpaired student's t-test in **a**, **b** and **c**, 2-way ANOVA with Tukey's multiple comparisons test in **e**. *Abbreviations: C, control; Pt, patient; IFN, interferon; PAMP, pathogen-associated molecular pattern.*

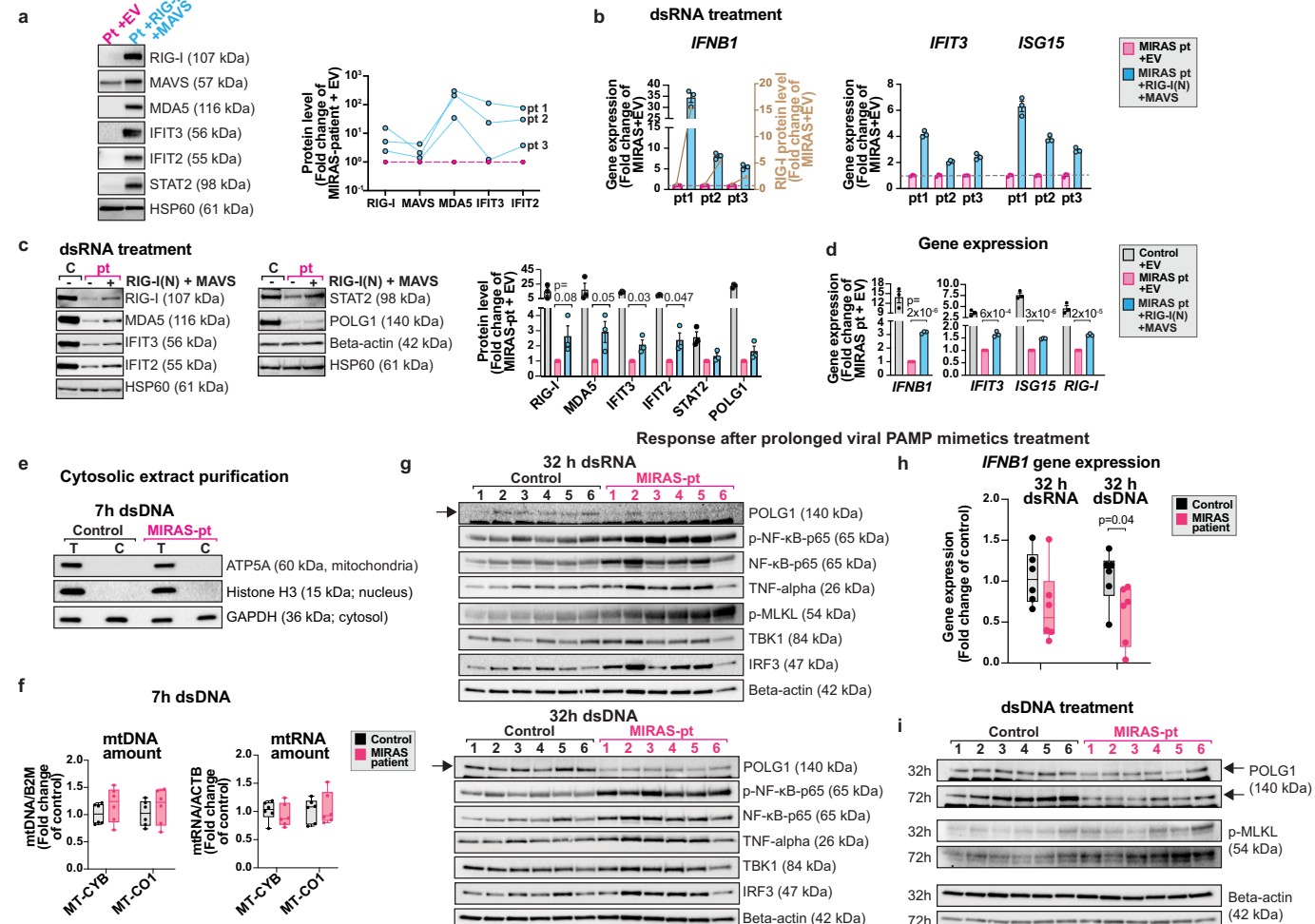

**Extended Data Fig. 3 | Immune signalling in MIRAS patient fibroblasts in response to viral PAMP mimetics.** (a) Induction of IFN-I signalling by exogenous RIG-I and MAVS expression. MIRAS patient fibroblasts transfected with empty vector (EV) or RIG-I (N; constitutive active) and MAVS vectors. Western blot and quantification, loading control HSP60; N = 3 female patients; protein level of each MIRAS patient fibroblast line quantified. (b) IFN-I signalling in response to dsRNA/poly(I:C) treatment upon exogenous expression of RIG-I and MAVS. MIRAS fibroblasts as described in **a**, further treated with 7 h of dsRNA/poly(I:C). qPCR of cDNA, reference gene *ACTB*; RIG-I protein level detected in each of three patient fibroblast lines before dsRNA/poly(I:C) treatment (as quantified in **a** is overlaid onto the chart, showing positive association between RIG-I protein amount and the subsequent dsRNA/poly(I:C)-induced expression of *IFN-β*. N = 3 female patients; 3 technical replicates per patient; mean ± SEM. (c,d) RIG-I and MAVS exogenous expressions improve IFN-I signalling in MIRAS fibroblasts in response to viral PAMP mimetics. **c:** Western blot representative, quantification, loading control HSP60; **d:** qPCR of cDNA, reference gene *ACTB*, in control and MIRAS fibroblasts with or without exogenous expression of RIG-I(N) and MAVS and subjected to 7 h dsRNA/poly(I:C) treatment. N = 3 female controls and patients; mean ± SEM; two-tailed unpaired student's t test. (e) Cytosolic extract

purification from fibroblasts. Western blotting for protein markers of mitochondria, nucleus and cytosol. T, total (whole-cells); C, cytosol. Representative of 3 independent experiments. (f) mtDNA and mtRNA amount in fibroblasts post 7 h dsDNA treatment. qPCR of DNA or cDNA (*MT-CYB* and *MT-CO1* relative to reference gene *B2M* (mtDNA) *or ACTB* (mtRNA)). N = 6 female controls and patients per condition. (g) Immune signalling and necroptosis activation in fibroblasts after prolonged viral PAMP mimetics exposure. Western blot at 32 h of dsRNA/poly(I:C) or dsDNA treatment, loading control beta-actin (Quantification in Fig. 1h, i). N = 6 female controls and patients per condition. Arrowhead: protein band of interest. (h) IFN-β (*IFNB1*) expression in fibroblasts after prolonged viral PAMP mimetics exposure. qPCR of cDNA, reference gene *ACTB*, at 32 h of treatment. N = 6 female controls and patients per condition. (i) Necroptosis activation in fibroblasts after prolonged (32 h or 72 h) dsDNA treatment. Western blot, loading control beta-actin (Quantification in Fig. 1i). N = 6 female controls and patients per condition. Arrowhead: protein band of interest. In **f** and **h**, box plots show minimum-25th-50th(median)-75th percentile-maximum (whiskers extend to the smallest and largest value); two-tailed unpaired student's t-test. *Abbreviations: C, control; Pt, patient; IFN, interferon; PAMP, pathogen-associated molecular pattern*.

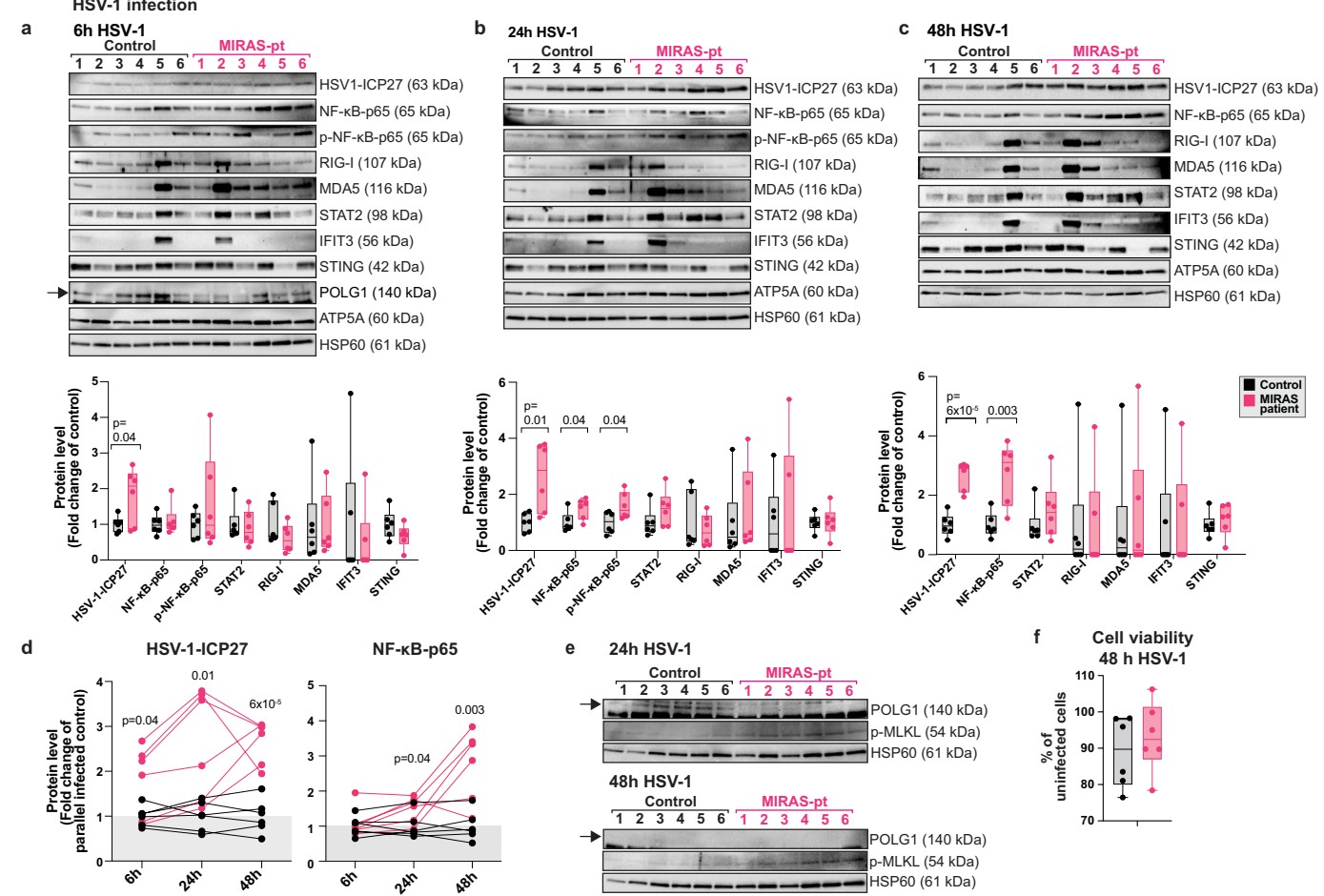

**Extended Data Fig. 4 | HSV-1 infection of MIRAS patient and control fibroblasts.** (a-c) Immune signalling pathway protein and viral protein amount in fibroblasts during HSV-1 infection. Western blot (top) and quantification (bottom) at **a:** 6 h, **b:** 24 h, and **c:** 48 h of HSV-1 infection, loading control HSP60; arrowhead: band of interest. N = 6 female controls and patients per condition. Quantification also in Fig. 2d. (d) Comparative fold expression of HSV-1 ICP-27 and host cellular pro-inflammatory protein, NF-κB-p65 in the parallelly infected MIRAS patient and control fibroblasts at the indicated infection time from panels **a-c** above; two-tailed unpaired student's t test. (e) POLG1 protein

level and necroptosis sensitivity of fibroblasts during TBEV infection. Western blot of POLG1 and necroptotic activating p-MLKL protein, loading control HSP60; arrowhead: band of interest. N = 6 female controls and patients per condition. Quantification in Fig. 2d. (f) Fibroblast cell viability at 48 h of HSV-1 infection. N = 6 female controls and patients per condition; In **a**, **b**, **c** and **f**, box plots show minimum-25th-50th(median)-75th percentile-maximum (whiskers extend to the smallest and largest value); two-tailed unpaired student's t-test. *Abbreviations: Pt, patient.*

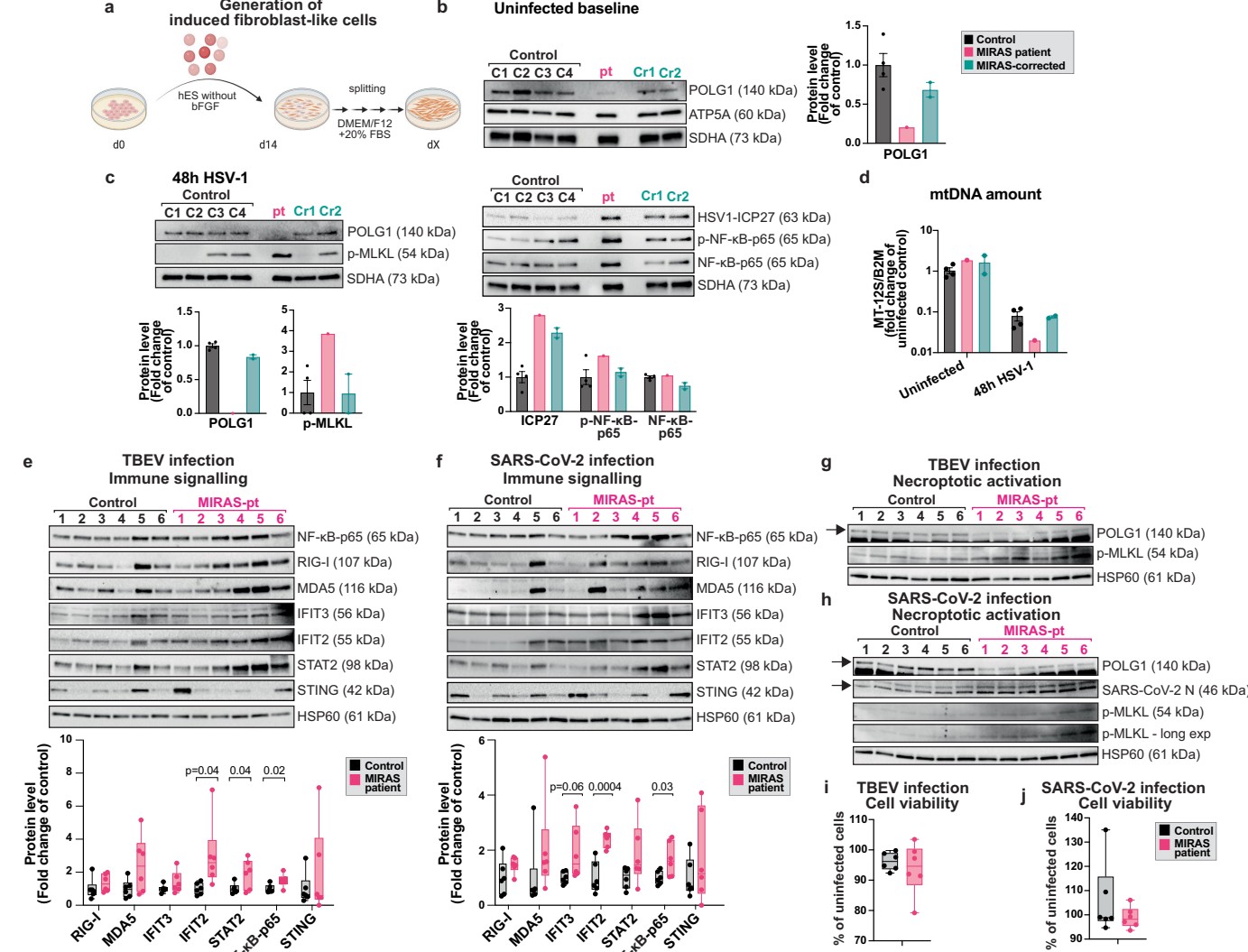

**Extended Data Fig. 5 | Response of MIRAS mutation-corrected fibroblast-like cells to HSV-1 infection, and control and patient fibroblasts response to TBEV or SARS-CoV-2 infection.** (a) Schematic for generation of induced fibroblast-like cells from iPSC cell lines. Created using BioRender.com. See method section for details. (b) POLG1 protein amount in controls, MIRAS patient and MIRAS mutation-corrected induced fibroblast-like cells. Western blot and quantification, loading control SDHA. Mean ± SEM. N = 4 controls (2 males, 2 females), 1 male patient, 2 independent MIRAS mutation-corrected clones of the male patient (Cr1, Cr2). (c) Viral protein, host inflammatory and necroptotic protein in cells (as in **b**) at 48 h post HSV-1 infection. Western blot and quantification, loading control SDHA. Mean ± SEM. (d) MtDNA amount at uninfected basal condition and 48 h post HSV-1 infection in cells (as in **b**). mtDNA, *MT-12S* to nuclear gene, *B2M*. Mean ± SEM. (e-h) Immune signalling (**e**,**f**) and necroptotic activation (**g**,**h**) of fibroblasts post 48 h of TBEV (**e**,**g**) or SARS-CoV-2 (**f**,**h**) infection. Western blot and quantification, loading control HSP60, arrowhead: band of interest. N = 6 female controls and patients per condition. Refer to Fig. 2g, h for other protein quantification. (i,j) Cellular viability of fibroblasts post 48 h of TBEV (**i**) and SARS-CoV-2 (**j**) infection. N = 6 female controls and patients per condition. In **e**, **f**, **i** and **j**, box plots show minimum-25th-50th(median)-75th percentile-maximum (whiskers extend to the smallest and largest value); two-tailed unpaired student's t-test. *Abbreviations: Pt, patient*.

**Extended Data Fig. 6 | Generation and characterization of MIRAS knock-in mice.** (a) Generation of MIRAS knock-in mice. (b) DNA sequencing verification of wildtype (control), heterozygote and homozygote MIRAS mice. DNA sequence alignment indicates MIRAS variants c. 2177G>C in exon 13 and c. 3362A>G in exon 21 of *Polg1*. (c) Genotype verification of wildtype, heterozygotic or homozygotic MIRAS mice. Amplification of exon 13 and lox^P site containing intron 13, shows a 120 bp longer fragment in MIRAS than wildtype control. The genotyping procedure was performed on all mice analysed in the study. (d) Weight of mice at 2, 6 and 12 months of age. Age 2 months: N = 9 control, 10 MIRAS; age 6 months: N = 10 control, 10 MIRAS; age 12 months: 10 control, 11 MIRAS; all females; mean ± SEM.

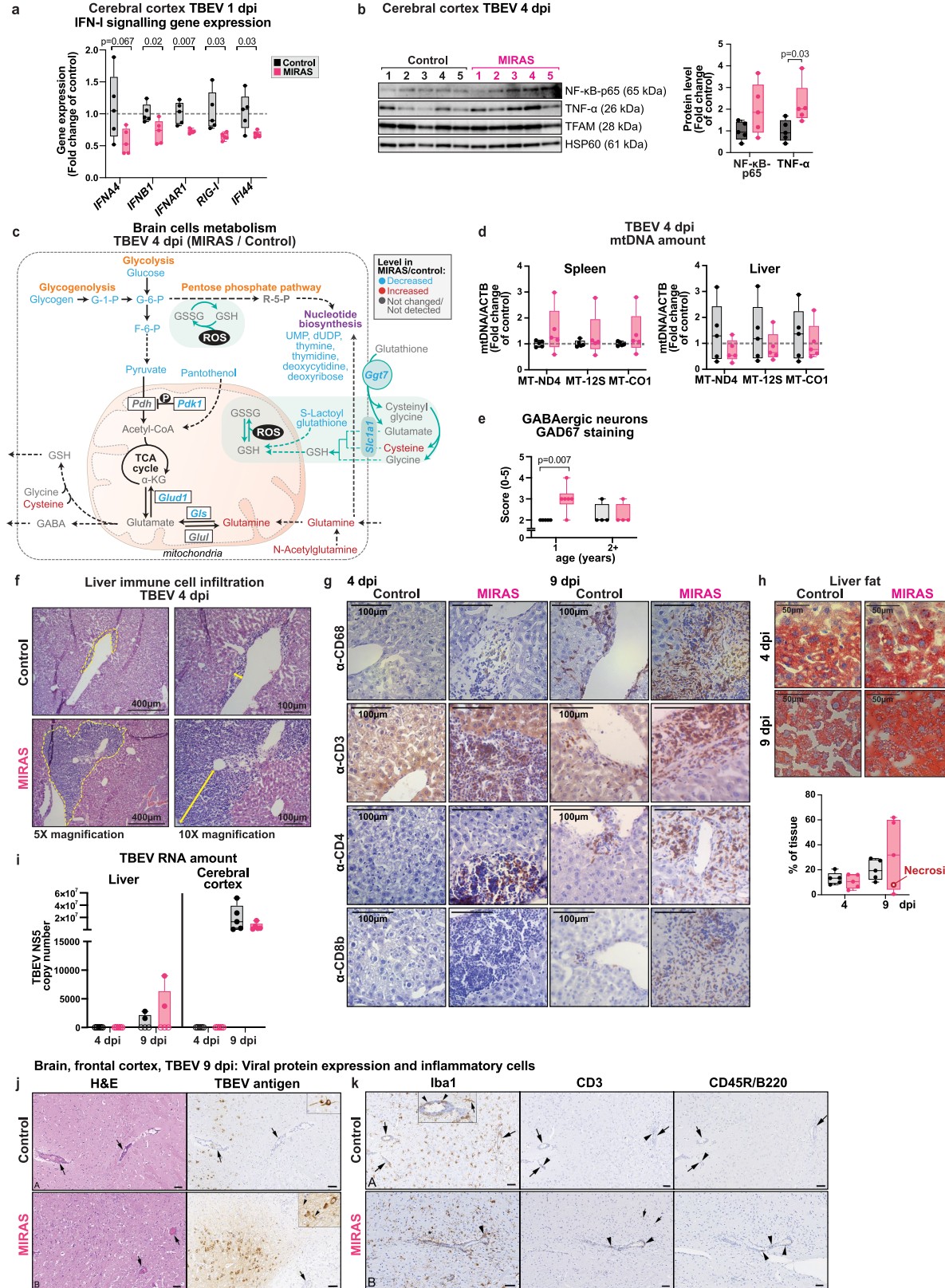

**Extended Data Fig. 7** | See next page for caption.

**Extended Data Fig. 7 | Impact of TBEV infection on mouse tissues.**
(a) IFN-I signalling, mouse cerebral cortex, TBEV 1 dpi. qPCR of cDNA, reference gene *ACTB*. (b) Inflammatory proteins in mouse cerebral cortex. TBEV 4 dpi. Western blot, quantification, loading control HSP60. (c) Schematic depicting brain cell metabolism, TBEV 4 dpi (MIRAS/Control). Alteration of metabolites (as in Fig. 4b, level indicated for metabolites of q-value ≤ 0.05) and transcripts encoding for pathway proteins (level indicated for transcripts of p-value < 0.05; RNA seq; Wald test). *Pdh*, pyruvate dehydrogenase; *Pdk1*, pyruvate dehydrogenase kinase 1; *Glud1*, glutamate dehydrogenase 1; *Gls*, glutaminase; *Glul*, glutamate-ammonia ligase (glutamine synthetase); *Ggt7*, gamma-glutamyl transpeptidase 7; *Slc1a1*, solute carrier family 1 member 1; GSSG/GSH, glutathione/glutathione-reduced; ROS, reactive oxygen species. (d) mtDNA amount; mouse spleen and liver; day 4 post-TBEV infection. qPCR of mtDNA (*MT-ND4, MT-12S* and *MT-CO1*) relative to nuclear gene (*ACTB*). (e) GABAergic neurons (GAD67 in uninfected mouse cerebral cortex at baseline. Semiquantitative scoring of histological staining. (f) Immune cell infiltration of mouse liver; TBEV 4 dpi. Representative Hematoxylin-eosin (H&E) staining. 5X; yellow dotted line: boundary of immune cell infiltrate. Right panel: 10X; yellow double-headed arrow: distance spanned by the immune cell infiltration from the portal tract. Refer to Fig. 5b for quantification. (g) Representative CD68+ (macrophages), CD3 + /4 + /8b+ (pan-T/helper-T/cytotoxic-T lymphocytes) staining in liver, TBEV 4 and 9 dpi. Quantification in Fig. 5d. (h) Liver fat content 4 and 9 dpi. Oil-red O staining. Red: lipid (on hematoxylin). Quantification: lipid as % of tissue. (i) TBEV RNA amount in mouse liver and cerebral cortex, 4 and 9 dpi. qPCR of cDNA (*TBEV-NS5*, Non-structural 5).

Samples with no detectable *TBEV-NS5* RNA: open circle at zero value. (j, k) Inflammatory response and viral protein expression in the mouse brain (frontal cortex; TBEV 9 dpi. H&E staining / immunohistochemistry, hematoxylin counterstaining. Bars = 50 µm. (j) Control mouse. A mild focal perivascular mononuclear infiltration in association with viral antigen expression in moderate numbers of neurons in the adjacent parenchyma (arrow). Inset: infected neurons with strong TBEV antigen expression in cell body and processes. MIRAS: mild focal perivascular mononuclear infiltration (arrows) and abundant infected neurons in the adjacent parenchyma. Inset: extensive viral antigen expression in neuronal cell bodies and cell processes (arrowheads). (k) Inflammatory cells, microglial response. Control; Perivascular infiltrates (arrows) contain several macrophages (Iba1 + ; inset: arrowheads), a few T cells (CD3 + ; arrowheads) and rare B cells (CD45R/B220 + ; arrowhead). Iba1 staining: disseminated activated microglial cells with their typical stellate shape (inset: arrow). MIRAS; perivascular infiltrates containing macrophages (Iba1 + ; arrowhead), T cells (CD3 + ; arrowheads) and several B cells CD45R/ B220 + ; arrowheads). A few individual T cells (small arrows) in adjacent parenchyma. Iba1: mild focal microgliosis of the parenchyma adjacent to the affected vessel (asterisk). In **a-d** and **f-k**, N = 5 mice per condition; age 12 months, females; In **e**, 1 year-old mice (N = 5 control, 6 MIRAS, males) and 2+ year-old mice (N = 4 control, 4 MIRAS, females). In **a**, **b**, **d**, **e**, **h** and **i**, box plots show minimum-25th-50th(median)-75th percentile-maximum (whiskers extend to the smallest and largest value); two-tailed unpaired student's t-test. *Abbreviations: Dpi, day post infection*.

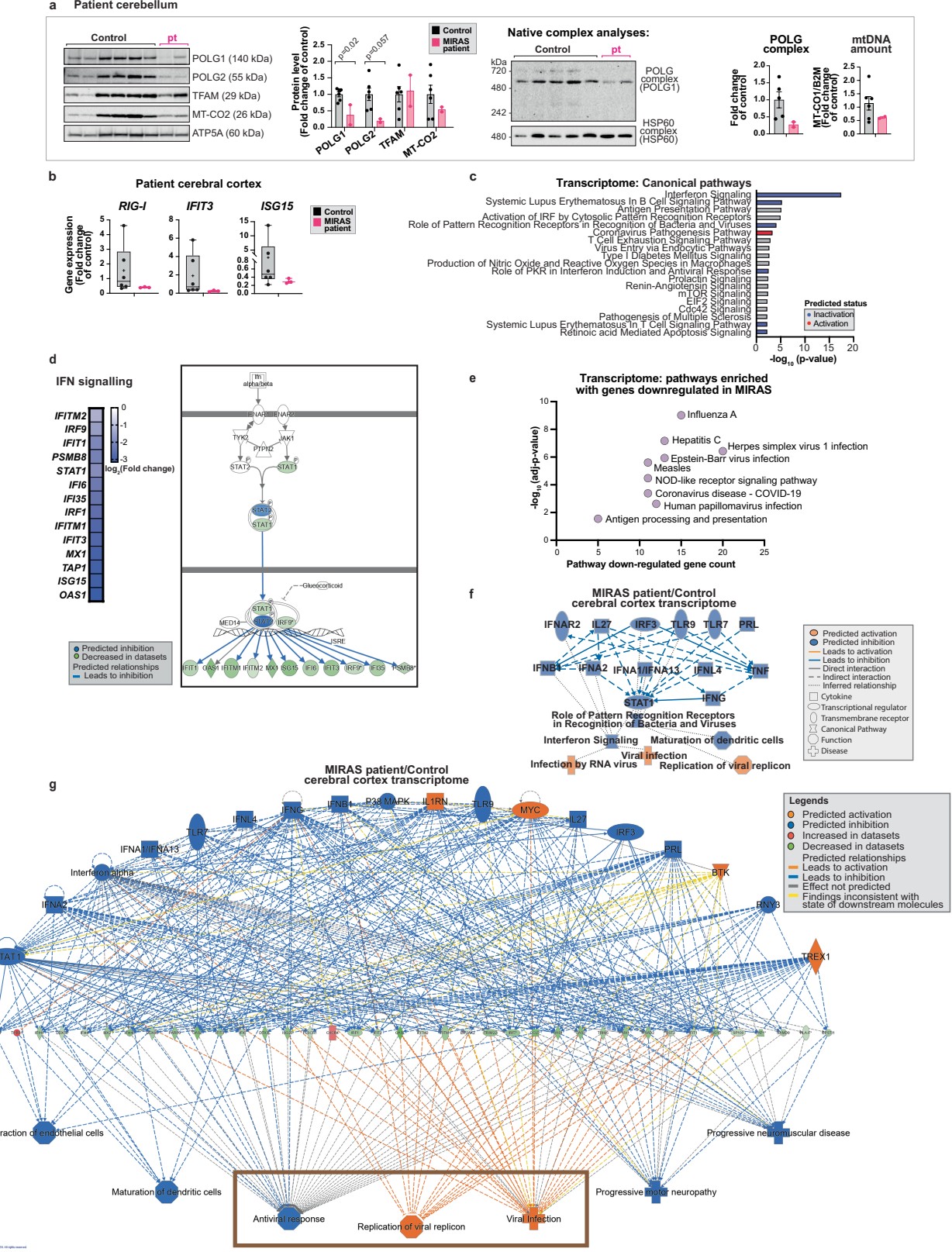

**Extended Data Fig. 8** | See next page for caption.

**Extended Data Fig. 8 | Analyses of MIRAS patient brain autopsy samples.**
(a) mtDNA replisome and mtDNA amount in patient cerebellum. Western blot and quantification (loading control ATP5A) and native POLG replisome complex amount (loading control HSP60 complex) from isolated mitochondria; qPCR for mtDNA amount (mtDNA, *MTCO1* relative to nuclear gene, *B2M*). N = 2 patient cerebellar samples and 6 (or 5 for native complex analysis) controls, all females. Mean ± SEM; two-tailed unpaired student's t test. (b) Immune-related gene expression in MIRAS patients' cerebral cortex. qPCR of cDNA, reference gene *ACTB*. N = 3 patients and 6 controls, all females; box plot: minimum-25th-50th(median)-75th percentile-maximum (whiskers extend to the smallest and largest value); + symbol indicates the mean expression value; two-tailed unpaired student's t test. (c-g) Pathway enrichment analyses of MIRAS patient cerebral cortex transcriptome. RNA-seq; N = 3 patients compared to 5 controls, all females. (c) Ingenuity pathway analysis using genes with p-value < 0.05 in MIRAS patient/control transcriptome. The most significantly affected (based on p-value; Fisher's exact test) canonical pathways with their predicted activation or inactivation status are tabulated. Red: predicted activation (z-score≥2); blue: predicted inactivation (z-score ≤ −2); grey: pathway with no predicted significant activation/inhibition. (d) Inhibition of Interferon (IFN) signalling. Heatmap: fold change of IFN signalling gene expressions in MIRAS patients compared to the controls, and gene mapping onto the IFN-signalling pathway (generated by Ingenuity Pathway Analysis). (e) KEGG pathway enrichment analysis of downregulated genes (p-value < 0.05; Fold change > −1.5) in MIRAS patients compared to controls; generated using gProfiler server; Fisher's one-tailed test with multiple testing correction using the server default algorithm g:SCS. (f, g) Integrated network summary of canonical pathways, upstream regulators, diseases and biological functions associated with patient/control cerebral cortex transcriptome (genes with p-value < 0.05). Generated using Ingenuity Pathway Analysis knowledgebase. The network in **g** shows also predicted relationships to transcripts with levels altered in the dataset.

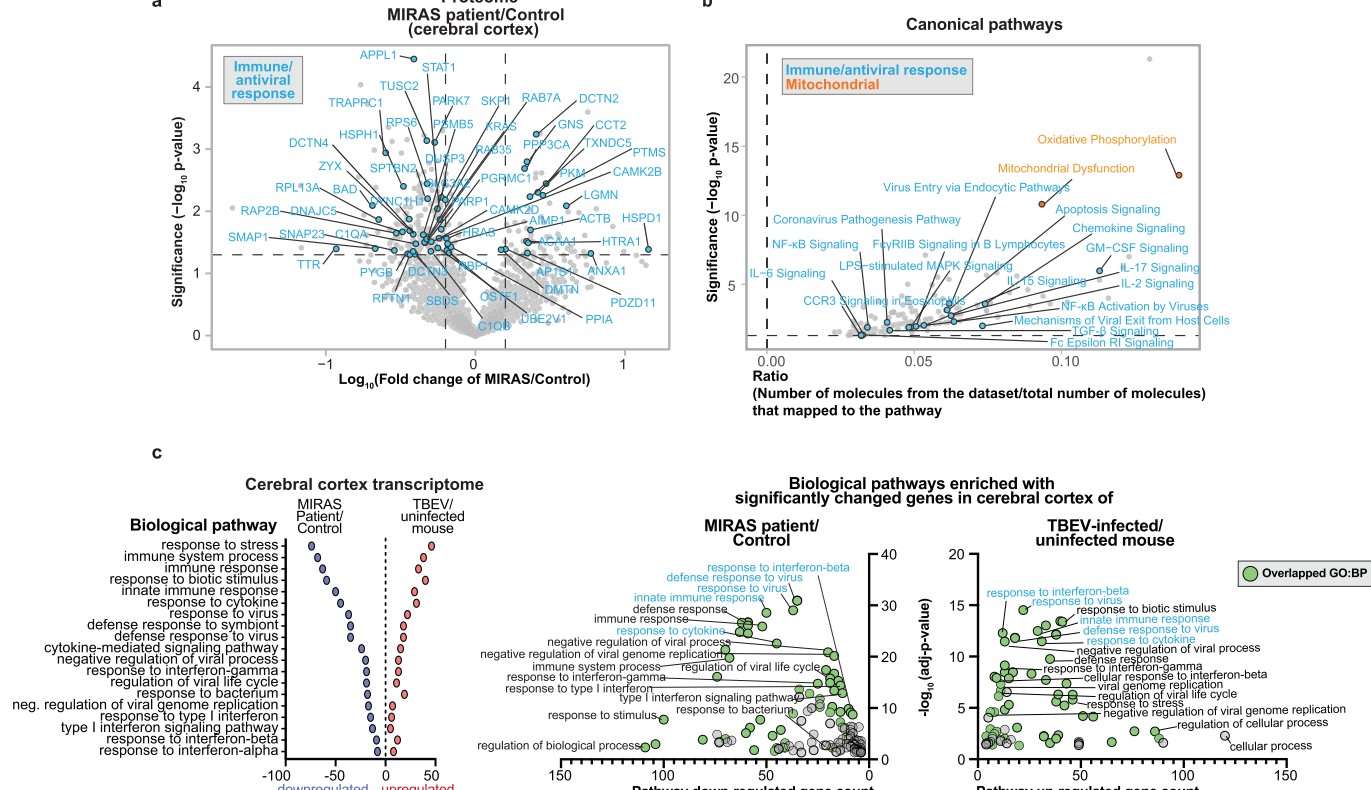

**Extended Data Fig. 9 | MIRAS patient cerebral cortex proteome and transcriptome.** (a) Proteome profiling of MIRAS patient cerebral cortex. Volcano plot: significance (p-value) and fold change. 57 of the proteins with p-value < 0.05 (two-sample Student's T-test) are associated with immune/antiviral pathways (highlighted in blue). N = 3 patients and 6 controls, all females. (b) Canonical pathway enrichment analysis of cerebral cortex proteome in MIRAS (as in **a**). Ingenuity pathways analyses on proteins significantly affected (p-value < 0.05) in MIRAS; pathway p-value (Fisher's exact test) and ratio (number of molecules from the dataset that map to the pathway divided by the total number of molecules that map to the canonical pathway). Blue: pathways related to immune and/or viral process; orange: pathways related to mitochondrial functions. (c) Comparative pathway profiling of cerebral cortex

transcriptome of patient/control (as in Fig. 5h) versus TBEV-infected/uninfected wildtype mice (as in Fig. 3f). Biological pathways (analysed using gProfiler server; Fisher's one-tailed test with multiple testing correction using the server default algorithm g:SCS) enriched with downregulated genes (p-value < 0.05, fold change > −1.5) in patient/control or with upregulated genes (adj-p-value < 0.05, fold change>1.5) in TBEV 4 dpi/uninfected wildtype control mice. Left: Top common overlapping pathways of the two datasets (pathway adj-p-value < 0.05) are tabulated with the pathway-associated gene count. Right: the pathway-associated gene count and adj-p-value. 58 overlapping pathways between the two datasets are highlighted in green shade. Selected top pathways related to virus and immune response are annotated in blue text to depict the opposed regulation.

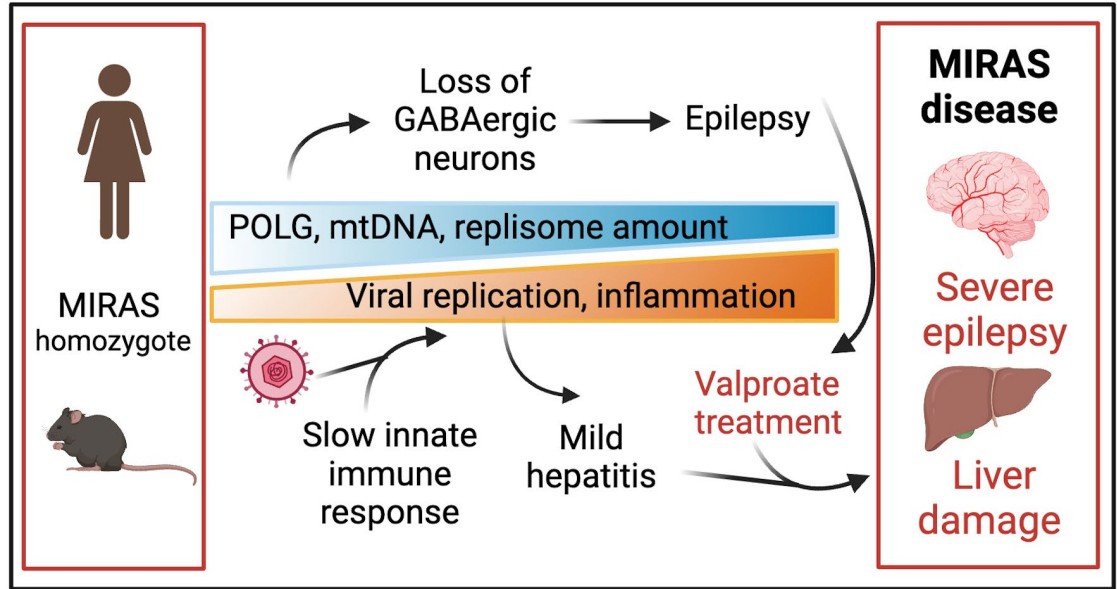

**Extended Data Fig. 10 | Overview of research data.** Created using BioRender.com.

# Reporting Summary

## Statistics

For all statistical analyses, confirm that the following items are present in the figure legend, table legend, main text, or Methods section.

| n/a | Confirmed | |
|---|---|---|
| ☐ | ☒ | The exact sample size (*n*) for each experimental group/condition, given as a discrete number and unit of measurement |
| ☐ | ☒ | A statement on whether measurements were taken from distinct samples or whether the same sample was measured repeatedly |
| ☐ | ☒ | The statistical test(s) used AND whether they are one- or two-sided *Only common tests should be described solely by name; describe more complex techniques in the Methods section.* |
| ☐ | ☒ | A description of all covariates tested |
| ☐ | ☒ | A description of any assumptions or corrections, such as tests of normality and adjustment for multiple comparisons |
| ☐ | ☒ | A full description of the statistical parameters including central tendency (e.g. means) or other basic estimates (e.g. regression coefficient) AND variation (e.g. standard deviation) or associated estimates of uncertainty (e.g. confidence intervals) |
| ☐ | ☒ | For null hypothesis testing, the test statistic (e.g. *F*, *t*, *r*) with confidence intervals, effect sizes, degrees of freedom and *P* value noted *Give P values as exact values whenever suitable.* |
| ☒ | ☐ | For Bayesian analysis, information on the choice of priors and Markov chain Monte Carlo settings |
| ☒ | ☐ | For hierarchical and complex designs, identification of the appropriate level for tests and full reporting of outcomes |
| ☒ | ☐ | Estimates of effect sizes (e.g. Cohen's *d*, Pearson's *r*), indicating how they were calculated |

*Our web collection on statistics for biologists contains articles on many of the points above.*

## Software and code

Policy information about availability of computer code

| | |
|---|---|
| Data collection | Fluorescence images were acquired using Zeiss AxioImager epifluorescent microscope<br>Histological images were collected using a light microscope equipped with a ProgRes® CCD Routine Camera (C7, JENOPTIK Optical Systems GmbH) with ProgRes® Capture Pro Camera Control Software 2.10.0.1.<br>Untargeted metabolite profiling was performed using flow injection analysis on an Agilent 6550 QTOF instrument (Agilent, Santa Clara, CA).<br>Proteomics data was collected by LC-MS/MS using an Easy nLC1000 liquid chromatography system (Thermo Electron, Bremen, Germany) coupled to a QExactive HF Hybrid Quadrupole-Orbitrap mass spectrometer (Thermo Electron) with a nanoelectrospray ion source (EasySpray, Thermo Electron).<br>For RNA sequencing analysis, the quality of the sequencing libraries was assessed using the TapeStation DNA High Sensitivity Assay (Agilent). The libraries were sequenced on a Illumina NextSeq 500.<br>Immunoblot and southern blot images were obtained using ChemiDoc™ XRS+ imaging machine (BioRad) and signals were quantified using Image Lab Version 6.1.0 build 7 software (BioRad).<br>Radioactive signals were developed using Typhoon™FLA7000 (GE Healthcare).<br>qPCR were performed using CFX96 or 384 Touch Real-Time PCR Detection System (BioRad).<br>ELISA assay was measured using Enspire Multimode plate reader (Perkin Elmer; Software version 4.13.3005.1482). |
| Data analysis | For proteomic analyses, the Uniprot human database (September 2018) was used was used for the database searches. Protein identification and label-free quantification was done using MaxQuant (ver 1.6.7.0) software followed by data filtering and statistical analysis with Perseus (ver 1.6.1.3) software.<br>For RNA seq, Read alignments were done using GRCm38 (mouse) reference genome and GENCODE Mouse Release 28 or GRCh38 (human) reference genome and GENCODE Human Release 33 comprehensive gene annotation files. Differential expression analysis was done in the DESeq2 software (as per default setting) in R environment. |

For metabolomic data analysis, HMDB v4.0 and 3.0 database were used.

For functional/pathway enrichment analyses, Qiagen Ingenuity Pathway Analyses (QIAGEN Inc., https://digitalinsights.qiagen.com/IPA), g:Profiler (https://biit.cs.ut.ee/gprofiler) toolset and KEGG database were utilized for the analyses of transcriptome, metabolome and/or proteome datasets. For immune pathway analyses, we further utilized the manually curated InnateDB database (https://www.innatedb.com/index.jsp). The server default test/algorithm was utilized to calculate the p value also described in the figure legend.

Statistical analyses were performed using Microsoft Excel version 16.80, GraphPad Prism version 10.1.1 for macOS (GraphPad Software, www.graphpad.com) or using the server default test as described above.

Quantification of immunofluorescence signal was performed using CellProfiler 4.2.6 software and for immunohistology, liver immune infiltration area was measured using ImageJ 2.0.0-rc-69/1.52n software (https://imagej.net/ij/) whereas liver ORO and CDs signal were quantified using CellProfiler 4.2.6 software after pixel classification by ilastik 1.3.3 software.

Data compiling and processing was performed using Microsoft Excel version 16.80 and Adobe Illustrator 26.2.1.

Images were created using Adobe Illustrator 26.2.1. or with BioRender.com

For manuscripts utilizing custom algorithms or software that are central to the research but not yet described in published literature, software must be made available to editors and reviewers. We strongly encourage code deposition in a community repository (e.g. GitHub). See the Nature Portfolio guidelines for submitting code & software for further information.

# Data

Policy information about availability of data

All manuscripts must include a data availability statement. This statement should provide the following information, where applicable:
- Accession codes, unique identifiers, or web links for publicly available datasets
- A description of any restrictions on data availability
- For clinical datasets or third party data, please ensure that the statement adheres to our policy

The mouse RNA-seq data generated in this study have been deposited into NCBI's Gene Expression Omnibus (GEO) and are accessible through GEO series accession number GSE249432. Metabolomic data have been deposited to the MassIVE database and are accessible through MSV000093634. Human omics data sharing is restricted because of European general data protection regulations (GDPR) laws. Individual enquiry of expression changes of specific genes /proteins can be directed to the corresponding author. Numerical source data giving rise to graphical representation and statistical description in Figures 1-6 and Extended Data Figures 1-9 are provided as Source Data file and in Supplementary Tables 2-5. Characteristics of human research participants and materials used in this study are provided in Supplementary Tables 1 and 6 respectively. Uncropped images of southern/immunoblots presented in the figures are included in Supplementary Fig 1.

# Research involving human participants, their data, or biological material

Policy information about studies with human participants or human data. See also policy information about sex, gender (identity/presentation), and sexual orientation and race, ethnicity and racism.

| Reporting on sex and gender | We selected gender matched patient and controls in the fibroblast studies to control for potential immune variation due to gender induced effect. Similarly, for autopsy sample analyses, we used material derived from individuals of the same gender. The sex/gender of human participants isolated materials are described in the respective figure legends and in Supplementary Table 1. |
|---|---|
| Reporting on race, ethnicity, or other socially relevant groupings | The human participants are from Finnish population which is clinically relevant for the study of the MIRAS disease in this report which has high prevalence within Finnish population. No race, ethnicity or socially relevant groupings were applied. |
| Population characteristics | For population, see above. The human tissues samples were from autopsies, where the families had consented the materials for research use. The skin biopsies and blood samples were collected with consent from patients and control individuals. The characteristics of the human research participants including age, gender, genotype and diagnosis, are provided in Supplementary Table 1. |
| Recruitment | The blood and fibroblast samples were from patients who were previously diagnosed to carry the POLG1 mutation of interest and recruited and informed by their physician, who also collected the consents. |
| Ethics oversight | Human samples were collected and used with informed consents, according to the Helsinki Declaration and approved by the Ethical Review Board of Kuopio University Central Hospital (license number 410/2019). Patient and control materials included fibroblasts (established from skin biopsies from individuals' forearms), blood and autopsy-derived brain samples. Control samples were from voluntary healthy subjects (fibroblasts and sera) and for brains, from people who died acutely for non-CNS-disease causes. Autopsy sample collection was approved by the governmental office for social topics and health. |

Note that full information on the approval of the study protocol must also be provided in the manuscript.

# Field-specific reporting

Please select the one below that is the best fit for your research. If you are not sure, read the appropriate sections before making your selection.

☒ Life sciences ☐ Behavioural & social sciences ☐ Ecological, evolutionary & environmental sciences

For a reference copy of the document with all sections, see nature.com/documents/nr-reporting-summary-flat.pdf

# Life sciences study design

All studies must disclose on these points even when the disclosure is negative.

| | |
|---|---|
| Sample size | No sample size calculation were performed. For mice, sample sizes were chosen to ensure adequate power and to account for potential inter-individual/animal, gender and age variance (age and gender-matched samples were utilized as controls). The number of biologically independent mouse or different individual derived human samples was described in the respective figure legends. The limitation here was the virus exposure capacity in safety-level-3 facilities. For mouse analyses, we have at least 4 or more mice of the same sex per condition. For human samples (primary fibroblasts, autopsy-derived brain samples) the availability of patients limits the sample size. The experiments were performed on six biologically independent patient-derived primary fibroblast lines compared to six age and gender matched control individual fibroblast lines (or otherwise as indicated in the figure legends). To further confirm the causal role of MIRAS p.W748S, we CRISPR-corrected the mutation in an iPSC cell line of a biological patient and generated two independent clones of iPSC-induced fibroblast from the corrected line, and compared their response to four different control individual cell lines. For brain samples we have n=3 patient cerebral cortex (n=2 cerebellum) and n=15 patient blood samples and their age/gender-matched controls (n=5/6 for brain samples or 23 for blood samples, respectively). For mouse analyses, we have at least 4 or more mice of the same sex per condition. The genetic association analyses were generated from the genome and health data collected by the FinnGen study from 309,154 individuals from Finnish population. |
| Data exclusions | No human or mouse data point were excluded. Specific filtering criteria are included in our omic analyses as described in the method section. |
| Replication | Biologically replicates or independent human-derived samples (of matched gender and age) are included to ensure reproducibility: 6 independent human derived primary fibroblast lines that were isolated from 6 different control individuals or patients (or as indicated in the figure legends) and 4 or more mice were analyzed (exact number of mice analyzed per analyses are indicated in the figure legends); for human tissue samples - the number is limited by the availability of the material and the exact number of different individual derived samples is described in the respective figure legends. The cell experiments were replicated at least once using independent individual derived line(s). For virus infection, the experiment was tested with a few biological cell lines and then replicated for the full set of N=6 human cell lines (or as indicated in the respective figure legends). Where representative data are shown, the experimental findings were replicated in independent biological cell lines/mice (number as indicated in the respective figure legends) and quantification of the each biological replicates are included in the data representation. |
| Randomization | This is not relevant to our study with human participants as no conduct of research to obtain data or samples through intervention or interaction with individuals.<br>No specific method of randomization was used to select animals or human cell lines. Human cell lines and mice of targeted genotype were screened and selected to investigate the effect of the disease gene mutation in the phenotype and response to viral infection. |
| Blinding | This is not relevant to our study with human participants as no conduct of research to obtain data or samples through intervention or interaction with individuals.<br>No blinding was done with the mouse group generation as this was predetermined by the mouse genotype.<br>Histological evaluation of mouse GABAergic defect evaluation was performed blindly.<br>Mouse liver histological evaluation was not performed blindly but was independently scored by 2 researchers. |

# Reporting for specific materials, systems and methods

We require information from authors about some types of materials, experimental systems and methods used in many studies. Here, indicate whether each material, system or method listed is relevant to your study. If you are not sure if a list item applies to your research, read the appropriate section before selecting a response.

## Materials & experimental systems

| n/a | Involved in the study |
|---|---|
| ☐ | ☒ Antibodies |
| ☐ | ☒ Eukaryotic cell lines |
| ☒ | ☐ Palaeontology and archaeology |
| ☐ | ☒ Animals and other organisms |
| ☒ | ☐ Clinical data |
| ☒ | ☐ Dual use research of concern |
| ☒ | ☐ Plants |

## Methods

| n/a | Involved in the study |
|---|---|
| ☒ | ☐ ChIP-seq |
| ☒ | ☐ Flow cytometry |
| ☒ | ☐ MRI-based neuroimaging |

## Antibodies

| | |
|---|---|
| Antibodies used | All primary antibodies used were described below and in Extended data table 6,<br>Antibodies; Source; Catalogue number (dilution factor: refer to the Extended Data Table 6)<br>Antibodies Source Catalogue number<br>Anti-TOM20 Santa Cruz Biotechnology sc-11415<br>Anti-HSP60 Santa Cruz Biotechnology sc-1052<br>Anti-MDA5 Proteintech 21775-1-AP<br>Anti-STING Cell Signaling Technology 50494S<br>Anti-STAT2 Proteintech 16674-1-AP |

Anti-RIG-I Cell Signaling Technology 3743S
Anti-TFAM Abcam Ab131607
Anti-POLG1 Abcam ab128899
Anti-POLG2 Proteintech 10997-2-AP
Anti-MT-CO2 Abcam ab110258
Anti-COX4 Abcam ab14744
Anti-IFIT3 Proteintech 15201-1-AP
Anti-IFIT2 Proteintech 12604-1-AP
Anti-Beta-Actin Cell Signalling Technology 3700S
Anti-Phospho-MLKL-T357/S358/S360 ABClonal AP0949
Anti-MLKL ABClonal A13451
Anti-ATP5A Abcam ab14748
Anti-Phospho-NF-kappaB p65 (Ser536) [93H1] Cell Signaling Technology 3033S
Anti-NF-kappaB p65 Santa Cruz Biotechnology sc-8008
Anti-TNF-alpha Abcam ab66579
Anti-IRF3 (D83B9)  Cell Signaling Technology 4302S
Anti-TBK1/NAK (D1B4) Cell Signaling Technology 3504
Anti-CD68 [FA-11] Bio-Rad MCA1957
Anti-CD3 - mouse liver BD Biosciences 555273
Anti-CD4  BD Biosciences 550280
Anti-CD8b BD Biosciences 550797
Anti-CD45R [B220/RA3-6B2] BD Pharmingen MLDP7
Anti-CD3 [SP7] - mouse brain Spring Bioscience Corp., Ventana Medical Systems; Abcam ab16669
Anti-Iba1 WAKO 019-19741
Anti-dsRNA Scicons 10010200
Anti-BrdU BD Pharmingen 555627
Anti-GAD67 [clone 1G10.2] Merck Millipore MAB5406
Anti-GABRB2 Abcam ab42598
Anti-GAPDH Cell Signaling Technology 2118S
Anti-Histone H3 Cell Signaling Technology 3638
Anti-HSV-1-ICP27 Santa Cruz Biotechnology sc-69806
Anti-SARS-CoV2-N Provided by Prof. Olli Vapalahti Department of Virology, Faculty of Medicine, University of Helsinki, Helsinki, Finland
Anti-TBEV  Provided by Prof. Olli Vapalahti Department of Virology, Faculty of Medicine, University of Helsinki, Helsinki, Finland
Anti-TBEV (Hochosterwitz isolate)  Donated by Prof. Karin Stiasny  Center for Virology, Medical University of Vienna, Austria
Peroxidase AffiniPure™ Goat Anti-Mouse IgG (H+L)  Jackson ImmunoResearch  115-035-146
Peroxidase AffiniPure™ Goat Anti-Rabbit IgG (H+L)  Jackson ImmunoResearch  111-035-144
Donkey Anti-Goat IgG H&L (HRP) Abcam ab97110
ImmPRESS® HRP Goat Anti-Rat IgG, Mouse adsorbed Polymer Detection Kit, Peroxidase Vector Laboratories MP-7444-15
VECTASTAIN® Elite® ABC-HRP Kit, Peroxidase (Mouse IgG) Vector Laboratories PK-6102
VECTASTAIN® Elite® ABC-HRP Kit, Peroxidase (Rabbit IgG) Vector Laboratories PK-6101
EnVision+ anti-HRP, Rabbit Agilent Dako K4003
OmniMap anti-Rabbit HRP  Ventana Medical Systems, Inc. 760-4311
Goat anti-Mouse IgG (H+L) Alexa fluor 594 Invitrogen A-11005
Goat anti-human IgG (H+L) Alexa fluor 488 Invitrogen A-11013

| | |
|---|---|
| Validation | Antibodies were used according to the validation listed in manufacturer's instructions. The details of antibody validation are given in Supplementary Table 6. |

# Eukaryotic cell lines

Policy information about cell lines and Sex and Gender in Research

| | |
|---|---|
| Cell line source(s) | Human skin biopsy-derived primary fibroblasts were generated in our lab from skin biopsies isolated from individuals' forearms.<br>Human neuroblastoma cells (SK-N-SH; https://www.atcc.org/products/htb-11), human non-small cell lung cancer cells (Calu-1; https://www.atcc.org/products/htb-54 ), and African green monkey kidney cells (Vero E6; https://www.atcc.org/products/crl-1586 and Vero; https://www.atcc.org/products/ccl-81) were purchased from ATCC company. |
| Authentication | All patient derived cell lines were verified to carry the POLG1 patient mutation of interest, validating them to represent the disease of interest. |
| Mycoplasma contamination | The cell lines have been tested for absence of mycoplasma contamination. |
| Commonly misidentified lines (See ICLAC register) | n/a |

# Animals and other research organisms

Policy information about studies involving animals; ARRIVE guidelines recommended for reporting animal research, and Sex and Gender in Research

| | |
|---|---|
| Laboratory animals | Mus musculus, C57BL/6JOlaHsd mice.<br>Mouse samples were collected from female mice age ~3 months, 1 and 2+ years old as indicated in the respective figure legends. |

Mice were housed in controlled room at 22°C (30-40% humidity) with 12h light/dark cycle and ad libitum access to food and water and were regularly monitored for weight and food consumption.

Wild animals | The study did not involve wild animals

Reporting on sex | We have used female mice in this study (except for behavioral analyses and in GABAergic marker staining where male and female mice were included but they were analysed separately based on the animal sex). The sex of the mice were stated in the respective figure legends.

Field-collected samples | The study did not involve samples collected from the field

Ethics oversight | Animal experimental procedures were approved by the Animal Experimental Board of Finland (license numbers ESAVI/689/4.10.07/2015 and ESAVI/3686/2021).

Note that full information on the approval of the study protocol must also be provided in the manuscript.

# Plants

Seed stocks | *Report on the source of all seed stocks or other plant material used. If applicable, state the seed stock centre and catalogue number. If plant specimens were collected from the field, describe the collection location, date and sampling procedures.*

Novel plant genotypes | *Describe the methods by which all novel plant genotypes were produced. This includes those generated by transgenic approaches, gene editing, chemical/radiation-based mutagenesis and hybridization. For transgenic lines, describe the transformation method, the number of independent lines analyzed and the generation upon which experiments were performed. For gene-edited lines, describe the editor used, the endogenous sequence targeted for editing, the targeting guide RNA sequence (if applicable) and how the editor was applied.*

Authentication | *Describe any authentication procedures for each seed stock used or novel genotype generated. Describe any experiments used to assess the effect of a mutation and, where applicable, how potential secondary effects (e.g. second site T-DNA insertions, mosiacism, off-target gene editing) were examined.*

