## [Peer Review File · Nature]

Manuscript Title: Ancestral allele of DNA polymerase gamma modifies antiviral tolerance

Reviewer Comments & Author Rebuttals

Reviewer Reports on the Initial Version:

Referees' comments:

Referee #1 (Remarks to the Author):

In this manuscript, Kang et al use an impressive array of patient samples to test the impact of disease-causing POLG1 alleles on anti-viral immune responses. The manuscript used MIRAS patient fibroblasts and brain samples to perform -omic analyses and found that patient-derived cells show reduced basal immune gene expression. They further verified their claims experimentally with in vitro cell culture work. The findings are highly novel and interesting. Moreover, it is on a very timely subject (SARS-CoV-2 infection) and is important for the field. However, in its current state, the manuscript seems incomplete and almost rushed. Some data are inconsistent, making it hard to reach a clear conclusion. Moreover, the inconsistencies are not addressed in the text – while the brevity of the text is appreciated, there are significant discussions necessary to clarify the data. This work has the potential to be highly illuminating and impactful if the authors can address the following weaknesses to substantiate their claims.

Major concerns:

1) The authors suggest that MIRAS cells have a delayed response to viral agents at first, which allows viral replication in its early stages resulting in a hyperactive response at later timepoints. This is mainly based on results of dsDNA stimulation (Fig 1H/K). However, in response to dsRNA, MIRAS MEF cells show increased immune gene expression at both timepoints (Fig 1H), whereas MIRAS patient fibroblasts show reduced immune gene expression at both timepoints (Fig 1K). The authors should comment on this seemingly contradicting data between mouse and humans. In addition, exposure directly to virus in Fig 2 show different results (A-C show a sustained decrease even at the 48hr timepoint, and D shows further inconsistencies between TBEV and SARS-CoV-2 for IL-6 levels). Finally, the authors attempt to substantiate their claims by measuring viral load at 48hrs. While the data is promising, without a time-course showing faster viral replication in MIRAS cells, it is hard to draw the conclusions the authors are making. Moreover, it is unclear from the data provided whether the increased viral load is actually due to dampened immune response. Ideally, the authors can demonstrate that dampening viral response in wild-type cells result in increased viral load to strengthen their conclusions.

2) In Fig 2 the authors performed all the experiments using two healthy donor fibroblasts and two MIRAS patient fibroblasts. However, Fig 2B-D show high variability between the two MIRAS fibroblasts. This undermines the validity of the related conclusions. More donor fibroblasts are required in these experiments to perform a proper statistical analysis for a convincing conclusion. While it is appreciated that acquiring patient samples is not trivial, the acquisition of 3 diseased and 5 control samples for brain samples suggest that the authors have the capacity to minimally bring their numbers up to the same level.

3) Fig 3 shows that POLG1-KD human fibroblasts have increased immune activity. This is contradictory to Fig 1 and 2 where low POLG1 expression (in MIRAS MEFs and patient cells) is associated with reduced immune activity. The authors need to address these differences.

4) In Fig 3, the authors found that inhibition of the proteasome restored POLG2 expression, which they mention is highly interesting since POLG2 is normally in the matrix. They then speculate this may be due to failed import into the mitochondria, which is another example of extrapolation from their data. Minimally, the authors should look at general mitochondrial import upon viral infection and see if that is different. Ideally, they should determine whether mitochondrial POLG2 actually decreases upon viral exposure or any of the conditions in this study.

5) Following the last point, the manuscript is lacking mechanistic insight on how POLG1 may

impact immune response. Since POLG1 is known to be a regulator of mitochondrial homeostasis and mitochondria are signaling hubs for innate immune signaling, the authors should perform more comprehensive analyses on mitochondrial integrity and functions to test whether mitochondria may play a role here.

6) Although fibroblasts have been shown to be able to produce immune factors and play a role in regulating innate and adaptive immunity, a much more complicated cellular network is involved in determining the severity and outcome of a viral infection. In this study, *in vivo* mouse models are lacking for assessing whether the POLG1 mutation may lead to delayed viral clearance, immune overactivation etc. Indeed, it seems that the authors do have access to the MIRAS mice. *In vivo* models are required to provide compelling evidence that MIRAS or POLG1 mutations lead to increased sensitivity to viral infections.

7) The lack of follow-up on the brain studies is shocking. The authors very substantiate their bioinformatics study for fibroblasts with follow-ups in both MEFs and patient-derived fibroblasts, but nothing was done for the brain section. Moreover, very little meta-analyses were performed to compare the fibroblast and brain computational studies, leading the reader desiring a lot more. This manuscript would benefit from moving the brain analyses to the beginning with the fibroblast studies, which would allow direct comparative analyses of the two datasets. Moreover, the brain tissue consists of multiple cell types. It is unknown whether the transcriptional changes are caused by altered cellular composition, and/or altered response of a certain cell type. Therefore, it would be ideal to perform single-cell transcriptomic analysis to gain more insights of this tissue. If this is unfeasible, the authors can also perform analyses in cultured brain cells using the same methods for fibroblasts, which should be an easy task for this group considering their expertise in culturing several different types of brain cells.

8) Fig 1 and Fig 4. The transcriptomics clearly show that anti-viral immune genes are down-regulated in MIRAS cells under basal, virus-free conditions. Although these genes are activated during viral infections and thus classified as "viral-associated" genes, their basal expression shall be controlled by cellular endogenous components, such as cytosolic mtDNA and mtRNA. In this case, their reduction shall not be interpreted as compromised anti-viral immunity (PAMP pathways), but may reflect changes in cellular stress responses (DAMP pathways).

9) Minor points: While the inclusion of SARS-CoV-2 is timely and interesting, it does not seem to fit the model presented by the authors and thus almost seems forced in this manuscript. Upon looking at Fig. 2D, it seems that the SARS-CoV-2 changes to IL-6 levels indicate there is actually a decrease in levels both at the early and late timepoints, which is inconsistent with other conditions showing decrease at early time points then an increase at late time points. Even comparing to TBEV in the same figure, you can clearly see the phenotypes at 48h are opposite. This begs the question of whether SARS-CoV-2 follows the same model and if it's even worth including in this manuscript. Minimally, the authors need to address this point.

10) Fig 2B and 2C are inconsistent. In response to TBEV infection, M1 actually shows stronger or comparable immune activation as C1 and C2 in Fig 2B, but in Fig 2C M1 response is potently suppressed. The way of quantification performed in Fig 2C is confusing and the presentation is unacceptable. Data should be displayed as a bar graph similar to Fig. 3A with clear display of data distribution. It seems that C1 and M1 as well as C2 and M2 were grouped for comparison. The decision to group these patients rather than averaging a population needs to be described in detail. Finally, the order of data presented in 2B and 2C are different, making it more difficult to look at the data.

11) Fig 2F shows that both M1 and M2 have very high TBEV viral load 48h post infection. However M1 cells do induce RIG1, MDA5, IFIT3 at 24h and 48h to a comparable extent as C2 (Fig 2B). M1 cells also show comparable IL-6 secretion in three out of four conditions (Fig 2D). Does that mean the intracellular viral load is not necessarily related to the efficiency of immune responses in these experiments? Moreover, Fig 2H shows that M2 releases less viral RNA into media than M1. Is this difference related to differential immune activation between M1 and M2?

12) Through directly comparing POLG1 and POLG2 protein levels upon shRNA (Fig. 3A), the authors claim that shRNA of one results in knockdown of both due to their interdependence. However, upon a more physiological condition (HSV-1 infection) there is a depletion of POLG2 with

only a minor change to POLG1. This contradicts the claim of interdependence. The authors need to clarify this difference between shRNA and actual infection.

13) Fig 4. The comparison of the MIRAS brain samples to MELAS syndrome is confusing. It is unclear what is the point the authors are trying to convey in this comparison. It either needs to be described better or removed from the manuscript.

14) Fig 4. Do these three MIRAS patients all carry POLG1 mutations?

15) All western blots shall be quantified as in Fig 3A.

Referee #2 (Remarks to the Author):

In this study, Kang et al. present evidence of dysregulated innate immune signaling in mitochondrial recessive ataxia syndrome (MIRAS). They found that MEFs from knock-in mice carrying a MIRAS patient pathogenic POLG1 allele display basal downregulation in immune-related gene expression and dampened activation of innate immune signaling pathways by nucleic acid agonists, which was recapitulated to a significant degree in fibroblasts from MIRAS patients (MIRAS-pt). Interestingly, infection of MIRAS-pt fibroblasts with TBEV or SARS-CoV2 revealed that cells harboring this POLG1 mutation had a dampened innate immune response following viral infection, which the authors postulate allows for the increased viral load observed in these cells. Transcriptomic analysis of MIRAS-pt brains also found suppression of innate immune signaling. The role of mitochondria in innate and adaptive immunity is an exciting and active area, yet the role of disruptions in this axis in mitochondrial diseases has not been addressed extensively to date. Thus, the main observation here that innate immunity is dysregulated in MIRAS is of great interest to the field and potentially relevant to pathogenic mechanisms in certain mitochondrial diseases. However, the study is largely descriptive and correlative, leaving the nature of the mitochondrial stress involved and the mechanism of immune system deregulation unclear. These and other issues diminish the impact of the study at this juncture.

Major Points:

1) It is not clear what the authors mean by "mtDNA replisome stress" other than simply mutations in POLG1, the replicative mtDNA polymerase. For example, the cells show no change in mtDNA copy number and other clear mitochondrial phenotypes are not elucidated, so the nature of the mtDNA/mitochondrial stress that the authors claim results in altered innate immune regulation remains a mystery.

2) The authors also do not provide a clear mechanism for the aberrant immune responses observed in MIRAS MEFs and MIRAS-pt fibroblasts. Although Fig. 3 is presented as an investigation into what mediates the signaling (line 198-199), no clear mechanism is elucidated or proposed. The authors do test immune responses under a variety of related conditions to try to unravel this (e.g. knock-down of POLG1 and 2, expression of HSV-1 UL12.5). But these cause quite different innate immune responses than those observed in the patient fibroblasts making interpretations muddy. The data in Fig. 3 presents interesting findings concerning the regulation of POLG1 and 2 during viral infection, but does not really inform what is occurring in cells expressing the POLG1 MIRAS-allele or why in the MIRAS case there is a dampening of the immune response (Figure 1K and 2B-C). Have the authors considered that reduced release of mtDNA and/or mtRNA in the MIRAS mouse and human cells is the reason for the reduced innate immune signaling?

3) There is contradictory data throughout the manuscript that is difficult to interpret. For example, the MIRAS MEFs in Fig. 1 appear to have diminished POLG1/2 as well as diminished innate immune protein expression (e.g., MDA5, RIG-I), but POLG1/2 knockdown (Fig. 3) causes increased innate immune protein expression. How do the authors explain this? Also, POLG1 does not appear to be knocked down to significant levels in Fig. 3A. Finally, it is also not clear from the methods section what the negative control was for the experiment in Fig. 3A. If an shRNA control (e.g.

scrambled) was not used, the increases in RIG-I, MDA5, and ISGs might be explained simply by introduction of innate immune activating double-stranded RNA.

4) Similar to point #3, it is confusing that in Fig. 1, the response to poly(I:C) in patient cells is diminished at early time points and not changed at later time points, whereas in the MIRAS MEFs there is a modestly enhanced response at early time points and a heightened response at later time points. These inconsistent responses in different cell types raise concerns about the role of POLG1 mutations per se in response to exogenous RNA, which is of additional concern in interpreting the data during viral infection.

5) The effects of TBEV on RIG-I/MDA5/IFIT3 (Fig. 2B) and IL6 levels (Fig. 2D) differ significantly in the two patients' cells, making the actual MIRAS phenotype unclear.

6) The authors state that HSV infection kills both their MIRAS-pt and control fibroblasts (lines 169-171) after 24 hours, yet show data from fibroblasts 48 hours after infection with HSV (Figure 3F).

7) More information on donor cells is needed. The authors may need to better control for sex or age to be able to make solid conclusions. For example, age has a huge impact on inflammation profiles/responses. Realizing this may be difficult due to the rarity of the disease, it remains hard to draw solid conclusion on such a small sample size.

8) According to Fig. 3C, there is only a ~20% reduction of mtDNA after 48 hrs of UL12.5 expression. Based on other published studies, this should be more than enough time to completely degrade mtDNA. There may be issues with the UL12.5 construct used or only a low percentage of transfected cells was achieved that might be confounding this experiment.

9) Throughout the manuscript, many of the western blots appear to contradict each other and/or do not appear to match the accompanying quantification (e.g., POLG1/2 levels are diminished in one experiment, but not the other in Fig. 1E, same for RIGI/MDA5/IFIT3 in Fig. 2B). In Figures 1E, 3B, Extended data 2B-C, the authors should provide a loading-control image, instead of just the quantification.

Minor Points:

1) In Figure 2D, please indicate in the legend what the bars on the side of the graph are comparing.

2) Please ensure that all extended data figure legends in which statistical analysis was performed contain p-value definitions.

3) In Fig. 2, the analysis should include control uninfected cells.

Referee #3 (Remarks to the Author):

In their manuscript "Mutant mitochondrial DNA polymerase impairs acute antiviral immunity" Kang et al investigate the innate immune status in patients, patient-derived cell-culture models and in vitro mouse models of Mitochondrial recessive ataxia syndrome (MIRAS). They report a decreased activation of the type I IFN system at steady state in all three systems. Although MIRAS is caused by mutations in POLG1 that lead to destabilization of the protein, the authors (confusingly) show that directly decreasing POLG1 levels (using shRNA) leads to the opposite outcome (increased IFN levels).

The author's observation of decreased IFN levels in MIRAS patients is noteworthy, because it is

validated in primary patient samples, and is unexpected because shRNA perturbation and other POLG mutations (e.g. <https://www.biorxiv.org/content/10.1101/2020.09.22.308171v1.full.pdf>) show the opposite effect. However, the study is mainly descriptive and lacks mechanistic insights. Most critically, it is not clear how the mutations in POLG1 result in suppression of the basal interferon response. Is there an IFN-inducing pathway defective in MIRAS-MEFs or is there an IFN-decreasing pathway over-activated in MIRAS-MEFs? Which pathway is that? Is the same pathway responsible for the phenotype seen in patients/brain of patients?

The authors overall model is that lower IFN levels in MIRAS patients lead to increased susceptibility/pathology to virus infection. Subsequent over-activation of the immune system or increased viral load is then proposed to promote disease. Except for the lower levels of IFN in MIRAS MEFs and brain samples, there is no solid evidence for this model. In fact, the authors find decreased IFN levels in MIRAS patient brains, so their idea of increased immune activation or viral infection as the driver of MIRAS disease is seemingly inconsistent with their data.

The authors claim that "Our evidence suggests that viral infections contribute to mitochondrial disease manifestations in patients with genetic predisposition" (line 33ff), "that the [...] immune overactivation and toxic cytokine induction contributes to nervous system pathology in MIRAS" (line 192ff) and that "our findings suggest [...] delayed and augmented inflammatory reaction to promote disease manifestation" (line 269ff). However none of these speculative claims is directly supported by data in the manuscript.

In fact, while the authors claim that there is an initially delay in interferon-stimulated gene expression in MIRAS MEFs, it is not clear this is really the case. For example, the panels "untreated" and "10h dsDNA" in Fig 1 K look exactly the same, indicating that while MIRAS-MEFs start at a lower level of ISG15, IFNB1 and IL1B, their induction is comparable to WT cells. This would be easier to discern if the samples were normalized to untreated rather than to wildtype at each timepoint.

Overall, while there are some interesting preliminary observations here that appear to link POLG1 to innate immunity, there is no mechanistic understanding of what is happening either in cells in vitro or in patients.

Other comments

- Why are IFN levels higher in MIRAS MEFs at later points of PAMP stimulation?
- Lack of controls, e.g. Fig 2A, B lack uninfected condition
- Why is there no band for POLG2 in lane 2 of Fig3E?
- Can the patient fibroblasts be complemented with WT POLG1 to prove that the phenotype of these fibroblasts is indeed due to the POLG1 mutation

Referee #4 (Remarks to the Author):

Kang Y. et al. explored how recessive conserved POLG1 mutations that originated in Nord-European populations, and which are linked to MIRAS etiology, affect the innate immune response to viral infections. Authors report that either murine or human cells harboring the MIRAS-linked POLG1 mutations display an aberrant innate-immune response to viral infection with a delayed antiviral transcriptional response that allows increased viral infection and then, an exacerbated production of inflammatory cytokines. The authors suggest these immune defects contribute to the development of the patient neurological pathologies.

After evaluating transcriptomes of autopsy-derived brain tissue from control, MIRAS and MELAS patients, the authors conclude that MIRAS pathology is related to a downregulation of immunity-

related genes and that this is a specific hallmark of POLG1-related MIRAS and not a common feature in mitochondrial neurodegenerative diseases. However, MELAS may be the exception and, as this is quite interesting, this point would be strengthened by analyzing other mitochondria protein mutation linked diseases.

The manuscript shows novel and interesting results linking POLG1 mutations to immune responses and viral infection, which may have clinical relevance. However, authors do not address any mechanisms underlying the relationship between POLG1 and the antiviral response or how immune dysregulation may be linked to neurodegenerative outcomes.

Specific points:

- 1) The knock-in mutant POLG1 mouse generated for this study is a very valuable MIRAS model. However, information about the phenotypes of the mutant animals is missing. Do the W726S/E1121G POLG1 mutant mice display any MIRAS related phenotypes? How is the immune response to viral infection in these animals? Does viral infection alter any phenotypes in the mice? Such experiments would get to the heart of the authors conclusions related to MIRAS disease mechanism.
- 2) Information about the overall consequences of the POLG1 mutations on transcription is missing. Which is the total number of transcriptionally altered genes (both upregulated and downregulated)? What proportion of these up/down regulated genes are immune-related? Volcano plots obtained from the statistical analysis of the transcriptomic results are missing. Highlighting the genes that are displayed in Fig.1B in the respective volcano plot would provide visual insight into the proportion of total genes displaying a different transcription pattern that are immune-related.
- 3) The protocol used to generate the gene ontology enrichment analyses is not mentioned in the methods section.
- 4) Figure 1C and D display only a very small set of genes and pathways - how representative are the changes in those categories compared to all the impaired biological processes? Adding a bubble plot of all the enriched GO categories that were altered due to the POLG1 mutations would ease the interpretation of the transcriptomics and proteomics experiments of the manuscript. This type of plot gives information on the GO enrichment score, p-value, and number of genes in each GO category, allowing a more complete picture.
- 5) Authors indicate that cells harboring the studied POLG1 mutations have a delayed and then overactive immune response, but the transcript levels of the cytokines evaluated are normalized and plotted independently at each infection time and not as a time-lapse, which does not allow a comparison of the magnitude of the expression changes between the short and long infections (Figures 1G, H, I, K and 2A). Also, it is not discussed why authors chose to analyze a specific set of genes. Was this decision derived from their transcriptomic/proteomic data? And why different sets of genes were used in the different times and experiments?
- 6) Immunoblot presented in the extended figure 3A and the quantification of this in Figure 1J shows that ~50% of the fibroblasts from control patients display a decreased expression of POLG2 and of the immune-related genes evaluated. A violin plot would yield a more clear visualization of the data.
- 7) In extended Fig. 3A the authors analyze 7 control and 5 MIRAS patients cell lines. Authors analyzed further only two of the patient-derived fibroblasts lines for the control and MIRAS groups. There is no information regarding which two and how these were chosen. Please explain.
- 8) There is an almost six-fold increase in the expression of IL-1 β observed in mutant MEFs after 24hr of dsRNA transfection, whereas patient derived fibroblasts present the opposite behavior upon the same treatment for the same time. This discrepancy in Fig 1I and Fig 1K between MEFs and patient fibroblast should be mentioned in the text.
- 9) In Figure 3, authors address the consequences of expressing the HSV-1 protein UL12.5. These data show that the POLG2 and innate immune gene expression is altered by HSV-1 but does not give any mechanistic insight on the role of POLG1 in the viral innate immune response, nor on how the W726S/E1121G POLG1 mutations are related to the immunosuppression phenotype. Is Fig. 3E labeled correctly? Why is there so little POLG2 in untreated cells in lane 2? And the large increase

upon treatment with MG132 in lane 1 is surprising.

13 Transcriptomic analysis of MIRAS autopsy brains shows changes in immune signaling pathways and translation machinery. It is not clear which of these, if either, is linked to disease phenotypes. As the authors suggest that virus infection may be exacerbated in MIRAS patients, evidence of increased virus gene expression in the autopsy brain would be interesting to assess.

More minor comments

1) The manuscript lacks information about the patient-derived fibroblasts and autopsy samples. Did authors analyze if the chosen subjects had the W726S/E1121G POLG1 mutations? More details are needed.

2) The number of biological replicates that were used for the transcriptomic and proteomic experiments should be included in the methods section.

3) The authors cite work on cytokine increases in certain PD patient serum samples. A subsequent and more thorough analysis would be good to mention as they also find increased mtDNA in patient sera (Brain 143: 3041, 2020).

Author Rebuttals to Initial Comments:

Referees' comments:

Referee #1 (Remarks to the Author):

In this manuscript, Kang et al use an impressive array of patient samples to test the impact of disease-causing POLG1 alleles on anti-viral immune responses. The manuscript used MIRAS patient fibroblasts and brain samples to perform -omic analyses and found that patient-derived cells show reduced basal immune gene expression. They further verified their claims experimentally with in vitro cell culture work. ***The findings are highly novel and interesting. Moreover, it is on a very timely subject (SARS-CoV-2 infection) and is important for the field.*** However, in its current state, the manuscript seems incomplete and almost rushed. Some data are inconsistent, making it hard to reach a clear conclusion. Moreover, the inconsistencies are not addressed in the text – while the brevity of the text is appreciated, there are significant discussions necessary to clarify the data. This work has the potential to be highly illuminating and impactful if the authors can address the following weaknesses to substantiate their claims.

We thank the Reviewer for considering our findings novel, interesting and timely and for the constructive feedback to substantiate our findings and have addressed all comments. We have carefully and extensively revised the manuscript, as detailed below.

Reviewer:

1) The authors suggest that MIRAS cells have a delayed response to viral agents at first, which allows viral replication in its early stages resulting in a hyperactive response at later timepoints. This is mainly based on results of dsDNA stimulation (Fig 1H/K). However, in response to dsRNA, MIRAS MEF cells show increased immune gene expression at both timepoints (Fig 1H), whereas MIRAS patient fibroblasts show reduced immune gene expression at both timepoints (Fig 1K). The authors should comment on this seemingly contradicting data between mouse and humans. In addition, exposure directly to virus in Fig 2 show different results (A-C show a sustained decrease even at the 48hr timepoint, and D shows further inconsistencies between TBEV and SARS-CoV-2 for IL-6 levels). Finally, the authors attempt to substantiate their claims by measuring viral load at 48hrs. While the data is promising, without a time-course showing faster viral replication in MIRAS cells, it is hard to draw the conclusions the authors are making. Moreover, it is unclear from the data provided whether the increased viral load is actually due to dampened immune response. Ideally, the authors can demonstrate that dampening viral response in wild-type cells result in increased viral load to strengthen their conclusions.

2) In Fig 2 the authors performed all the experiments using two healthy donor fibroblasts and two MIRAS patient fibroblasts. However, Fig 2B-D show high variability between the two MIRAS fibroblasts. This undermines the validity of the related conclusions. More donor fibroblasts are required in these experiments to perform a proper statistical analysis for a convincing conclusion. While it is appreciated that acquiring patient samples is not trivial, the acquisition of 3 diseased and 5 control samples for brain samples suggest that the authors have the capacity to minimally bring their numbers up to the same level.

Authors: *Combined response for comments 1 and 2*

We thank for the comments and agree with the reviewer. To exclude the immunological differences between human and mouse lines and to reach our overarching goal to understand immune modulation in human MIRAS disease, we made an effort to gather more human and *in vivo* data. We succeeded to sample ***more MIRAS patients, all homozygous for the ancestral gene variant leading to p.W748S amino acid change***, and established their primary fibroblasts cultures. We also introduce

a new in vivo model: MIRAS-knockin mouse model carrying the homologous, conserved variants of our patients (human p.W748S), and analyse these for immune responses.

In the revised manuscript we report robust findings of aberrant immunity of MIRAS patients in new systems (patient cell cultures, serum samples, MIRAS knockin mice), viral exposures as well as utilize population genetics of >342,000 individuals with genetic and medical data (FinnGen database):

- i) ***Dysregulated innate immune responses to viral mimetic and virus infections of primary MIRAS patient cells.*** We tested six biologically independent MIRAS patient primary fibroblast lines with their age and gender-matched controls, overall eliciting an attenuated initial IFN-1 signaling favouring viral replication with promoted cytotoxicity during viral infection. We subjected them to time course treatment of either **(1) immunogenic mimetics for viral-associated molecular patterns (dsDNA or dsRNA i.e. poly (I:C)) or (2) DNA or RNA virus infection (HSV-1, TBEV and SARS-CoV-2)**. These viruses were selected based on their ability to cause severe, delayed complications in a minority of infected patients, suggesting susceptibility factors predisposing to severe delayed symptoms. We demonstrate that innate immune signaling, especially the IFN-1 dependent pathways, are inhibited, including sequential timepoints as suggested by the reviewer. Furthermore, we show that the patient cells allow higher replication of the virus in the initial infection compared to the controls. This is true for all the tested viruses. These new results strengthen our results in the original submission. **[New Data in revised Figure 1-2].**
- ii) ***Dysregulated immune profile, including IL-6 activation in MIRAS patients sera (n=15 patients) [Data in revised Figure 4J]***
- iii) ***Reduced POLG1 protein and its associated complex, as well as dampened IFN-1 signaling/virus defense gene expression profile in MIRAS patient cerebral cortex and cerebellum autopsy samples (n=3 patients) [Data in revised Figure 5]***
- iv) ***Population genetic and medical data from >342,000 subjects (FinnGen) indicate with high significance that specifically the p.W748S MIRAS variant carriers exhibit enriched immunodeficiency-linked morbidities.*** Similar association was not found in a set of other mitochondrial disease genes. **[New data in revised Figure 1A]**
- v) ***Reduced basal level of POLG1 and mtDNA synthesis rate in the cerebral cortex and liver of MIRAS mice.*** The mice showed diminished release of protective antiviral IFN- α and IFN- β cytokines into the sera of MIRAS mice in early stage infection (p.i. day 4) and weakened IFN-1 signaling also in the cerebral cortex of MIRAS mice. Progressively during the infection, the pro-inflammatory IL-6 was released into their sera with an increasing trend. The liver showed remarkable sensitivity to TBEV infection. It was a surprise for us to find such hepatic sensitivity of MIRAS-mice to a neurotropic virus TBEV, but some previous reports have found liver inflammation associated with this virus (Bogovič et al. 2022; Misić-Majerus et al. 2005). While recognizing potential differences in species, the finding brings an interesting potential relevance to the mechanisms of the dramatic MIRAS-related valproate hepatotoxicity: a subclinical hepatic inflammation during the acute manifestation of MIRAS upon a viral infection would rapidly worsen when challenged with valproate and cause subacute liver failure. ***Our in vivo data indicate that MIRAS allele results in dysregulated immune response and decreased murine tolerance particularly to virus infection, most remarkable in the liver. [New data revised Figure 3-4].***

All in all, these data in patient cells, autopsy samples, population data from 342,499 individuals as well as in vivo mouse findings together indicate aberrant immune and inflammatory responses in MIRAS. We propose that these events contribute to juvenile, valproate-sensitive MIRAS manifestation mimicking in severity and acuteness of viral-induced encephalitis (especially HSV-1 and TBEV).

Reviewer:

3) Fig 3 shows that POLG1-KD human fibroblasts have increased immune activity. This is contradictory to Fig 1 and 2 where low POLG1 expression (in MIRAS MEFs and patient cells) is associated with reduced immune activity. The authors need to address these differences.

4) In Fig 3, the authors found that inhibition of the proteasome restored POLG2 expression, which they mention is highly interesting since POLG2 is normally in the matrix. They then speculate this may be due to failed import into the mitochondria, which is another example of extrapolation from their data. Minimally, the authors should look at general mitochondrial import upon viral infection and see if that is different. Ideally, they should determine whether mitochondrial POLG2 actually decreases upon viral exposure or any of the conditions in this study.

Authors: Combined response for comments 3 and 4

We agree with the Reviewer. Based on his/her and other reviewers' comments we have now completely revised our manuscript with extensive new data focusing on MIRAS patients primary cells, a novel knockin-mouse and *in vivo* infection data. All our data is now based on models that carry the homozygous MIRAS founder mutation c.2243G>C (p.W748S; coinciding with neutral p.E1143G variant in cis in the same allele) or the homologous conserved site in mouse POLG1, all giving insights specifically relevant for MIRAS disease. We have therefore removed the POLG1-KD and MG132 proteasome inhibition data because their relevance to the current findings was still unclear. This, to our opinion, improved clarity and focus of the revised manuscript.

Reviewer:

5) Following the last point, the manuscript is lacking mechanistic insight on how POLG1 may impact immune response. Since POLG1 is known to be a regulator of mitochondrial homeostasis and mitochondria are signaling hubs for innate immune signaling, the authors should perform more comprehensive analyses on mitochondrial integrity and functions to test whether mitochondria may play a role here.

Authors: Thank you for the suggestions. The new *in vivo* data deepened the mechanistic aspect. Recent reports in experimental knock-outs of mitochondrial replisome proteins or mitochondrial directed endonuclease-induced mtDNA breaks (West et al. 2015; Tigano et al. 2021) induced mtDNA stress that **activates** viral sensors. Our data from an actual mitochondrial disease indicate **attenuated/delayed** initial activation of the IFN-1 signaling particularly via RIG-I sensing pathway allowing early replication of the virus in MIRAS models with promoted inflammatory response.

Previously, the W748S mutant POLG has been structurally modelled and analysed by atomistic molecular dynamics simulation to affect replisome protein interactions (Euro et al. 2011; Euro et al. 2017). Furthermore, increased mtDNA replisome activity was shown to affect cellular nucleotide pools, especially their prioritization to mitochondria vs. nucleus (Hämäläinen et al. 2019). These data suggest that W748S POLG1 mutant replicase would lead to instable mtDNA replisome, decreased processivity and lowered nucleotide pool prioritization for mitochondria. Indeed, our experimental data support the modelling results:

We show that the brain, liver (the major organs affected in MIRAS disease) and immune-active organ spleen, all had remarkably diminished amount of POLG1 protein and modest decrease in POLG2 in MIRAS mice (Figure 3A). We also show the depletion of POLG1 protein and its associated native replisome complex in MIRAS patient brain autopsy (cerebral cortex and cerebellum) samples (Figure 5A)

indicating decreased assembly of the replisome complex. These data, together with the basal mtDNA decrease in the MIRAS mouse liver (Figure 3B) suggest decreased mtDNA replication activity / processivity. This we show to be true: *in vivo* mtDNA replication is lowered in MIRAS mice – BrdU labelling of nascent mtDNA compared to total mtDNA is decreased in MIRAS liver and brain (Figure 3B). All these data together indicate that p.W748S variant of POLG causes insufficient mtDNA replisome assembly and processivity *in vivo* in the basal state of the disease.

Upon infection, TBEV replication causes rapid mtDNA depletion in MIRAS mouse brain compared to similarly infected controls (Figure 4D), supporting lowered availability of nucleotides for mtDNA replication in MIRAS after activation of viral replication. While it remains possible that TBEV has a direct effect on mtDNA stability, similar to what has been shown for HSV-1, we consider the nucleotide pool depletion effect to mtDNA replication as the most likely cause of mtDNA depletion. The metabolomic analyses of the TBEV-infected MIRAS mouse brains revealed significantly lowered nucleotides, with pyrimidine deoxyribonucleotides and pyrimidine ribonucleotide interconversion being the two most significantly changed pathways (Figure 4A, B, C). Also MIRAS patient fibroblasts showed low steady-state POLG1 protein and accelerated mtDNA depletion when infected with HSV-1.

The data suggest the following sequence of pathogenesis: 1) W748S mutation causes poor POLG1 contacts with other mtDNA replisome proteins, causing replisome instability and consequent decreased processivity; 2) lowered mtDNA replication activity results in mtDNA depletion; 3) decreased mtDNA replication and amount reduce the cellular ability to present mtDNA /RNA fragments to cytoplasm to activate viral sensors; 4) enhanced viral replication in the immunocompromised cellular environment hijacks dNTP pools; 5) rapid viral invasion together with acute mtDNA depletion and dysfunctional MIRAS replisome, cause progressive mtDNA depletion leading to cellular energy crisis, delayed strong proinflammatory response and reduced viability via necroptosis.

Our findings and those from Finnish population, (FinnGen) POLG1-W748S variant is strongly associated with delayed/ decreased innate immunity. The evidence suggests that active mtDNA replication, instead of mtDNA fragmentation, is necessary for innate immunity response to viral infection in the brain and liver.

[New data in revised Figure 1B, 2C (patient fibroblasts), 3A, 3B, 4D (mouse tissues), 5A (Patient brain autopsies), presented and discussed in the manuscript text].

Of mtDNA integrity: previous studies of ours and others have characterized in detail mtDNA integrity and respiratory chain functions in MIRAS/MSCA-E patients (Norwegian researchers that identified MIRAS the same time as us called it MSCA-E) from biopsy samples and autopsy-derived tissues. These studies indicated presence of a marginal mtDNA depletion and/or deletions or no such findings in the skeletal muscle, and mtDNA depletion in the liver and the brain. We had included examples of these references in the paper, and also include them here (e.g. (Hakonen et al. 2008; Hakonen et al. 2005; Van Goethem et al. 2004; Winterthun et al. 2005; Lujan et al. 2020; Palin et al. 2010). For example, paper by Winterthun commented in their original description of MSCA-E” Multiple deletions are found in the muscle of all of our patients, but may be present in low concentration and not associated with a biochemical phenotype.” And in Hakonen: “histological signs of respiratory chain deficiency was not present in any of the patients”. The mild mtDNA- and respiratory chain-related findings are in remarkable contrast with the dramatic encephalitic disease course of the teenagers with MIRAS and do not explain the severe phenotype. Therefore, alternative mechanisms needed to be sought and are the topic of this manuscript.

Indeed, our data demonstrate the intriguing cross-talk of viruses and mtDNA maintenance. The delayed activation of innate immunity in MIRAS cells and mice indicates that mitochondria modify innate immunity via different signals, but the responses are dysfunction-specific.

Reviewer:

6) Although fibroblasts have been shown to be able to produce immune factors and play a role in regulating innate and adaptive immunity, a much more complicated cellular network is involved in determining the severity and outcome of a viral infection. In this study, *in vivo* mouse models are lacking for assessing whether the POLG1 mutation may lead to delayed viral clearance, immune overactivation etc. Indeed, it seems that the authors do have access to the MIRAS mice. *In vivo* models are required to provide compelling evidence that MIRAS or POLG1 mutations lead to increased sensitivity to viral infections.

Authors: To address the comment of the Reviewer, we took the challenge and now introduce and report here a novel MIRAS mouse model carrying the homologous conserved changes in the MIRAS patients (human p.W748S+ E1143G in cis in the POLG1 polypeptide – mouse protein changes W726S + E1121G). This has also been addressed in combined response for comments 1+2 above. The results strongly support our findings in the human materials. The aberrant immune and inflammatory profiles of MIRAS mice provide novel *in vivo* evidence for impaired immune responses to viral infection and reduced viral infection tolerance in MIRAS. [Data in revised Figure 3-4]

Reviewer:

7) The lack of follow-up on the brain studies is shocking. The authors very substantiate their bioinformatics study for fibroblasts with follow-ups in both MEFs and patient-derived fibroblasts, but nothing was done for the brain section. Moreover, very little meta-analyses were performed to compare the fibroblast and brain computational studies, leading the reader desiring a lot more. This manuscript would benefit from moving the brain analyses to the beginning with the fibroblast studies, which would allow direct comparative analyses of the two datasets. Moreover, the brain tissue consists of multiple cell types. It is unknown whether the transcriptional changes are caused by altered cellular composition, and/or altered response of a certain cell type. Therefore, it would be ideal to perform single-cell transcriptomic analysis to gain more insights of this tissue. If this is unfeasible, the authors can also perform analyses in cultured brain cells using the same methods for fibroblasts, which should be an easy task for this group considering their expertise in culturing several different types of brain cells.

8) Fig 1 and Fig 4. The transcriptomics clearly show that anti-viral immune genes are down-regulated in MIRAS cells under basal, virus-free conditions. Although these genes are activated during viral infections and thus classified as “viral-associated” genes, their basal expression shall be controlled by cellular endogenous components, such as cytosolic mtDNA and mtRNA. In this case, their reduction shall not be interpreted as compromised anti-viral immunity (PAMP pathways), but may reflect changes in cellular stress responses (DAMP pathways).

Authors: Combined response for comments 7 and 8

We agree with the reviewer and have expanded this aspect especially with *in vivo* materials, as explained in detail above. In short, we added data from isolated mitochondria from the cerebral cortex and cerebellum of our patients’ brains (to control for potential neuronal death/dysfunction-induced mitochondrial loss) and found **significant reduced level of POLG1 protein level and its associated POLG complex** (native gel analyses), with modest mtDNA copy number reduction. This defect was also observed in the MIRAS mouse brain and patient fibroblast model. We also performed **comprehensive functional enrichment analyses on our patients cerebral cortex transcriptome** using additional toolsets, including manually curated IPA and InnateDB (curated for mammalian innate immune response) database analyses. We further investigated **cerebral cortex transcriptome of mice infected with TBEV** to establish the transcriptome typically changed by the virus and performed comparative analyses against the cerebral cortex transcriptome of patients’ autopsies to elucidate conserved responses. These pathways showed remarkable overlap: the pathways induced typically by TBEV in the mouse brain were significantly downregulated in the patient brains [New data in the Revised Figure 5D, Extended Data Figure 14A]. We explained in the text the rationale behind selecting TBEV

as the virus for infecting MIRAS mice model: TBEV is neurotropic and induces biphasic disease with timeline similar to MIRAS symptom onset in patients. HSV-1 or similar mouse viruses are not allowed to be used in our animal facility, even in safety level 3.

We agree with reviewer that bulk brain analysis is inherently challenged by the heterogeneity of cell types, but still the results reflect the joint function, the relevant context of the neural cells. Single-cell transcriptomic analyses performed from human brain autopsy samples are unfortunately to our opinion not reliable or even feasible both because of postmortem changes and storage as frozen. However, we argue that the *in vivo* results are more relevant than a single cell type analysis from the brain, especially when analyzing immune-related pathways. The neural culture lacks microglia and astrocytes that are immune-active cells of central nervous system; indeed, astrocytes were recently shown to be able to drive mitochondrial encephalopathies (Ignatenko et al. 2018; Ignatenko et al. 2020). To differentiate astrocytes, neurons and microglia from patient and control iPSCs for co-culture experiments, or to make brain organoids and introducing microglia to the culture are experiments that take several years and are beyond the scope of this article. We wish that our extensive new data, including knock-in mice, patient samples and even population genetic/phenotype data are convincing to the reviewer, as they indicate that MIRAS-mutation challenges innate immunity *in vivo*.

Reviewer:

Minor points:

9) While the inclusion of SARS-CoV-2 is timely and interesting, it does not seem to fit the model presented by the authors and thus almost seems forced in this manuscript. Upon looking at Fig. 2D, it seems that the SARS-CoV-2 changes to IL-6 levels indicate there is actually a decrease in levels both at the early and late timepoints, which is inconsistent with other conditions showing decrease at early time points then an increase at late time points. Even comparing to TBEV in the same figure, you can clearly see the phenotypes at 48h are opposite. This begs the question of whether SARS-CoV-2 follows the same model and if it's even worth including in this manuscript. Minimally, the authors need to address this point.

We thank the reviewer for pointing this out. In this revised manuscript, we have analysed in more detail the responses, using a larger panel of patient fibroblast lines. We wanted to test different kinds of viruses to see if the responses are specific to certain virus types, or more general. HSV-1, TBEV and SARS-CoV-2 infection represent ds-DNA and single-stranded positive-sense RNA viruses. They also share the feature of causing mild infection to most people, but in a minority of patients a biphasic severe disease. While some variation – not surprisingly – was present because of the distinct invasion/replication strategies of these viruses and host responses, we did find aberrant responses of MIRAS to all these viruses. The MIRAS patient fibroblasts showed an increase of pro-inflammatory NF- κ B transcription factors upon prolonged viral infection of HSV-1, TBEV or SARS-CoV-2 infections, all showing an increased viral load in MIRAS cells [*New data in revised Figure 2*]. The data show that MIRAS mutation does allow increased viral replication of several types of viruses in the early infection stage. To our opinion the SARS-CoV-2 data is important to make this point and therefore it is included in the revised manuscript.

Reviewer:

10) Fig 2B and 2C are inconsistent. In response to TBEV infection, M1 actually shows stronger or comparable immune activation as C1 and C2 in Fig 2B, but in Fig 2C M1 response is potentially suppressed. The way of quantification performed in Fig 2C is confusing and the presentation is unacceptable. Data should be displayed as a bar graph similar to Fig. 3A with clear display of data distribution. It seems that C1 and M1 as well as C2 and M2 were grouped for comparison. The decision to group these patients rather than averaging a population needs to be described in detail. Finally, the order of data presented in 2B and 2C are different, making it more difficult to look at the data.

Authors: We thank reviewer for careful reading of the paper. As explained above, we have completely revised the manuscript, expanding and concentrating to the human cell lines and *in vivo* data from mice. Each biological patient and control individual data points are now tabulated in box-and-whiskers plots, showing the median with interquartile range. The statistical significance of all the analyses are calculated by comparing the MIRAS patients or mice to the control group, using the 2-tailed unpaired student's t-test, with their p-value indicated on the plots.

Reviewer:

11) Fig 2F shows that both M1 and M2 have very high TBEV viral load 48h post infection. However M1 cells do induce RIG1, MDA5, IFIT3 at 24h and 48h to a comparable extent as C2 (Fig 2B). M1 cells also show comparable IL-6 secretion in three out of four conditions (Fig 2D). Does that mean the intracellular viral load is not necessarily related to the efficiency of immune responses in these experiments? Moreover, Fig 2H shows that M2 releases less viral RNA into media than M1. Is this difference related to differential immune activation between M1 and M2?

Authors: To account for inter-individual immune response variation, we have expanded our MIRAS patient primary fibroblast panel to six biological individual derived cell lines and compared them to control fibroblast cell lines that are age and gender matched, and subjected all cell lines to viral infection, in addition to viral mimetics. We quantified and tabulated each individual data point for comparison of MIRAS patient to control group. This has also been addressed in combined response for comments 1+2 above.

Reviewer:

12) Through directly comparing POLG1 and POLG2 protein levels upon shRNA (Fig. 3A), the authors claim that shRNA of one results in knockdown of both due to their interdependence. However, upon a more physiological condition (HSV-1 infection) there is a depletion of POLG2 with only a minor change to POLG1. This contradicts the claim of interdependence. The authors need to clarify this difference between shRNA and actual infection.

Authors: In the revised manuscript, we have now concentrated to actual human patient materials and *in vivo* mouse data as explained above. These data have been omitted.

Reviewer:

13) Fig 4. The comparison of the MIRAS brain samples to MELAS syndrome is confusing. It is unclear what is the point the authors are trying to convey in this comparison. It either needs to be described better or removed from the manuscript.

Authors: We agree and have removed the MELAS sample in the revised manuscript, as suggested by the reviewer.

Reviewer:

14) Fig 4. Do these three MIRAS patients all carry POLG1 mutations?

Authors: Yes, all the MIRAS patients studied here carry the same, identical POLG1 mutation c.2243G>C (p.W748S; coinciding with neutral p.E1143G variant *in cis* in the same allele). This is an ancestral founder allele (as described by us in (Hakonen et al. 2007; Hakonen et al. 2005) that has spread from an ancient European founder to different Western populations from USA to Europe and Australia. The heterozygous carrier frequencies are high in Finland and Norway, 1:84 and 1:100, respectively. The Finnish patients all carry the same founder allele as homozygous, thus sharing the identical chromosomal area around the gene much larger than the core POLG1 gene, including also its regulatory regions (Hakonen et al. 2005), and all other studied individuals globally who have the same

mutation (p.W748S) have also identical chromosomal region around the mutation, indicating that they originate from a single ancient founder dating back to Viking times. Therefore we call it the “MIRAS-allele”. The patient collection in Finland, with so many individuals carrying identical genetic background of disease, but still manifesting in various ways in different patients (referring to the three main types of MIRAS as described in the manuscript – juvenile-onset epilepsy, early-adult-onset ataxia-polyneuropathy-epilepsy or middle-age-onset parkinson’s) is genetically exceptionally homogenous and unique. ***These clinical and genetic data indicate that strong risk or protective factors contribute to disease manifestation.*** This is actually the original motivation of this study: to explore environmental / risk factors that induce MIRAS to manifest sometimes in teenagers as epilepsy and sometimes in middle-aged subjects with parkinson’s. We had emphasized this in the original manuscript but ***have further clarified the genetic homogeneity and the founder allele in the revised manuscript.***

Reviewer:

15) All western blots shall be quantified as in Fig 3A.

Authors: We have now quantified all western blots and tabulated each individual data point in box-and-whiskers plots, showing the median with interquartile range, and determined the statistical significance using the 2-tailed unpaired student’s t-test with their p-value indicated. The details are described in the figure legends in the revised manuscript.

Referee #2 (Remarks to the Author):

In this study, Kang et al. present evidence of dysregulated innate immune signaling in mitochondrial recessive ataxia syndrome (MIRAS). They found that MEFs from knock-in mice carrying a MIRAS patient pathogenic POLG1 allele display basal downregulation in immune-related gene expression and dampened activation of innate immune signaling pathways by nucleic acid agonists, which was recapitulated to a significant degree in fibroblasts from MIRAS patients (MIRAS-pt). Interestingly, infection of MIRAS-pt fibroblasts with TBEV or SARS-CoV2 revealed that cells harboring this POLG1 mutation had a dampened innate immune response following viral infection, which the authors postulate allows for the increased viral load observed in these cells. Transcriptomic analysis of MIRAS-pt brains also found suppression of innate immune signaling. The role of mitochondria in innate in adaptive immunity is an **exciting and active area**, yet the role of disruptions in this axis in mitochondrial diseases has not been addressed extensively to date. Thus, the **main observation here that innate immunity is dysregulated in MIRAS is of great interest to the field and potentially relevant to pathogenic mechanisms in certain mitochondrial diseases**. However, the study is largely descriptive and correlative, leaving the nature of the mitochondrial stress involved and the mechanism of immune system deregulation unclear. These and other issues diminish the impact of the study at this juncture.

We thank the Reviewer for the interest and very positive comments concerning our manuscript. We also appreciate the reviewer's constructive feedback to substantiate our findings and have addressed all the comments. For the mechanistic aspect, our extensive new data indicate the following sequence of events:

1) W748S mutation causes poor POLG1 contacts with other mtDNA replisome proteins, causing replisome instability and consequent decreased processivity; 2) lowered mtDNA replication activity results in mtDNA depletion; 3) decreased mtDNA replication and amount reduce the cellular ability to present mtDNA /RNA fragments to cytoplasm to activate viral sensors; 4) enhanced viral replication in the immunocompromised cellular environment hijacks dNTP pools; 5) rapid viral invasion together with acute mtDNA depletion and dysfunctional MIRAS replisome, cause progressive mtDNA depletion leading to cellular energy crisis, delayed strong proinflammatory response and reduced viability via necroptosis.

Our findings, including those from Finnish population, (FinnGen), indicate that POLG1-W748S variant is strongly associated with delayed/ decreased innate immunity. The evidence suggests that active mtDNA replication, instead of mtDNA fragmentation, is necessary for innate immunity response to viral infection in the brain and liver.

Our detailed responses are detailed below.

Major Points:

Reviewer:

1) It is not clear what the authors mean by “mtDNA replisome stress” other than simply mutations in POLG1, the replicative mtDNA polymerase. For example, the cells show no change in mtDNA copy number and other clear mitochondrial phenotypes are not elucidated, so the nature of the mtDNA/mitochondrial stress that the authors claim results in altered innate immune regulation remains a mystery.

Authors: We thank reviewer for pointing out this. We indeed had used the term for defective function of the DNA-polymerase gamma in the mtDNA replisome, which is the shared primary cause of all the aberrant cellular consequences we report. For clarity, as suggested by the reviewer, we have now

removed the term and included detailed description for MIRAS-POLG1 mutation in the revised text, as also explained below.

Please see our detailed response for comment 3-4 on the immunity mechanisms.

Reviewer:

2) The authors also do not provide a clear mechanism for the aberrant immune responses observed in MIRAS MEFs and MIRAS-pt fibroblasts. Although Fig. 3 is presented as an investigation into what mediates the signaling (line 198-199), no clear mechanism is elucidated or proposed. The authors do test immune responses under a variety of related condition to try an unravel this (e.g. knock-down of POLG1 and 2, expression of HSV-1 UL12.5). But these cause quite different innate immune responses than those observed in the patient fibroblasts making interpretations muddy. The data in Fig. 3 presents interesting findings concerning the regulation of POLG1 and 2 during viral infection, but does not really inform what is occurring in cells expressing the POLG1 MIRAS-allele or why in the MIRAS case there is a dampening of the immune response (Figure 1K and 2B-C). Have the authors considered that reduced release of mtDNA and/or mtRNA in the MIRAS mouse and human cells is the reason for the reduced innate immune signaling?

Authors: We thank the Reviewer for these constructive comments. In our manuscript, revised in a major manner according to the comments by the reviewers, we have expanded our human materials and focused in these as well as generated a novel *in vivo* model, the MIRAS knock-in mouse, here challenged with viruses. Further, human genomic databank FinnGen, linked with medical histories and prescription drug data of >342,000 subjects, showed that the specific MIRAS-W748S mutation carriers show highly significant enrichment of immunodeficient symptoms. Such enrichment was not present in a selected set of other mitochondrial disease genes. To reduce heterogeneity of our materials (MEFs vs human cells), we omitted the MEF data and now only report results of an increased amount of patient cells, patient materials and MIRAS-knockin mice, all homozygous for the same homologous MIRAS-mutation of POLG1. These models are detailed below.

Please see our detailed response for Comments 3+4.

Reviewer:

3) There is contradictory data throughout the manuscript that is difficult to interpret. For example, the MIRAS MEFs in Fig. 1 appear to have diminished POLG1/2 as well as diminished innate immune protein expression (e.g., MDA5, RIG-I), but POLG1/2 knockdown (Fig. 3) causes increased innate immune protein expression. How do the authors explain this? Also, POLG1 does not appear to be knocked down to significant levels in Fig. 3A. Finally, it is also not clear from the methods section what the negative control was for the experiment in in Fig. 3A. If an shRNA control (e.g. scrambled) was not used, the increases in RIG-I, MDA5, and ISGs might be explained simply by introduction of innate immune activating double-stranded RNA.

4) Similar to point #3, it is confusing that in Fig. 1, the response to poly(I:C) in patient cells is diminished at early time points and not changed at later time points, whereas in the MIRAS MEFs there is a modestly enhanced response at early time points and a heightened response at later time points. These inconsistent reponses in different cell types raise concerns about the role of POLG1 mutations per se in response to exogenous RNA, which is of additional concern in interpreting the data during viral infection.

Authors: Combined response for comments 3 and 4

We agree with the Reviewer and have now significantly revised the manuscript also according to the other reviewers comments. In the current paper, we emphasize models that carry homozygous MIRAS allele, instead of shRNA mediated silencing of POLG1 expression that may represent different signaling activation compared to patient mutations. With an overarching goal to understand triggers of human MIRAS disease manifestations, we were successful to sample more MIRAS patients and establish fibroblast cell lines with

gender and age-matched controls, extend the human cell materials to six and focused in their responses, together with other human materials and a novel knock-in mouse model that carries similar variants as the human POLG patients (human p.W748S+ neutral E1143G in cis in the POLG1 polypeptide – mouse protein changes W726S + E1121G). We show that the W748S mutation causes remarkable consequences both *in vivo* in mice and in human materials, with high relevance to actual disease.

In the revised manuscript we report robust findings of aberrant immunity of MIRAS patients in new systems (patient cell cultures, serum samples, MIRAS knockin mice), viral exposures as well as utilize population genetics of >342,000 individuals with genetic and medical data (FinnGen database):

- i) ***Dysregulated innate immune responses to viral mimetic and virus infections of primary MIRAS patient cells.*** We tested six biologically independent MIRAS patient primary fibroblast lines with their age and gender-matched controls, overall eliciting an attenuated initial IFN-1 signaling favouring viral replication with promoted cytotoxicity during viral infection. We subjected them to time course treatment of either **(1) immunogenic mimetics for viral-associated molecular patterns (dsDNA or dsRNA i.e. poly (I:C)) or (2) DNA or RNA virus infection (HSV-1, TBEV and SARS-CoV-2)**. These viruses were selected based on their ability to cause severe, delayed complications in a minority of infected patients, suggesting susceptibility factors predisposing to severe delayed symptoms. We demonstrate that innate immune signaling, especially the IFN-1 dependent pathways, are inhibited, including sequential timepoints as suggested by the reviewer. Furthermore, we show that the patient cells allow higher replication of the virus in the initial infection compared to the controls. This is true for all the tested viruses. These new results strengthen our results in the original submission. ***[New Data in revised Figure 1-2].***
- ii) ***Dysregulated immune profile, including IL-6 activation in MIRAS patients sera (n=15 patients) [Data in revised Figure 4J]***
- iii) ***Reduced POLG1 protein and its associated complex, as well as dampened IFN-1 signaling/virus defense gene expression profile in MIRAS patient cerebral cortex and cerebellum autopsy samples (n=3 patients) [Data in revised Figure 5]***
- iv) ***Population genetic and medical data from >342,000 subjects (FinnGen) indicate with high significance that specifically the p.W748S MIRAS variant carriers exhibit enriched immunodeficiency-linked morbidities.*** Similar association was not found in a set of other mitochondrial disease genes. ***[New data in revised Figure 1A]***
- v) ***Reduced basal level of POLG1 and mtDNA synthesis rate in the cerebral cortex and liver of MIRAS mice.*** The mice showed diminished release of protective antiviral IFN- α and IFN- β cytokines into the sera of MIRAS mice in early stage infection (p.i. day 4) and weakened IFN-1 signaling also in the cerebral cortex of MIRAS mice. Progressively during the infection, the pro-inflammatory IL-6 was released into their sera with an increasing trend. The liver showed remarkable sensitivity to TBEV infection. It was a surprise for us to find such hepatic sensitivity of MIRAS-mice to a neurotropic virus TBEV, but some previous reports have found liver inflammation associated with this virus (Bogovič et al. 2022; Misić-Majerus et al. 2005). While recognizing potential differences in species, the finding brings an interesting potential relevance to the mechanisms of the dramatic MIRAS-related valproate hepatotoxicity: a subclinical hepatic inflammation during the acute manifestation of MIRAS upon a viral infection would rapidly worsen when challenged with valproate and cause subacute liver failure. ***Our in vivo data indicate that MIRAS allele***

results in dysregulated immune response and decreased murine tolerance particularly to virus infection, most remarkable in the liver. [New data revised Figure 3-4].

All in all, these data in patient cells, autopsy samples, population data from 342,499 individuals as well as in vivo mouse findings together indicate aberrant immune and inflammatory responses in MIRAS. We propose that these events contribute to juvenile, valproate-sensitive MIRAS manifestation mimicking in severity and acuteness of viral-induced encephalitis (especially HSV-1 and TBEV).

Reviewer:

5) The effects of TBEV on RIG-I/MDA5/IFIT3 (Fig. 2B) and IL6 levels (Fig. 2D) different significantly in the two patients' cells, making the actual MIRAS phenotype unclear.

Authors: We agree with the reviewer and therefore were able to expand our patient cell materials and focus in these as described above in detail. We have included six biologically independent, age and gender matched control and MIRAS patient primary fibroblast lines for all viral mimetic and virus infection analyses strengthening our original findings *[Data in revised Figure 1-2]*. The different viruses (HSV-1, TBEV or SARS-CoV-2) showed enhanced replication in MIRAS cell lines compared to controls, and MIRAS cells showed increased pro-inflammatory NF- κ B transcription factor and necroptotic activating protein marker in the context of viral infections. Also, we got blood samples from additional MIRAS patients (n=15) and controls (n=23) indicating chronically increased IL-6 levels in MIRAS patient circulation.

Reviewer:

6) The authors state that HSV infection kills both their MIRAS-pt and control fibroblasts (lines 169-171) after 24 hours, yet show data from fibroblasts 48 hours after infection with HSV (Figure 3F).

Authors: We thank the reviewer for careful reading and in this revised manuscript, we have performed new analyses on our expanded panel of patient and control fibroblasts subjected to a time course of HSV-1 infection at 6, 24 and 48 hours of infection. Cellular viability of HSV-1 infected cells were also monitored and compared to mock-infected cells. In this time course with the used viral load, the viability was not significantly affected *[Data in revised Figure 2 and Extended Data Figure 5]*.

Reviewer:

7) More information on donor cells is needed. The authors may need to better control for sex or age to be able to make solid conclusions. For example, age has a huge impact on inflammation profiles/responses. Realizing this may be difficult due to the rarity of the disease, it remains hard to draw solid conclusion on such a small sample size.

Authors: We agree with the reviewer and indeed have made an effort to expand the biological number and match our MIRAS patient samples and the controls for both gender and age as explained above. In this revised manuscript, we have analysed six independent female patient derived fibroblasts with a mean age (SD) of 30 (\pm 9.7) for MIRAS patients, 33 (\pm 8.8) for the controls. As described by us and others (Hakonen et al. 2005; Neeve et al. 2012), females are more often manifesting early, severe form of MIRAS. *Details of the patients' age, gender and disease symptoms are now provided in the Extended Data Table 1 in the revised manuscript.*

Reviewer:

8) According to Fig. 3C, there is only a ~20% reduction of mtDNA after 48 hrs of UL12.5 expression. Based on other published studies, this should be more than enough time to completely degrade

mtDNA. There may be issues with the UL12.5 construct used or only a low percentage of transfected cells was achieved that might be confounding this experiment.

Authors: We agree with reviewer. As mentioned before, we succeeded to expand our materials and data to focus especially in the MIRAS-founder-allele and we argue that acute experimental depletion of POLG1/POLG2 or mtDNA via UL12.5 do not necessarily reflect MIRAS disease. We have now revised extensively the paper to focus on the MIRAS-allele and omitted the UL12.5 data.

Reviewer:

9) Throughout the manuscript, many of the western blots appear to contradict each other and/or do not appear to match the accompanying quantification (e.g., POLG1/2 levels are diminished in one experiment, but not the other in Fig. 1E, same for RIGI/MDA5/IFIT3 in Fig. 2B). In Figures 1E, 3B, Extended data 2B-C, the authors should provide a loading-control image, instead of just the quantification.

Authors: We apologize for the unclarity and have therefore now extensively revised the paper; quantified all western blots and provided the protein loading controls as suggested by reviewer. Changes in the levels of the proteins of interest were evaluated by normalisation against loading control protein of the sample: HSP60 (all but Figure 5A patient brain autopsies mitochondria – normalisation against loading control protein ATP5A; Figure 2A (human fibroblasts on viral infection time-course) and 4G (TBEV infected mouse liver) - normalisation against protein loading stain). Our extended patient materials strengthened the significance of analyses, each data point represents an independent biological individual fibroblast cell line and is tabulated in box-and-whisker plot showing median with IQR. Their respective p-values were calculated using the unpaired two-tailed student's t test to determine the statistical difference between the groups.

Minor Points:

Reviewer:

1) In Figure 2D, please indicate in the legend what the bars on the side of the graph are comparing.

Authors: We have updated the analyses as detailed above and have six biologically independent patient and control individual fibroblast lines, with description of the statistical comparison performed described in the figure legend.

Reviewer:

2) Please ensure that all extended data figure legends in which statistical analysis was performed contain p-value definitions.

Authors: We have updated all main and extended data figure legends with p-value definitions.

Reviewer:

3) In Fig. 2, the analysis should include control uninfected cells.

Authors: We have updated the analyses to include uninfected controls and a timecourse of HSV-1 infection [*Data in revised Figure 2A*].

Referee #3 (Remarks to the Author):

1. Reviewer:

In their manuscript “Mutant mitochondrial DNA polymerase impairs acute antiviral immunity” Kang et al investigate the innate immune status in patients, patient-derived cell-culture models and in vitro mouse models of Mitochondrial recessive ataxia syndrome (MIRAS). They report a decreased activation of the type I IFN system at steady state in all three systems. Although MIRAS is caused by mutations in POLG1 that lead to destabilization of the protein, the authors (confusingly) show that directly decreasing POLG1 levels (using shRNA) leads to the opposite outcome (increased IFN levels). The author’s observation of decreased IFN levels in MIRAS patients is noteworthy, because it is validated in primary patient samples, and is unexpected because shRNA perturbation and other POLG mutations (e.g. <https://www.biorxiv.org/content/10.1101/2020.09.22.308171v1.full.pdf>) show the opposite effect.

Authors: We thank the reviewer for the insightful comments. We have now extensively revised the manuscript according to this and other reviewers’ comments and focus in the actual human mutation carrying models and its conserved new knock-in mouse model. We would like to note that the POLG mutation in the Mutator mouse (the BiorXiv paper referred above) is not representing a mitochondrial disease. The POLG-exonuclease mutation of mtDNA mutator causes random mutagenesis of mtDNA, which has so far never been reported in actual patients. Secondly, we and others have shown that the Mutator’s exo-mutation increases mtDNA replication several fold (Hämäläinen et al. 2019; Macao et al. 2015) which shifts nucleotide pools in the whole cell challenging genomic DNA stability in stem cells. The p.W748S of POLG1 in this manuscript, the MIRAS mutation, is a human disease causing variant, and it causes reduced contacts with other replisome proteins, reduced replisome stability and lowered processivity compared to wild type mice (Euro et al. 2011; Euro et al. 2017). Therefore, the effects of the exo-mutant mtDNA mutator mouse are not relevant to the current findings with our models and patients with p.W748S patient mutation - they are even opposite for mtDNA synthesis (increased mtDNA in mutator vs decreased in POLG-p.W748S mice) and it is not a surprise that the effects to IFN-pathways are different.

2. However, the study is mainly descriptive and lacks mechanistic insights. Most critically, is not clear how the mutations in POLG1 result in suppression of the basal interferon response. Is there an IFN-inducing pathway defective in MIRAS-MEFs or is there an IFN-decreasing pathway over-activated in MIRAS-MEFs? Which pathway is that? Is the same pathway responsible for the phenotype seen in patients/brain of patients?

For the mechanistic aspect, our extensive new data indicate the following sequence of events:

1) W748S mutation causes poor POLG1 contacts with other mtDNA replisome proteins, causing replisome instability and consequent decreased processivity; 2) lowered mtDNA replication activity results in mtDNA depletion; 3) decreased mtDNA replication and amount reduce the cellular ability to present mtDNA /RNA fragments to cytoplasm to activate viral sensors; 4) enhanced viral replication in the immunocompromised cellular environment hijacks dNTP pools; 5) rapid viral invasion together with acute mtDNA depletion and dysfunctional MIRAS replisome, cause progressive mtDNA depletion leading to cellular energy crisis, delayed strong proinflammatory response and reduced viability via necroptosis.

Our findings, including those from Finnish population, (FinnGen), indicate that POLG1-W748S variant is strongly associated with delayed/ decreased innate immunity. The evidence suggests that active mtDNA replication, instead of mtDNA fragmentation, is necessary for innate immunity response to viral infection in the brain and liver.

In the revised paper, we concentrated in generating more models with MIRAS allele, relevant for disease, and omitted the data from POLG shRNAs. We succeeded to get consents from more MIRAS patients to give a sample for fibroblasts, and - importantly - we engineered a knock-in mouse carrying the MIRAS allele with variants homologous to the human allele. *Our intriguing conclusion based on our data in cells and in vivo in mice is indeed that reduced mtDNA replication by MIRAS-POLG1 causes delayed activation of innate immunity allowing increased viral replication in the early infection, compromising cellular/tissue tolerance to viral infection.* Therefore, POLG1 / active mtDNA replication is required for IFN-1 pathway activation, a mechanism being previously connected to mtDNA or mtRNA damage in experimental models but not in mitochondrial diseases. The MIRAS-POLG1 sensitivity to viral infections is true for the three different tested viruses (new data) and *in vivo*, in MIRAS mice infected with TBEV, all data pointing to the increased viral replication in the early infection and aberrant activation of innate immunity.

The MIRAS mice show remarkable sensitivity to TBEV infection. Previous experimental models that decrease mtDNA packaging (TFAM knockdown) were shown to induce viral sensors ((West et al. 2015) and others). Our data is the first that directly addresses the responses in both cellular and *in vivo* models carrying a patient mutation. All our patients carry the same MIRAS allele that is originating from a single ancient European founder and has spread to the Western world (as we reported in (Hakonen et al. 2007; Hakonen et al. 2005). Therefore, the patient material is genetically exceptionally homogeneous, and an optimal one to make conclusions of extrinsic modifiers in the disease manifestation.

In the revised manuscript we report robust findings of aberrant immunity of MIRAS patients in new systems (patient cell cultures, serum samples, MIRAS knockin mice), viral exposures as well as utilize population genetics of >342,000 individuals with genetic and medical data (FinnGen database):

- i) ***Dysregulated innate immune responses to viral mimetic and virus infections of primary MIRAS patient cells.*** We tested six biologically independent MIRAS patient primary fibroblast lines with their age and gender-matched controls, overall eliciting an attenuated initial IFN-1 signaling favouring viral replication with promoted cytotoxicity during viral infection. We subjected them to time course treatment of either **(1) immunogenic mimetics for viral-associated molecular patterns (dsDNA or dsRNA i.e. poly (I:C)) or (2) DNA or RNA virus infection (HSV-1, TBEV and SARS-CoV-2)**. These viruses were selected based on their ability to cause severe, delayed complications in a minority of infected patients, suggesting susceptibility factors predisposing to severe delayed symptoms. We demonstrate that innate immune signaling, especially the IFN-1 dependent pathways, are inhibited, including sequential timepoints as suggested by the reviewer. Furthermore, we show that the patient cells allow higher replication of the virus in the initial infection compared to the controls. This is true for all the tested viruses. These new results strengthen our results in the original submission. ***[New Data in revised Figure 1-2].***
- ii) ***Dysregulated immune profile, including IL-6 activation in MIRAS patients sera (n=15 patients) [Data in revised Figure 4J]***
- iii) ***Reduced POLG1 protein and its associated complex, as well as dampened IFN-1 signaling/virus defense gene expression profile in MIRAS patient cerebral cortex and cerebellum autopsy samples (n=3 patients) [Data in revised Figure 5]***
- iv) ***Population genetic and medical data from >342,000 subjects (FinnGen) indicate with high significance that specifically the p.W748S MIRAS variant carriers exhibit enriched***

immunodeficiency-linked morbidities. Similar association was not found in a set of other mitochondrial disease genes. **[New data in revised Figure 1A]**

- v) ***Reduced basal level of POLG1 and mtDNA synthesis rate in the cerebral cortex and liver of MIRAS mice.*** The mice showed diminished release of protective antiviral IFN- α and IFN- β cytokines into the sera of MIRAS mice in early stage infection (p.i. day 4) and weakened IFN-1 signaling also in the cerebral cortex of MIRAS mice. Progressively during the infection, the pro-inflammatory IL-6 was released into their sera with an increasing trend. The liver showed remarkable sensitivity to TBEV infection. It was a surprise for us to find such hepatic sensitivity of MIRAS-mice to a neurotropic virus TBEV, but some previous reports have found liver inflammation associated with this virus (Bogovič et al. 2022; Misić-Majerus et al. 2005). While recognizing potential differences in species, the finding brings an interesting potential relevance to the mechanisms of the dramatic MIRAS-related valproate hepatotoxicity: a subclinical hepatic inflammation during the acute manifestation of MIRAS upon a viral infection would rapidly worsen when challenged with valproate and cause subacute liver failure. ***Our in vivo data indicate that MIRAS allele results in dysregulated immune response and decreased murine tolerance particularly to virus infection, most remarkable in the liver. [New data revised Figure 3-4].***

All in all, these data in patient cells, autopsy samples, population data from 342,499 individuals as well as in vivo mouse findings together indicate aberrant immune and inflammatory responses in MIRAS. We propose that these events contribute to juvenile, valproate-sensitive MIRAS manifestation mimicking in severity and acuteness of viral-induced encephalitis (especially HSV-1 and TBEV).

3. Reviewer:

The authors overall model is that lower IFN levels in MIRAS patients lead to increased susceptibility/pathology to virus infection. Subsequent over-activation of the immune system or increased viral load is then proposed to promote disease. Except for the lower levels of IFN in MIRAS MEFs and brain samples, there is no solid evidence for this model. In fact, the authors find decreased IFN levels in MIRAS patient brains, so their idea of increased immune activation or viral infection as the driver of MIRAS disease is seemingly inconsistent with their data.

Authors: In our revised paper, we'd like to highlight that ***our data indicate sensitivity of MIRAS patient cell lines and MIRAS-knock-in mice to viral infection, and the immune response particularly the IFN-1 is activated in a delayed manner whereas the inflammatory response was over-activated. [Revised Figure 1-4].***

4. Reviewer:

The authors claim that "Our evidence suggests that viral infections contribute to mitochondrial disease manifestations in patients with genetic predisposition" (line 33ff), "that the [...] immune overactivation and toxic cytokine induction contributes to nervous system pathology in MIRAS" (line 192ff) and that "our findings suggest [...] delayed and augmented inflammatory reaction to promote disease manifestation" (line 269ff). However none of these speculative claims is directly supported by data in the manuscript.

Authors: ***The new experiments have considerably strengthened the conclusion.*** Previously, the W748S mutant POLG1 has been structurally modelled and analysed by atomistic molecular dynamics simulation to affect replisome protein interactions (Euro et al. 2011; Euro et al. 2017). Furthermore, increased mtDNA replisome activity was shown to affect cellular nucleotide pools, especially their prioritization to mitochondria vs. nucleus (Hämäläinen et al. 2019). These data suggest that W748S POLG1 mutant replicase

would lead to instable mtDNA replisome, decreased processivity and lowered nucleotide pool prioritization for mitochondria. Indeed, our data support that suggestion:

We show that the brain, liver (the major organs affected in MIRAS disease) and immune-active organ spleen, all had remarkably diminished amount of POLG1 protein and modest decrease in POLG2 in MIRAS mice (Figure 3A). We also show the depletion of POLG1 protein and its associated native replisome complex in MIRAS patient brain autopsy (cerebral cortex and cerebellum) samples (Figure 5A). These data, together with the basal mtDNA decrease in the MIRAS mouse liver (Figure 3B) suggest decreased mtDNA replication activity / processivity. This we show to be true: *in vivo* mtDNA replication is lowered in MIRAS mice – BrdU labelling of nascent mtDNA compared to total mtDNA is decreased in MIRAS liver and brain (Figure 3B). TBEV replication causes rapid mtDNA depletion in MIRAS mouse brain (Figure 4D), supporting lowered availability of nucleotides for mtDNA replication in MIRAS after activation of viral replication. While it remains possible that TBEV has a direct effect on mtDNA, similar to HSV-1, we consider the nucleotide pool contribution likely: the metabolomic analyses of the viral infected MIRAS mouse brains revealed significantly lowered nucleotides, with pyrimidine deoxyribonucleotides and pyrimidine ribonucleotide interconversion being the two most significantly changed pathways (Figure 4A, B, C). Also MIRAS patient fibroblasts showed low steady-state POLG1 protein and accelerated mtDNA depletion when infected with HSV-1.

The data suggest a sequence of pathogenesis: 1) W748S mutation causes poor POLG1 contacts with other mtDNA replisome proteins, causing replisome instability and consequent decreased processivity; 2) lowered mtDNA replication activity which together with replisome dysfunction causes mtDNA depletion; 3) virus hijacks dNTP pools to serve its replication, which especially cripples the dysfunctional MIRAS replisome, causing progressive mtDNA depletion and reduced viability.

However, because POLG1-W748S variant is strongly associated with delayed/ decreased innate immunity (our experimental data and human population data - FinnGen), we propose that POLG replication activity is required for proper innate immunity response induction in the brain and liver, as a response to viral infection.

5. Reviewer:

In fact, while the authors claim that there is an initially delay in interferon-stimulated gene expression in MIRAS MEFs, it is not clear this is really the case. For example, the panels “untreated” and “10h dsDNA” in Fig 1 K look exactly the same, indicating that while MIRAS-MEFs start at a lower level of ISG15, IFNB1 and IL1B, their induction is comparable to WT cells. This would be easier to discern if the samples were normalized to untreated rather than to wildtype at each timepoint.

Authors: To minimize species-heterogeneity, we decided to concentrate in the patient cell lines only in this manuscript, as we were able to obtain more of them, and removed the MEF-data. As suggested, we have now normalized the infected/treated samples of human cells and of the knock-in-mice to their baseline controls.

6. Reviewer:

Overall, while there are some interesting preliminary observations here that appear to link POLG1 to innate immunity, there is no mechanistic understanding of what is happening either in cells *in vitro* or in patients.

Authors: We thank the reviewer for constructive criticism. We refer to the detailed explanation to point 4 above.

Other comments

- Why are IFN levels higher in MIRAS MEFs at later points of PAMP stimulation?

Authors: Based on the reviewer and other reviewers' comments and to account for human vs. mouse and cell-type specific immune response, we have significantly revised the manuscript, we made an effort to generate more MIRAS patients-derived primary fibroblasts, expanded the panel to six biological individual derived lines with age and gender matched and now concentrate in human materials. We wish that the reviewer is satisfied by our quite extensive new data focusing on MIRAS patients primary cells and novel knock-in-mouse and *in vivo* infection data. All our models currently carry the homozygous mutation c.2243G>C (p.W748S; coinciding with neutral p.E1143G variant in cis in the same allele) or the homologous conserved site in mouse POLG1, all giving insights for MIRAS disease. This we find a considerable strength of the paper. We have therefore removed the MEF data to improve clarity of the revised manuscript.

- Lack of controls, e.g. Fig 2A, B lack uninfected condition

Authors: We have now performed new viral infection of our human cells and include the uninfected control.

- Why is there no band for POLG2 in lane 2 of Fig3E?

Authors: As mentioned before, we succeeded to expand our materials and data to focus especially in the MIRAS-founder-allele and we argue that acute experimental depletion of POLG1/POLG2 via UL12.5 do not necessarily reflect MIRAS disease. We have now revised extensively the paper to focus on the MIRAS-allele and omitted the UL12.5 data in the previous Figure 3E.

- Can the patient fibroblasts be complemented with WT POLG1 to prove that the phenotype of these fibroblasts is indeed due to the POLG1 mutation

Authors: We succeeded to expand our patient materials and generated mouse model carrying homologous patient mutation. We argue that exogenous complementation of WT POLG1 into the patient fibroblasts do not necessarily reflect MIRAS disease and that our patient, mouse and cell data strongly indicate that MIRAS-mutation modifies innate immunity.

Referee #4 (Remarks to the Author):

Kang Y. et al. explored how recessive conserved POLG1 mutations that originated in Nord-European populations, and which are linked to MIRAS etiology, affect the innate immune response to viral infections. Authors report that either murine or human cells harboring the MIRAS-linked POLG1 mutations display an aberrant innate-immune response to viral infection with a delayed antiviral transcriptional response that allows increased viral infection and then, an exacerbated production of inflammatory cytokines. The authors suggest these immune defects contribute to the development of the patient neurological pathologies.

1. After evaluating transcriptomes of autopsy-derived brain tissue from control, MIRAS and MELAS patients, the authors conclude that MIRAS pathology is related to a downregulation of immunity-related genes and that this is a specific hallmark of POLG1-related MIRAS and not a common feature in mitochondrial neurodegenerative diseases. However, MELAS may be the exception and, as this is quite interesting, this point would be strengthened by analyzing other mitochondria protein mutation linked diseases.

Authors: We unfortunately don't have access to autopsies from many mitochondrial patients. We removed MELAS because of its limited information content, but have now redone all cell culture data in human cells and included a novel *in vivo* model in mice infected with virus [**Revised Figure 1-4**]. We also compared the genetic and medical data from population databank with >342,000 subjects and detected a strong correlation of MIRAS carriers to immunodeficiency trait that is not enriched in a set of other mitochondrial disease gene variant carriers [**Revised Figure 1A, Extended Data Figure 1A**]. The data strongly support deficient immune response in MIRAS patients.

2. The manuscript shows novel and interesting results linking POLG1 mutations to immune responses and viral infection, which may have clinical relevance. However, authors do not address any mechanisms underlying the relationship between POLG1 and the antiviral response or how immune dysregulation may be linked to neurodegenerative outcomes.

Authors: We agree with the reviewer, and have now improved the mechanistic aspect with the new *in vivo* data.

For the mechanistic aspect, our extensive new data indicate the following sequence of events:

1) W748S mutation causes poor POLG1 contacts with other mtDNA replisome proteins, causing replisome instability and consequent decreased processivity; 2) lowered mtDNA replication activity results in mtDNA depletion; 3) decreased mtDNA replication and amount reduce the cellular ability to present mtDNA /RNA fragments to cytoplasm to activate viral sensors; 4) enhanced viral replication in the immunocompromised cellular environment hijacks dNTP pools; 5) rapid viral invasion together with acute mtDNA depletion and dysfunctional MIRAS replisome, cause progressive mtDNA depletion leading to cellular energy crisis, delayed strong proinflammatory response and reduced viability via necroptosis.

Our findings, including those from Finnish population, (FinnGen), indicate that POLG1-W748S variant is strongly associated with delayed/ decreased innate immunity. The evidence suggests that active mtDNA replication, instead of mtDNA fragmentation, is necessary for innate immunity response to viral infection in the brain and liver.

In detail:

We generated a *novel MIRAS mouse model carrying the homologous conserved POLG1 disease variant in the MIRAS patients (human p.W748S + neutral E1143G in cis in the POLG1 polypeptide – mouse protein changes W726S + E1121G).*

What are the mitochondrial mechanisms? Previously, the W748S mutant POLG1 has been structurally modelled and analysed by atomistic molecular dynamics simulation to affect replisome protein interactions (Euro et al. 2011; Euro et al. 2017). Furthermore, increased mtDNA replisome activity was shown to affect cellular nucleotide pools, especially their prioritization to mitochondria vs. nucleus (Hämäläinen et al. 2019). These data suggest that W748S mutant replicase would lead to instable mtDNA replisome, decreased processivity and lowered nucleotide pool prioritization for mitochondria.

Indeed, our data support that suggestion:

We show that the brain, liver (the major organs affected in MIRAS disease) and immune-active organ spleen, all had remarkably diminished amount of POLG1 protein and modest decrease in POLG2 in MIRAS mice (Figure 3A). We also show the depletion of POLG1 protein and its associated native replisome complex in MIRAS patient brain autopsy (cerebral cortex and cerebellum) samples (Figure 5A). These data, together with the basal mtDNA decrease in the MIRAS mouse liver (Figure 3B) suggest decreased mtDNA replication activity / processivity. This we show to be true: *in vivo* mtDNA replication is lowered in MIRAS mice – BrdU labelling of nascent mtDNA compared to total mtDNA is decreased in MIRAS liver and brain (Figure 3B). TBEV replication causes rapid mtDNA depletion in MIRAS mouse brain (Figure 4D), supporting lowered availability of nucleotides for mtDNA replication in MIRAS after activation of viral replication. While it remains possible that TBEV has a direct effect on mtDNA, similar to HSV-1, we consider the nucleotide pool contribution likely: the metabolomic analyses of the viral infected MIRAS mouse brains revealed significantly lowered nucleotides, with pyrimidine deoxyribonucleotides and pyrimidine ribonucleotide interconversion being the two most significantly changed pathways (Figure 4A, B, C). Also MIRAS patient fibroblasts showed low steady-state POLG1 protein and accelerated mtDNA depletion when infected with HSV-1.

The data suggest a sequence of pathogenesis: 1) W748S mutation causes poor POLG1 contacts with other mtDNA replisome proteins, causing replisome instability and consequent decreased processivity; 2) lowered mtDNA replication activity which together with replisome dysfunction causes mtDNA depletion; 3) virus hijacks dNTP pools to serve its replication, which especially cripples the dysfunctional MIRAS replisome, causing progressive mtDNA depletion and reduced viability.

However, because POLG1-W748S variant is strongly associated with delayed/ decreased innate immunity (our experimental data and human population data - FinnGen), we propose that POLG replication activity is required for proper innate immunity response induction in the brain and liver, as a response to viral infection.

In the revised manuscript we report robust findings of aberrant immunity of MIRAS patients in new systems (patient cell cultures, serum samples, MIRAS knockin mice), viral exposures as well as utilize population genetics of >342,000 individuals with genetic and medical data (FinnGen database) as summarized here:

- i) ***Dysregulated innate immune responses to viral mimetic and virus infections of primary MIRAS patient cells.*** We tested six biologically independent MIRAS patient primary fibroblast lines with their age and gender-matched controls, overall eliciting an attenuated initial IFN-1 signaling favouring viral replication with promoted cytotoxicity during viral infection. We subjected them to time course treatment of either **(1) immunogenic mimetics for viral-associated molecular patterns (dsDNA or dsRNA i.e. poly (I:C))** or **(2) DNA or RNA virus infection (HSV-1, TBEV and SARS-CoV-2)**. These viruses were selected based on their ability to cause severe, delayed complications in a minority of infected patients, suggesting susceptibility factors predisposing to severe delayed symptoms. We demonstrate that innate immune signaling, especially the IFN-1 dependent pathways, are inhibited, including sequential timepoints as suggested by the reviewer. Furthermore, we show that the patient cells allow higher replication of the virus

in the initial infection compared to the controls. This is true for all the tested viruses. These new results strengthen our results in the original submission. **[New Data in revised Figure 1-2].**

- ii) ***Dysregulated immune profile, including IL-6 activation in MIRAS patients sera (n=15 patients) [Data in revised Figure 4J]***
- iii) ***Reduced POLG1 protein and its associated complex, as well as dampened IFN-1 signaling/virus defense gene expression profile in MIRAS patient cerebral cortex and cerebellum autopsy samples (n=3 patients) [Data in revised Figure 5]***
- iv) ***Population genetic and medical data from >342,000 subjects (FinnGen) indicate with high significance that specifically the p.W748S MIRAS variant carriers exhibit enriched immunodeficiency-linked morbidities.*** Similar association was not found in a set of other mitochondrial disease genes. ***[New data in revised Figure 1A]***
- v) ***Reduced basal level of POLG1 and mtDNA synthesis rate in the cerebral cortex and liver of MIRAS mice.*** The mice showed diminished release of protective antiviral IFN- α and IFN- β cytokines into the sera of MIRAS mice in early stage infection (p.i. day 4) and weakened IFN-1 signaling also in the cerebral cortex of MIRAS mice. Progressively during the infection, the pro-inflammatory IL-6 was released into their sera with an increasing trend. The liver showed remarkable sensitivity to TBEV infection. It was a surprise for us to find such hepatic sensitivity of MIRAS-mice to a neurotropic virus TBEV, but some previous reports have found liver inflammation associated with this virus (Bogovič et al. 2022; Misić-Majerus et al. 2005). While recognizing potential differences in species, the finding brings an interesting potential relevance to the mechanisms of the dramatic MIRAS-related valproate hepatotoxicity: a subclinical hepatic inflammation during the acute manifestation of MIRAS upon a viral infection would rapidly worsen when challenged with valproate and cause subacute liver failure. ***Our in vivo data indicate that MIRAS allele results in dysregulated immune response and decreased murine tolerance particularly to virus infection, most remarkable in the liver. [New data revised Figure 3-4].***

All in all, these data in patient cells, autopsy samples, population data from 342,499 individuals as well as in vivo mouse findings together indicate aberrant immune and inflammatory responses in MIRAS. We propose that these events contribute to juvenile, valproate-sensitive MIRAS manifestation mimicking in severity and acuteness of viral-induced encephalitis (especially HSV-1 and TBEV).

Specific points:

3. Reviewer:

1) The knock-in mutant POLG1 mouse generated for this study is a very valuable MIRAS model. However, information about the phenotypes of the mutant animals is missing. Do the W726S/E1121G POLG1 mutant mice display any MIRAS related phenotypes?

Authors: We agree with the Reviewer and have now performed detailed studies of our *in vivo* MIRAS mouse model at baseline status and following exposure to viral infection ***[Data of our MIRAS mouse model is in revised Figure 3-4 and Extended Data Figure 7-11].*** These mice are born in normal Mendelian proportions and have a normal lifespan. We performed behavioral analyses of our MIRAS mouse - they develop mild motor symptoms after 12 months of age. Here, we report POLG1-W748S effects to replisome stability, mtDNA maintenance and immune response activation in liver and brain (the two major tissues impacted in MIRAS patients) and spleen at baseline and following viral infection

at day 1, 4 and 9 post infection. The mechanistic sequence is described in detail in point 2 above. Intriguingly, MIRAS mouse began to exhibit more severe phenotype, showing reduced tissue tolerance when challenged with viruses compared to the parallelly infected control mice. We propose that the viral trigger provides an explanation to the dramatic encephalitic disease course of the teenagers with the W748S-MIRAS variant accompanied by a preceding minor viral infection. **[More details are in the response below]**

Reviewer:

How is the immune response to viral infection in these animals? Does viral infection alter any phenotypes in the mice? Such experiments would get to the heart of the authors conclusions related to MIRAS disease mechanism.

Authors: We thank reviewer for the suggestion. We indeed now did an *in vivo* MIRAS mouse study with viral infection and showed sensitivity of MIRAS mice to TBE-virus. **[Data in revised Figure 3-4]**

We chose to infect the mice with TBE virus as the infection causes mild infection to most people, but in a minority of patients a biphasic severe disease that mimicks our patients disease course of onset (text described in revised manuscript). Also, TBEV infects mice well. Diminished amount of protective antiviral IFN- α and IFN- β cytokines were released into the sera of MIRAS mice in early stage infection (p.i. day 4) and weakened IFN-1 signaling also in the cerebral cortex and spleen of MIRAS mice. Concomitant metabolomic analyses of TBEV infected MIRAS mouse brain revealed significant nucleotide depletion which is paralleled by aggravated depletion of mtDNA compared to the infected control mice, indicating a role for MIRAS-POLG1 mediated mtDNA maintenance in response to acute viral infection. Progressively during the infection, the pro-inflammatory IL-6 was released into MIRAS mouse sera with an increasing trend. Although we were unable to detect convincing increase of the viral protein load in the brain or the liver during TBEV infection course, the liver showed remarkable sensitivity to TBEV infection. While it was a surprise to us to find such hepatic sensitivity of MIRAS-mice to viral-induced inflammation as a consequenc of neurotropic virus TBEV, previous reports have found liver inflammation associated with this virus (Bogovič et al. 2022; Misić-Majerus et al. 2005). While recognizing potential differences in species, the finding brings an interesting potential explanation to the dramatic MIRAS-related valproate hepatotoxicity: a viral infection causing a subclinical hepatitis during the manifestation of MIRAS would rapidly worsen when challenged with valproate and cause subacute liver failure. All in all, our *in vivo* data indicates that MIRAS allele results in dysregulated IFN-1 and IL-6 immune response and decreased murine tolerance particularly to viral infection, most remarkable in the liver. The experimental findings were re-inforced by our MIRAS patients blood samples: the serum cytokine profiles showed elevated pro-inflammatory IL-6 (n=15 patients) compared to healthy individuals (n=23) **[Data in revised Figure 4J]** and dampened IFN-1 and virus defense genes expression profile in cerebral cortex of patients' autopsies **[Data in revised Figure 5]**.

These *in vivo* mouse findings and the patient data together indicate aberrant immune and inflammatory responses in MIRAS. We propose that these events amplify virus-infection-induced tissue damage and contribute to juvenile, valproate-sensitive MIRAS manifestation.

Reviewer:

2) Information about the overall consequences of the POLG1 mutations on transcription is missing. Which is the total number of transcriptionally altered genes (both upregulated and downregulated)? What proportion of these up/down regulated genes are immune-related? Volcano plots obtained from the statistical analysis of the transcriptomic results are missing. Highlighting the genes that are displayed in Fig.1B in the respective volcano plot would provide visual insight into the proportion of total genes displaying a different transcription pattern that are immune-related.

Authors: We thank reviewer for the suggestion. We have now generated volcano plots for all transcriptomic datasets in the revised manuscript, and indicated the immune-related or other relevant pathway genes on the plots. Similar representation is now also utilised for the new metabolomic data in the revised manuscript. **[Data in revised Figure 3E, 4C, and 5B]**

Reviewer:

3) The protocol used to generate the gene ontology enrichment analyses is not mentioned in the methods section.

Authors: We have now included the details for pathway/functional enrichment analyses in the revised manuscript. **[Details in revised method section]**

Reviewer:

4) Figure 1C and D display only a very small set of genes and pathways - how representative are the changes in those categories compared to all the impaired biological processes? Adding a bubble plot of all the enriched GO categories that were altered due to the POLG1 mutations would ease the interpretation of the transcriptomics and proteomics experiments of the manuscript. This type of plot gives information on the GO enrichment score, p-value, and number of genes in each GO category, allowing a more complete picture.

Authors: We thank reviewer for the suggestion. And per reviewer's suggestion, we utilised now bubble plot to illustrate the pathways/functions enriched with the 'omic' datasets, showing their p-value and the pathway gene/metabolite number or activation z-score **[Data in revised Figure 3F, 4A and 5C]**. For the patients' cerebral cortex transcriptome analyses, the bubble plot pathway representation illustrates a significant dampening of IFN-1 and virus defense pathways, and neatly demonstrates the opposed regulation of these pathways in the cerebral cortex of mice acutely infected with TBEV **[Data in revised Figure 5C-D; Extended Data Figure 14A]**. The data support overall attenuated anti-viral immune transcriptomic landscape in MIRAS patient cerebral cortex and we thank the reviewer for this constructive suggestion.

Reviewer:

5) Authors indicate that cells harboring the studied POLG1 mutations have a delayed and then overactive immune response, but the transcript levels of the cytokines evaluated are normalized and plotted independently at each infection time and not as a time-lapse, which does not allow a comparison of the magnitude of the expression changes between the short and long infections (Figures 1G, H, I, K and 2A). Also, it is not discussed why authors chose to analyze a specific set of genes. Was this decision derived from their transcriptomic/proteomic data? And why different sets of genes were used in the different times and experiments?

Authors: As suggested by the reviewer, we have now analysed the response from our expanded biological patient fibroblast lines and our *in vivo* mouse model on a time course assay challenged with viral mimetic and/or viral infection. Also, we have included RNAseq analyses of our knock-in MIRAS mice and controls before and after challenged by TBEV infection and new metabolomics analyses to get a broad view on the responses.

In the revised manuscript, we have analysed a conserved set of gene and protein players in our human and mouse models focussing on the antiviral IFN-1 response in the early stage of viral mimetic or viral infection exposure, and the inflammatory response to trace for aberrant activation and potential cytotoxicity. The rationale behind this selection is included in the revised text and also described here: For study on initial innate immune activation of MIRAS models in response to viral mimetic, we focused on IFN-1 signaling because IFN-1 response (i) is known to be the first antiviral responder, (ii)

has recently reported to be inducible by mtDNA replisome protein defect or mtDNA breaks (Tigano et al. 2021; West et al. 2015), and (iii) was reported in mtDNA mutator mice (Lei et al. 2021).

Here, we detected weakened IFN-1 signaling activation in MIRAS. Because dampened early IFN-1 response during viral infection is known to elicit a secondary aberrant pro-inflammatory responses, particularly the NF- κ B signalling, we investigated if pro-inflammatory response was induced in MIRAS fibroblasts after prolonged viral mimetic treatment. Indeed, we found increased level of pro-inflammatory NF- κ B transcription factor after prolonged treatment with the viral mimetics and increased necroptotic activating protein marker **[New data in revised Figure 1 and the explanation has now been included in revised text]**.

However, IFN-1 signaling is known to be modified distinctly in response to different viruses. While some variation (including the IFN-1 signaling activation) – not surprisingly – was present because of the distinct invasion/replication strategies of these viruses and host responses in our fibroblast lines, we did find aberrant immune responses to all these viruses. The MIRAS patient fibroblasts showed an increase of pro-inflammatory NF- κ B transcription factor protein and necroptotic activating protein marker upon prolonged virus infection of HSV-1, TBEV or SARS-CoV-2 infections, all showing an increased viral load in MIRAS cells **[New data in revised Figure 2 and the explanation has now been included in revised text]**.

We further extended our studies to *in vivo* mouse model and examined their IFN-1 signaling activation at different days post TBEV infection in different mouse tissues particularly focusing on brain and liver – the two major tissues affected in MIRAS disease course (more mouse details are in the response to reviewer's comment 1 above). We found at 4 days post infection in MIRAS mice, a dampened release of IFN-1 cytokines into circulation and reduced IFN-1 signaling gene expression in the brain and spleen while their livers exhibit increased inflammation which exacerbated as infection prolonged. MIRAS mouse sera IL-6 level increased more as infection prolonged, and we found higher IL-6 level in MIRAS patients' circulation compared to healthy subjects. **[New data in revised Figure 3-4]**

The revised data overall support increased initial viral replication and weakened cellular IFN-1 response, followed by overactivated pro-inflammatory response pointing to sensitivity of MIRAS allele carriers to viral infection-induced disease. For the fundamental mitochondrial sequence of pathology, please see the detailed response above to point 2.

Reviewer:

6) Immunoblot presented in the extended figure 3A and the quantification of this in Figure 1J shows that ~50% of the fibroblasts from control patients display a decreased expression of POLG2 and of the immune-related genes evaluated. A violin plot would yield a more clear visualization of the data.

Authors: Based on this and other reviewers' comments, we have revised our manuscript and analysed our new patient fibroblast panel focusing on the six age and gender matched MIRAS and control individual fibroblast lines. Following another reviewer's suggestion, we presented the data in box and whisker plots showing median with IQR. Each individual data points were tabulated to show the relative distribution of each biological replicates and the statistical significance between the MIRAS and control group were examined using unpaired 2-tailed student's t test. We hope that the reviewer is happy with the revised data representation.

Reviewer:

7) In extended Fig. 3A the authors analyze 7 control and 5 MIRAS patients cell lines. Authors analyzed

further only two of the patient-derived fibroblasts lines for the control and MIRAS groups. There is no information regarding which two and how these were chosen. Please explain.

Authors: We apologise the unclarity and as detailed in the responses above, to account for inter-individual immune variation, we obtained consents for fibroblast lines from **more MIRAS patients, all homozygous for the ancestral gene variant leading to p.W748S + neutral E1143G amino acid change**, and established their primary fibroblasts cultures. We have now expanded our analyses to **six biologically independent, age and gender-matched MIRAS patient and control individuals-derived fibroblasts** and examined their responses to different viral mimetics and viral infection. **[Data in revised Figure 1-2; details of subjects is included in Extended Data Table 1]**

Reviewer:

8) There is an almost six-fold increase in the expression of IL-1 β observed in mutant MEFs after 24hr of dsRNA transfection, whereas patient derived fibroblasts present the opposite behavior upon the same treatment for the same time. This discrepancy in Fig 1I and Fig 1K between MEFs and patient fibroblast should be mentioned in the text.

Authors: As explained above, we removed the MEF data to improve clarity of the revised manuscript.

Reviewer:

9) In Figure 3, authors address the consequences of expressing the HSV-1 protein UL12.5. These data show that the POLG2 and innate immune gene expression is altered by HSV-1 but does not give any mechanistic insight on the role of POLG1 in the viral innate immune response, nor on how the W726S/E1121G POLG1 mutations are related to the immunosuppression phenotype. Is Fig. 3E labeled correctly? Why is there so little POLG2 in untreated cells in lane 2? And the large increase upon treatment with MG132 in lane 1 is surprising.

Authors: We agree with reviewer. As mentioned before, we succeeded to expand our materials and data to focus especially in the MIRAS-founder-allele and we argue that acute experimental depletion of POLG/POLG2 via UL12.5 do not necessarily reflect MIRAS disease. We have now revised extensively the paper to focus on the MIRAS-allele and omitted the UL12.5 data. For the fundamental mitochondrial sequence of MIRAS W748S pathology, please see the detailed response above to point 2.

Reviewer:

10) Transcriptomic analysis of MIRAS autopsy brains shows changes in immune signaling pathways and translation machinery. It is not clear which of these, if either, is linked to disease phenotypes. As the authors suggest that virus infection may be exacerbated in MIRAS patients, evidence of increased virus gene expression in the autopsy brain would be interesting to assess.

Authors: We agree with reviewer. We included now the *in vivo* data of MIRAS mice exposing to viral infection (more mouse details are in the response to reviewer's comment 1 and 2 above), which is directly addressing this question. Pathogen analysis of post-mortem brain was not available for us to analyse.

More minor comments

Reviewer:

1) The manuscript lacks information about the patient-derived fibroblasts and autopsy samples. Did authors analyze if the chosen subjects had the W726S/E1121G POLG1 mutations? More details are needed.

Authors: Every patient in this study carries the identical homozygous mutation c.2243G>C (p.W748S; coinciding with neutral p.E1143G variant in cis in the same allele). This is an ancestral founder allele (as described by us in (Hakonen et al. 2007; Hakonen et al. 2005) that has spread from an ancient European founder to different Western populations from USA to Europe and Australia. The heterozygous carrier frequencies are high in Finland and Norway, 1:84 and 1:100, respectively. The Finnish patients all carry the same founder allele as homozygous, thus sharing the identical chromosomal area around the gene much larger than the core POLG1 gene, including also its regulatory regions (Hakonen et al. 2005), and all other studied individuals globally who have the same mutation (p.W748S) have also identical chromosomal region around the mutation, indicating that they originate from a single ancient founder dating back to Viking times. Therefore we call it the “MIRAS-allele” in the revised manuscript. The patient collection in Finland, with so many individuals carrying identical genetic background of disease, but still manifesting in various ways in different patients (referring to the three main types of MIRAS as described in the manuscript – juvenile-onset epilepsy, early-adult-onset ataxia-polyneuropathy-epilepsy or middle-age-onset parkinson’s) is genetically exceptionally homogenous and unique.

The mutations that the reviewer mentions are referring to mouse POLG1 sequence, homologous for the human variants, and these variants are carried by our knock-in mice. All models studied in the revised paper are homozygous for the same pathogenic MIRAS allele, or the homologous mouse variants. The information of MIRAS patients and control individuals studied are now detailed in the ***Extended Data Table 1***.

Reviewer:

2) The number of biological replicates that were used for the transcriptomic and proteomic experiments should be included in the methods section.

Authors: We have updated the information and indicate in our revised manuscript the number of biological replicates of human and mouse samples analysed in each of their corresponded figure legends.

Reviewer:

3) The authors cite work on cytokine increases in certain PD patient serum samples. A subsequent and more thorough analysis would be good to mention as they also find increased mtDNA in patient sera (Brain 143: 3041, 2020).

Authors: Thank you for the suggestion. We have revised our introduction in the revised manuscript and included the reference suggested.

Reference list:

- Bogovič, P., A. Kastrin, S. Lotrič-Furlan, K. Ogrinc, T. A. Županc, M. Korva, N. Knap, and F. Strle. 2022. 'Clinical and Laboratory Characteristics and Outcome of Illness Caused by Tick-Borne Encephalitis Virus without Central Nervous System Involvement', *Emerg Infect Dis*, 28: 291-301.
- Euro, L., G. A. Farnum, E. Palin, A. Suomalainen, and L. S. Kaguni. 2011. 'Clustering of Alpers disease mutations and catalytic defects in biochemical variants reveal new features of molecular mechanism of the human mitochondrial replicase, Pol γ ', *Nucleic acids research*, 39: 9072-84.
- Euro, L., O. Haapanen, T. Róg, I. Vattulainen, A. Suomalainen, and V. Sharma. 2017. 'Atomistic Molecular Dynamics Simulations of Mitochondrial DNA Polymerase γ : Novel Mechanisms of Function and Pathogenesis', *Biochemistry*, 56: 1227-38.
- Hakonen, A. H., G. Davidzon, R. Salemi, L. A. Bindoff, G. Van Goethem, S. Dimauro, D. R. Thorburn, and A. Suomalainen. 2007. 'Abundance of the POLG disease mutations in Europe, Australia, New Zealand, and the United States explained by single ancient European founders', *Eur J Hum Genet*, 15: 779-83.
- Hakonen, A. H., S. Heiskanen, V. Juvonen, I. Lappalainen, P. T. Luoma, M. Rantamaki, G. V. Goethem, A. Lofgren, P. Hackman, A. Paetau, S. Kaakkola, K. Majamaa, T. Varilo, B. Udd, H. Kaariainen, L. A. Bindoff, and A. Suomalainen. 2005. 'Mitochondrial DNA polymerase W748S mutation: a common cause of autosomal recessive ataxia with ancient European origin', *Am J Hum Genet*, 77: 430-41.
- Hakonen, Anna, Steffi Goffart, Sanna Marjavaara, Anders Paetau, Helen Cooper, Kimmo Mattila, Milla Lampinen, Antti Sajantila, Tuula Lönnqvist, Johannes Spelbrink, and Anu Suomalainen Wartiovaara. 2008. 'Infantile-onset spinocerebellar ataxia and mitochondrial recessive ataxia syndrome are associated with neuronal complex I defect and mtDNA depletion', *Human Molecular Genetics*, 17: 3822-35.
- Hämäläinen, Riikka H., Juan C. Landoni, Kati J. Ahlqvist, Steffi Goffart, Sanna Ryytty, M. Obaidur Rahman, Virginia Brilhante, Katherine Icaý, Sampsa Hautaniemi, Liya Wang, Marikki Laiho, and Anu Suomalainen. 2019. 'Defects in mtDNA replication challenge nuclear genome stability through nucleotide depletion and provide a unifying mechanism for mouse progerias', *Nat Metab*, 1: 958-65.
- Ignatenko, O., J. Nikkanen, A. Kononov, N. Zamboni, G. Ince-Dunn, and A. Suomalainen. 2020. 'Mitochondrial spongiotic brain disease: astrocytic stress and harmful rapamycin and ketosis effect', *Life Sci Alliance*, 3.
- Ignatenko, Olesia, Dmitri Chilov, Ilse Paetau, Elena de Miguel, Christopher B. Jackson, Gabrielle Capin, Anders Paetau, Mugen Terzioglu, Liliya Euro, and Anu Suomalainen. 2018. 'Loss of mtDNA activates astrocytes and leads to spongiotic encephalopathy', *Nature Communications*, 9: 70.
- Lei, Yuanjiu, Camila Guerra Martinez, Sylvia Torres-Odio, Samantha L. Bell, Christine E. Birdwell, Joshua D. Bryant, Carl W. Tong, Robert O. Watson, Laura Ciaccia West, and A. Phillip West. 2021. 'Elevated type I interferon responses potentiate metabolic dysfunction, inflammation, and accelerated aging in mtDNA mutator mice', *Science Advances*, 7: eabe7548.
- Lujan, S. A., M. J. Longley, M. H. Humble, C. A. Lavender, A. Burkholder, E. L. Blakely, C. L. Alston, G. S. Gorman, D. M. Turnbull, R. McFarland, R. W. Taylor, T. A. Kunkel, and W. C. Copeland. 2020. 'Ultrasensitive deletion detection links mitochondrial DNA replication, disease, and aging', *Genome Biol*, 21: 248.

- Macao, Bertil, Jay P. Uhler, Triinu Siibak, Xuefeng Zhu, Yonghong Shi, Wenwen Sheng, Monica Olsson, James B. Stewart, Claes M. Gustafsson, and Maria Falkenberg. 2015. 'The exonuclease activity of DNA polymerase γ is required for ligation during mitochondrial DNA replication', *Nature Communications*, 6: 7303.
- Misić-Majerus, L., N. Bujić, V. Madarić, and T. Avsić-Zupanc. 2005. '[Hepatitis caused by tick-borne meningoencephalitis virus (TBEV)--a rare clinical manifestation outside the central nervous system involvement]', *Acta Med Croatica*, 59: 347-52.
- Neeve, Vivienne C. M., David C. Samuels, Laurence A. Bindoff, Bianca van den Bosch, Gert Van Goethem, Hubert Smeets, Anne Lombès, Claude Jardel, Michio Hirano, Salvatore DiMauro, Maaïke De Vries, Jan Smeitink, Bart W. Smits, Ireneus F. M. de Coo, Carsten Saft, Thomas Klopstock, Bianca-Cortina Keiling, Birgit Czermin, Angela Abicht, Hanns Lochmüller, Gavin Hudson, Grainne G. Gorman, Doug M. Turnbull, Robert W. Taylor, Elke Holinski-Feder, Patrick F. Chinnery, and Rita Horvath. 2012. 'What is influencing the phenotype of the common homozygous polymerase- γ mutation p.Ala467Thr?', *Brain*, 135: 3614-26.
- Palin, E. J., A. Lesonen, C. L. Farr, L. Euro, A. Suomalainen, and L. S. Kaguni. 2010. 'Functional analysis of H. sapiens DNA polymerase gamma spacer mutation W748S with and without common variant E1143G', *Biochim Biophys Acta*, 1802: 545-51.
- Tigano, Marco, Danielle C. Vargas, Samuel Tremblay-Belzile, Yi Fu, and Agnel Sfeir. 2021. 'Nuclear sensing of breaks in mitochondrial DNA enhances immune surveillance', *Nature*.
- Van Goethem, G., P. Luoma, M. Rantamaki, A. Al Memar, S. Kaakkola, P. Hackman, R. Krahe, A. Lofgren, J. J. Martin, P. De Jonghe, A. Suomalainen, B. Udd, and C. Van Broeckhoven. 2004. 'POLG mutations in neurodegenerative disorders with ataxia but no muscle involvement', *Neurology*, 63: 1251-7.
- West, A. P., W. Khoury-Hanold, M. Staron, M. C. Tal, C. M. Pineda, S. M. Lang, M. Bestwick, B. A. Duguay, N. Raimundo, D. A. MacDuff, S. M. Kaech, J. R. Smiley, R. E. Means, A. Iwasaki, and G. S. Shadel. 2015. 'Mitochondrial DNA stress primes the antiviral innate immune response', *Nature*, 520: 553-7.
- Winterthun, S., G. Ferrari, L. He, R. W. Taylor, M. Zeviani, D. M. Turnbull, B. A. Engelsen, G. Moen, and L. A. Bindoff. 2005. 'Autosomal recessive mitochondrial ataxic syndrome due to mitochondrial polymerase gamma mutations', *Neurology*, 64: 1204-8.

Reviewer Reports on the First Revision:

Referees' comments:

Referee #1 (Remarks to the Author):

I would like to congratulate the authors on a true tour de force effort to respond to all of the reviewers comments. This is the very first time I have ever seen a new mouse model created, validated and supportive of the major conclusions of a paper. On top of this, the collection of more patient samples and new knock-in cell lines is truly amazing.

You have addressed all of my concerns and your paper is extremely well executed and the results support the conclusions.

Referee #2 (Remarks to the Author):

In their revised manuscript "Ancestral allele of DNA polymerase gamma modifies mammalian antiviral tolerance", the authors have better focused the study and provided a wealth of new data (derived from patient and in vivo mouse models) that has greatly improved the manuscript. In doing so, they have adequately addressed the majority of my initial concerns. The study makes also makes an important connection of a mitochondrial disease to suppressed innate immune activation and highlights that viral infection can be a confounding environmental factor for this class of diseases...both very novel findings and of great general interest!

The one remaining gap that was not really addressed is exactly how decreased POLG1 activity (due to the MIRAS mutation) leads to dampened innate immune signaling. While the new data regarding decreased BrdU incorporation and replisome levels to demonstrate lower rates of mtDNA synthesis partially address this issue, it remains unclear how this connects to lower sensitivity of the innate immune sensors. Reduced release of mtRNA/mtDNA seems to be one prevailing idea of the authors (Fig. 5F), but they do not show that this is occurring in MIRAS mice or patient cells, which seems like a relatively straightforward experiment that could have been done, which would really tie the story together. In this regard, it has been previously shown (Ref 20) that depletion of cellular pyrimidine pools and increased expression of Slc25a33 (similar to what the authors report in MIRAS mice upon infection) stimulate mtDNA release.

Minor points:

1. How do the authors interpret data in Figure 1 and Extended Data Figure 2 that show a dampening of innate immune responses even below that of untreated cells (STAT2, IL6, IL-1beta)?
2. The boxes in Fig. 1D don't align with the bar graphs
3. Please indicate in Fig. 1 legend that the dsRNA mimetic used is poly(I:C).
4. dsRNA signal in Fig. 2D is impossible to see, please show grayscale image of single channels
5. All imaging data needs to be quantified
6. Figure legends should indicate reference genes used to quantify qPCR and loading controls used to quantify western blots
7. Subdividing extended data figures into more panels would improve readability (particularly extended data fig. 6, where 8 graphs/blots are split between two panels)
8. Figure 3A – the quantification of POLG2 in the brain doesn't appear to match the western blot shown

9. Dots on the quantification of western blots should be smaller, so that the extent of variability in the data can be evaluated more easily

Referee #3 (Remarks to the Author):

In the first round of reviews, the main concern was that the study was descriptive and correlative and did not provide causal or mechanistic insight. Unfortunately, despite addition of considerable amounts of new data, this remains true in the revised version.

Although it is interesting that cells harboring MIRAS mutations exhibit defective antiviral responses, it is not shown that defective antiviral responses explains (causes) the phenotypes of MIRAS patients; some other effect of the POLG1 mutation might be responsible, e.g., defective metabolic function, and the defects in innate signaling might just be a correlation.

It is also still not explained or shown exactly how the POLG1 mutations result in defective antiviral responses. It is speculated that decreased mtDNA results in decreased cytosolic nucleic acids to activate immune sensors, and impaired IFN responses, but this causal chain is not actually proven.

In this revised manuscript, the authors now present a mouse model of MIRAS, which may be a valuable tool, but at this point, it hasn't been used to address the mechanistic issues above.

Minor point

The second sentence of the abstract is not true. As the authors are clearly aware, there are prior examples of human mutations affecting mitochondrial function that elicit cGAS-STING signaling and cause disease, e.g., PARKIN, TFAM etc. (e.g., Sliter 2018 Nature, West 2015 Nature)

Referee #4 (Remarks to the Author):

The authors have extensively revised the manuscript and eliminated some of the contradictory or confusing data in the prior version. In sum, the manuscript now reports that hospital records show MIRAS patients have increased histories of immunodeficiencies. After exposure of MIRAS patient fibroblasts to either dsDNA or dsRNA there is initially a decreased innate immune response relative to wild type fibroblasts followed later by an increased NFkB response. After exposure to HSV-1, TBEV and SARS viruses, MIRAS patient fibroblasts show increased viral load relative to normal human fibroblasts supporting the idea of immune deficiency linked to MIRAS. The authors also include a new mouse model of MIRAS that reveals POLG1 and mtDNA depletion, a decrease in IFN signaling in brain after TBEV infection and a transcriptome change indirectly consistent with neurodegeneration. Metabolomics revealed that after TBEV infection several pathway changes occurred in MIRAS brain relative to normal mice and especially nucleotide metabolism suggesting to the authors that this may lead to further mtDNA depletion.

1) The authors conclude that altered responses to viral infection is linked to MIRAS disease phenotypes and may explain the variability in disease manifestation among POLG W748S carriers. However, the evidence for this important conclusion is quite tentative and based on correlations.

2) Another weakness is that the key mechanistic issue of how MIRAS alters innate immune responses is not clear. The authors speculate that it could be related to lower mtDNA copy

number that might alter innate immunity. The idea presented without support is that mtDNA is tonically released into the cytosol to tonically potentiate innate immunity – allowing cells to rapidly respond to virus infection. But how is mtDNA released at a steady level to accomplish this? The authors need to experimentally test this model that lower mtDNA copy number causes lower innate immune responses. For example, would increasing mtDNA copy number (such as by overexpressing TFAM or Twinkle at the right levels) increase innate immunity upon challenge? Would rescuing the MIRAS fibroblast mtDNA copy number deficiency with WT POLG (or even higher levels of W748S POLG) reduce innate immune responses to dsDNA or dsRNA? Overall, how distinct dsRNA and dsDNA innate immune pathways sense mtDNA copy number in the mitochondrial matrix remains an important missing link in the authors' suggested mechanistic model.

3) The metabolomic data shows many changes after a viral infection in MIRAS mice that are hard to interpret relative to MIRAS phenotypes or the authors' model of innate immunity.

Author Rebuttals to First Revision:

The authors responses to the Reviewers' comments

We sincerely thank the reviewers for their insightful comments. We are confident that these greatly improved the manuscript. We were able to respond to all their points and hope that these are now satisfactorily addressed.

Referee #1 (Remarks to the Author):

I would like to congratulate the authors on a **true tour de force effort to respond** to all of the reviewers comments. This is the very first time I have ever seen a new mouse model created, validated and supportive of the major conclusions of a paper. On top of this, the collection of more patient samples and new knock-in cell lines is truly amazing.

You have **addressed all of my concerns** and your paper is extremely well executed and the results support the conclusions.

Authors: Thank you very much for the enthusiastic and kind feedback!

Referee #2 (Remarks to the Author):

In their revised manuscript “Ancestral allele of DNA polymerase gamma modifies mammalian antiviral tolerance”, the authors have better focused the study and **provided a wealth of new data** (derived from patient and in vivo mouse models) that has **greatly improved the manuscript**. In doing so, they have adequately **addressed the majority of my initial concerns**. The study makes also makes an important connection of a mitochondrial disease to suppressed innate immune activation and highlights that viral infection can be a confounding environmental factor for this class of diseases...both very novel findings and of great general interest!

Authors: Thank you, we appreciate your positive comments.

The one remaining gap that was not really addressed is exactly how decreased POLG1 activity (due to the MIRAS mutation) leads to dampened innate immune signaling. While the new data regarding decreased BrdU incorporation and replisome levels to demonstrate lower rates of mtDNA synthesis partially address this issue, it remains unclear **how this connects to lower sensitivity of the innate** immune sensors. Reduced release of mtRNA/mtDNA seems to be one prevailing idea of the authors (Fig. 5F), **but they do not show that this is occurring in MIRAS mice or patient cells**, which seems like a relatively straightforward experiment that could have been done, which would really tie the story together.

In this regard, it has been previously shown (Ref 20) that depletion of cellular pyrimidine pools and increased expression of Slc25a33 (similar to what the authors report in MIRAS mice upon infection) stimulate mtDNA release.

Referees' comments:

Authors:

*Thank you, the comment was valuable and the new experiment has further improved the paper. We have now rigorously studied the DNA and RNA release after exposure to viral PAMP mimetics. We isolated cytosolic fraction from fibroblasts following 7 h dsDNA treatment and detected lower cytosolic mtDNA and mtRNA amount in MIRAS cells than the treated control lines. The 7h dsDNA treatment timepoint coincides with the IFN-I response, dampened in MIRAS, and likely contributes to the reduced initial activation of IFN response **(the new data is now in Figure 1g)**. Because of safety restrictions of handling TBEV infected tissues, we were not able to isolate cytosol fractions from mouse tissues without inactivating the virus. The inactivation however is not compatible with fractionation.*

The main viral sensor pathway activated is the RIG-I-related, a classical ds-RNA sensor. Recent reports show that both mtRNA and mtDNA can activate RIG-I with the latter via the RNA polymerase III/RIG-I pathway (Berry et al. mBio, 2021).

We also tried to visualize the released mtDNA fragments by immunofluorescence. At the early infection timepoint, however, the cells still have some viral mimetic (dsDNA), used to elicit the response, in the cytoplasm. This exogenous dsDNA is similarly detected by anti-DNA-antibody as endogenous mtDNA. Therefore, the data are not included.

Minor points:

1. How do the authors interpret data in Figure 1 and Extended Data Figure 2 that show a dampening of innate immune responses even below that of untreated cells (STAT2, IL6, IL-1beta)?

Authors: We thank the reviewer for careful reading. This is indeed an interesting observation, consistent both in controls and in MIRAS-patient cells and indicates a regulatory pathway more complex than direct activation vs inactivation of expression of the responses. The regulation of IL-6, IL-1beta and STAT2 expression are known to involve transcriptional, translational, posttranslational regulatory and miR regulation, some of which may explain the finding. We have not modified the manuscript to discuss this, because of length restriction.

2. The boxes in Fig. 1D don't align with the bar graphs

Authors: We have re-aligned the bar graphs with their respective boxes, thanks for noticing this. Data is in Figure 1d.

3. Please indicate in Fig. 1 legend that the dsRNA mimetic used is poly(I:C).

Authors: We have indicated poly(I:C) as the dsRNA mimetic treatment in Figure 1 legend and the related extended data figure legends.

4. dsRNA signal in Fig. 2D is impossible to see, please show grayscale image of single channels

Authors: We have now provided the grayscale image of all single channels, showing the increased number of cells with positive signals for TBEV and the viral dsRNA in MIRAS patient cell lines. Data is in Figure 2f.

Referees' comments:

5. All imaging data needs to be quantified

Authors: We have quantified and tabulated the imaging data including (I) immunofluorescence microscopy images (Figure 2g) showing TBEV positive cells as % of the total cells, (ii) histology: H&E, ORO and anti-CD3,4,8b and 68 staining of mouse livers (Figure 5b, 5e, Extended Data Figure 7g), measuring for number and size of immune cell infiltrates, or staining signal (% of tissue). Semiquantitative scoring to assess the overall severity of liver inflammation post viral infection has also been performed and is included in Figure 5b.

6. Figure legends should indicate reference genes used to quantify qPCR and loading controls used to quantify western blots

Authors: We have indicated now in the respective figure legends the reference gene used for gene expression/mtDNA amount normalization in qPCR and the loading control used for normalization of the western blot protein signal.

7. Subdividing extended data figures into more panels would improve readability (particularly extended data fig. 6, where 8 graphs/blots are split between two panels)

Authors: We have subdivided the extended data figures into more panels as suggested. The figure panels are now in Extended Data Figure 5e-j.

8. Figure 3A – the quantification of POLG2 in the brain doesn't appear to match the western blot shown

Authors: Thank you for noticing this. We have now re-quantified the Figure 3A western blotting signal for POLG2 and the other proteins examined in the brain. The quantified signal is tabulated in the box-and-whisker blot, with its significance calculated using unpaired 2-tailed student's t test – please see the new Figure 3a.

9. Dots on the quantification of western blots should be smaller, so that the extent of variability in the data can be evaluated more easily

Authors: We have reduced the size of the dots on the plots to improve the visualization as suggested.

Referees' comments:

Referee #3 (Remarks to the Author):

In the first round of reviews, the main concern was that the study was descriptive and correlative and did not provide causal or mechanistic insight. Unfortunately, despite addition of considerable amounts of new data, this remains true in the revised version.

Although it is interesting that cells harboring MIRAS mutations exhibit defective antiviral responses, it is not shown that defective antiviral responses explains (causes) the phenotypes of MIRAS patients; some other effect of the POLG1 mutation might be responsible, e.g., defective metabolic function, and the defects in innate signaling might just be a correlation.

***Authors:** To directly answer the question of the reviewer, whether the patients' disease is caused/contributed by antiviral responses, would require a blinded clinical intervention study on patients and controls, which is not feasible and certainly not in the scope of this article. Mitochondrial disease manifestations and progression are contributed by multiple mechanisms, because mitochondria are central in metabolism. We show here that immunity mechanism is an important such factor, especially as a trigger of symptoms.*

*In the manuscript, we have clarified the causal link as far as one can go with preclinical models. Our data from MIRAS mice, carrying the same homologous mutation as the human patients, **indicate without doubt that this POLG mutation causes increased sensitivity to viral infection.** Please note that the mice are asymptomatic before infection, develop a rapidly progressive and fulminant immune response in the liver inducing acute hepatitis, lipid accumulation and hepatic necrosis. **As new data, we present especial sensitivity of GABAergic interneurons to the viral insult (new data in Figure 4g,h).** These neurons are affected **only after the infection** in MIRAS mice. The results from PAMP mimetic and viral infection of the patient and control cells, and mouse model support the findings of initially lowered innate immunity response, followed by an overactivated pro-inflammatory response.*

To emphasize: MIRAS-knock-in mice have the POLG point mutation in all their organs. When facing a viral infection they manifest a disease in the tissues that are also affected in MIRAS patients – liver and brain.

*The sequence of infection and POLG disease-onset and progression have been reported in several case studies of POLG patients (POLG mutation identified after severe disease symptoms post Herpesvirus 6 or borrelia infection) (Hakonen et al. AJHG, 2005; Al-Zubeidi et al. Pediatr Neurol, 2014; Gaudó et al. Neurogenetics, 2020). **In all these cases, the patients were asymptomatic before the infection followed by acute onset of symptoms and rapid deterioration.** The infections were proposed to trigger/exacerbate MIRAS symptoms onset and deterioration. Also, GABAergic neuronal loss has been demonstrated in POLG patient brain autopsy studies (Hayhurst et al. Brain Pathol, 2019). Our MIRAS models replicate these reports in the human patients, after being infected by virus.*

Because of the high sensitivity of MIRAS mouse brain and liver to viral infection, we propose a sequence of the epileptic MIRAS form of 1) primary genetic POLG mutation 2) insufficient response to viral-infection, 3) GABA-interneuron dysfunction early in the infection -> Epilepsy, 4) Subclinical / mild hepatitis and, if valproate is given, induces liver damage caused by moderate/subclinical liver inflammation, which is not detected because of the devastating epilepsy. The sequence is now included as an overview image as Extended data figure 10.

Referees' comments:

Indeed, in a patient who has status epilepticus, the typical liver function markers (AST, especially) are not informative, because the seizures induce release of these enzymes also from the muscle. Therefore, a liver inflammation would not be easily detected in the acute epileptic phase.

We could not perform a valproate trial on top of the TBEV infection for the mice. Our ethical license prevents us from doing a potentially deleterious treatment on top of a severely damaging TBEV infection.

*All in all, our key question here is, why MIRAS patients **who all carry the same p.W748S variant as homozygotes**, manifest sometimes as status epilepticus in teenage years, while some patients get their first symptoms after 45 years, manifesting with Parkinson's disease. This indicates that modifying risk factor(s) affects the manifestation. We show here that viral infections can trigger a severe disease in the brain and liver.*

Rev: It is also still not explained or shown exactly how the POLG1 mutations result in defective antiviral responses. It is speculated that decreased mtDNA results in decreased cytosolic nucleic acids to activate immune sensors, and impaired IFN responses, but this causal chain is not actually proven.

*We agree with the Reviewer that this is an important point. We have now rigorously studied the DNA and RNA release of patient and control cells after exposure to viral PAMP mimetics. We found **lowered amount of both mtDNA and mtRNA** in the cytoplasm of MIRAS cells compared to controls. We performed cellular fractionation to isolate cytosolic fraction out of the cells treated with 7h dsDNA (that induced weaker IFN response in MIRAS cells) and qPCR analysis for mtDNA/RNA amount. **The new data is now in Figure 1g.***

In this revised manuscript, the authors now present a mouse model of MIRAS, which may be a valuable tool, but at this point, it hasn't been used to address the mechanistic issues above.

*As explained above, the mice show remarkable replication of findings in MIRAS patients, when induced with a viral infection. The loss of GABAergic neurons (**new data in Figure 4g,h**) as previously has been shown in patients. We show here that this loss can be induced by a viral infection in MIRAS mutation carriers. We also show the sensitivity of the liver to infection, with fulminant hepatitis inducing necrosis.*

Because of safety restrictions of handling TBEV-infected tissues, we were not able to isolate cytosol fractions from TBEV-infected mouse tissues. The inactivation procedure required is not compatible with fractionation. Therefore, we did the experiments on cell fractionation and mtDNA/mtRNA release in PAMP-treated patient and control cells. We demonstrated decreased release of these in MIRAS-mutant cells, consistent for slow innate immunity induction and mtDNA depletion in the different MIRAS models of ours. We wish that these experiments, together with GABA data convince the reviewer. No further experiments were made here, as no suggestions of other specific mechanistic studies were made by the reviewer.

Rev:

Minor

point

The second sentence of the abstract is not true. As the authors are clearly aware, there are prior examples of human mutations affecting mitochondrial function that elicit cGAS-STING signaling and cause disease, e.g., PARKIN, TFAM etc. (e.g., Sliter 2018 Nature, West 2015 Nature)

Referees' comments:

*Authors: We consider disease models to be such that carry genetic defects present in human patients. Mechanistic models are valuable (e.g. conditional KOs) to test physiological roles of proteins, but relevance to disease pathogenesis is not self-evident. Of the references that the reviewer mentions: (i) the *Tfam*^{+/-} MEFs in West 2015 Nature: hemizyosity for *TFAM*, decrease of protein amount, no mutations. The rare cases of *TFAM* mutation in patients have all been single amino acid changes – homozygous or compound heterozygous mutations causing a recessive disease. The heterozygous carriers of *TFAM* mutations in those families do not show symptoms. Therefore, *TFAM* knockouts are truly valuable experimental models, but not human disease models (full KO fatal in embryogenesis). (ii) Sliter, 2018 Nature: *Pink1* or *Parkin* KO mice. Human diseases with inactivating copy number variants (deletions, frame-shifts) do exist for both *Parkin* and *Pink1*. Therefore, these full-body KO-models fill the criteria of disease models. (iii) mtDNA mutator mice: Mutator mouse model has *POLG1* protein mutation that inactivates exo-domain and the proof-reading function. While the mice accumulate random mtDNA mutations, the progeric phenotype of theirs is associated with accumulating genomic DNA breaks and nucleotide pool depletion in stem cells (Hämäläinen et al. Nature Metabolism 2019; Ahlqvist et al. Cell Metab 2012). Also, so far, no patients with *POLG* exonuclease-deficient diseases and random mtDNA mutagenesis or progeria have been reported. MtDNA mutator is an interesting and valuable mechanistic model for stem-cell pool-related progeria.*

For conciseness, we have revised the sentence in the abstract as follow: Mitochondria are arising as critical modulators of... “The relevance of these mechanisms for mitochondrial diseases remains understudied”.

Referee #4 (Remarks to the Author):

The authors have extensively revised the manuscript and eliminated some of the contradictory or confusing data in the prior version. In sum, the manuscript now reports that hospital records show MIRAS patients have increased histories of immunodeficiencies.

Authors: The data was derived from a human population database, FinnGen, of >300,000 individuals with their genotypes linked to their medical diagnoses. MIRAS carriers can be identified because of the common occurrence of the variant in Finland, and such epidemiological data links the variant p.W748S to immunodeficiencies. This is quite remarkable, to our opinion, and strongly supports our data linking viral immunity from human and mice homozygous for the same variant. Similar association is not present in other nuclear-encoded mitochondrial disease gene variants tested.

After exposure of MIRAS patient fibroblasts to either dsDNA or dsRNA there is initially a decreased innate immune response relative to wild type fibroblasts followed later by an increased NFkB response. After exposure to HSV-1, TBEV and SARS viruses, MIRAS patient fibroblasts show increased viral load relative to normal human fibroblasts supporting the idea of immune deficiency linked to MIRAS.

Authors: Yes, this is accurate.

Referees' comments:

The authors also include a new mouse model of MIRAS that reveals POLG1 and mtDNA depletion, a decrease in IFN signaling in brain after TBEV infection and a transcriptome change indirectly consistent with neurodegeneration. Metabolomics revealed that after TBEV infection several pathway changes occurred in MIRAS brain relative to normal mice and especially nucleotide metabolism suggesting to the authors that this may lead to further mtDNA depletion.

Authors: We actually do show that mtDNA depletion progresses in the brain during the infection, showing significantly less mtDNA amount in MIRAS at 4 dpi consistent with the brain metabolome alteration at 4 dpi (Data in Figure 4e). More explanations are in responses to comments 2 and 3.

1) The authors conclude that altered responses to viral infection is linked to MIRAS disease phenotypes and may explain the variability in disease manifestation among POLG W748S carriers. However, the evidence for this important conclusion is quite tentative and based on correlations.

*Authors: In the manuscript, we have clarified the causal link as far as one can go with preclinical models. Our data from MIRAS mice, carrying the same homologous mutation as the human patients, **indicate without doubt that this POLG mutation causes increased sensitivity to viral infection.** Please note that the mice are asymptomatic before infection, and develop a rapidly progressive and fulminant immune response in the liver inducing acute hepatitis, lipid accumulation and hepatic necrosis. **As new data, we present especial sensitivity of GABAergic interneurons to the viral insult (new data in Figure 4g,h).** These neurons are affected **only after the infection** of MIRAS mice. The controls do not show similar depletion. The results from PAMP mimetics and viral infection of the patient and control cells support the findings of initially lowered innate immunity response, followed by an overactivated pro-inflammatory response.*

*The sequence of infection and POLG disease-onset and progression have been reported in several case studies of POLG patients (POLG mutation identified after severe disease symptoms post Herpesvirus 6 or borrelia infection) (Hakonen et al. AJHG, 2005; Al-Zubeidi et al. Pediatr Neurol, 2014; Gaudó et al. Neurogenetics, 2020). **In all these cases, the patients were asymptomatic before the infection followed by acute onset of symptoms and rapid deterioration.** The infection was proposed to trigger/exacerbate MIRAS symptoms onset and deterioration. Also, GABAergic neuronal loss has been demonstrated in POLG patient brain autopsy studies (Hayhurst et al. Brain Pathol, 2019). These we replicate fully in our MIRAS models, the symptoms being induced by an acute viral infection. We show that the mechanism involves increased early viral replication and overactivated pro-inflammatory response.*

Because of the high sensitivity of MIRAS mouse brain and liver to viral infection in line with clinical symptoms of adolescent-onset MIRAS patients showing epilepsy and valproate-induced liver failure, we propose a sequence of the epileptic MIRAS form of 1) primary genetic POLG mutation 2) insufficient response to viral-infection, 3) GABA-interneuron dysfunction early in the infection -> Epilepsy, 4) Subclinical / mild hepatitis and, if valproate is given, induces liver damage caused by moderate/subclinical liver inflammation, which is not detected because of the devastating epilepsy. An overview of this sequence has now been included as the Extended Data Figure 10.

Indeed, in a patient who has status epilepticus, the typical liver function markers (AST, especially) are not informative, because the seizures induce release of these enzymes also from

Referees' comments:

the muscle. Therefore, a liver inflammation would not be easily detected in the acute epileptic phase.

Our ethical license prevents us from doing a potentially deleterious treatment on top of a severely damaging TBEV infection, which is the reason we could not perform a valproate trial on top of the TBEV infection for the mice.

*All in all, our key question here is, why MIRAS patients **who all carry the same p.W748S variant as homozygotes**, manifest sometimes as status epilepticus in teenage years, while some patients get their first symptoms after 45 years, manifesting with Parkinson's disease. This indicates that modifying risk factor(s) affects the manifestation. We show here that viral infections can trigger a severe MIRAS-like disease in the brain and liver of knock-in mice that carry the homologous MIRAS mutation.*

2) Another weakness is that the key mechanistic issue of how MIRAS alters innate immune responses is not clear. The authors speculate that it could be related to lower mtDNA copy number that might alter innate immunity. The idea presented without support is that mtDNA is tonically released into the cytosol to tonically potentiate innate immunity – allowing cells to rapidly respond to virus infection. But how is mtDNA released at a steady level to accomplish this?

*Authors: We have now rigorously studied the DNA and RNA release of patient and control cells after exposure to viral PAMP mimetics. We isolated cytosolic fraction from fibroblasts following 7 h dsDNA treatment and detected lower cytosolic mtDNA and mtRNA amount in MIRAS cells than the treated control lines. The timepoint coincides with the IFN-I response, dampened in MIRAS, and likely contributes to the reduced initial activation of IFN response **(the new data is now in Figure 1g)**. Because of safety restrictions of handling TBEV-infected tissues, we were not able to isolate cytosol fractions from TBEV-infected mouse tissues. The inactivation procedure required is not compatible with fractionation.*

We also tried to visualize the released mtDNA fragments by immunofluorescence. At the early infection timepoint, however, the cells still have some viral mimetic (dsDNA) in the cytoplasm, which is similarly detected by anti-DNA-antibody as the endogenous mtDNA. These data are therefore not included.

The authors **need to experimentally test this model that lower mtDNA copy number causes lower innate immune responses**. For example, would increasing mtDNA copy number (such as by overexpressing TFAM or Twinkle at the right levels) increase innate immunity upon challenge? Would rescuing the MIRAS fibroblast mtDNA copy number deficiency with WT POLG (or even higher levels of W748S POLG) reduce innate immune responses to dsDNA or dsRNA?

Authors: The ideas are interesting. However, for Twinkle-overexpression to work and increase mtDNA copy number, POLG protein is required as well. To add helicase to cells that lack mtDNA replicase does not increase mtDNA copy number. TFAM supercoils mtDNA and slows down replication and transcription and increases mtDNA half-life. (Ylikallio et al. Hum Mol Genet, 2010; Hämäläinen et al. Nature Metab, 2019). Reduction of TFAM amount, not increase, has been shown to increase mtDNA release and innate immunity activation (West et al. Nature, 2015). Neither TFAM or Twinkle overexpression is optimal.

Referees' comments:

We therefore chose to correct the MIRAS mutation (p.W748S), as suggested by the reviewer. Primary fibroblasts resist CRISPR/Cas9 editing, but we created induced fibroblast-like cells from the mutation-corrected iPSC clones. Correction of MIRAS mutation p.W748S ("MIRAS-corrected") successfully restored POLG1 protein stability (Extended Data Figure 5a,b). Following 48 h of HSV-1 infection, activation of p-MLKL and p-NF- κ B-p65 were milder in MIRAS-corrected lines compared to the mutant-carrying patient cells (Extended Data Figure 5c). Furthermore, mtDNA depletion, induced by HSV-1, was now similar to that in control-lines - the severe depletion of mtDNA induced by viral infection detected in the original MIRAS cells was rescued (Extended Data Figure 5d). These new data support a role for POLG1 in mediating the cellular tolerance to viral infection.

Rev: Overall, how distinct dsRNA and dsDNA innate immune pathways sense mtDNA copy number in the mitochondrial matrix remains an important missing link in the authors suggested mechanistic model.

*Authors: Actually, we propose that mitochondria sense the viral infection and mediate the signal to the cytoplasm. Previous data show that HSV-1 targets mtDNA early in its infection by inserting its protein UL12.5, which causes mtDNA depletion (Saffran et al. EMBO Rep, 2007). Release of mtDNA into cytoplasm during HSV-1 infection, promote IFN induction via the RIG-I-RNA polymerase III pathway (Berry et al. mBio, 2021). The concept of mtDNA/mtRNA release has therefore been proven in this and other articles (Tigano et al. Nature, 2021; de Regt et al. Nucleic Acids Res, 2023; Doke et al., Nature Metabolism, 2023), the first one being West et al. Nature, 2015. **However, our evidence indicates that this mechanism is relevant in a disease: a major modifying factor for manifestation of a neurological disease. The mechanism involves reduced mtDNA/RNA presentation, as a consequence of low amounts of POLG, mtDNA replisome and mtDNA.***

A putative mechanism of mitochondrial sensing are nucleotide pools that are depleted by virus replication. Low nucleotide availability makes POLG fall off the mtDNA template, and the unfinished fragments are available to be presented to cytoplasm. In post-mitotic cells the nucleotide pools are mostly serving mtDNA replication by salvage pathway and are even 1000-fold lower than in proliferating cells.

*The MIRAS-variant POLG-p.W748S is known to cause decreased binding of POLG to mtDNA template and challenge the interaction with other replisome proteins (Euro et al. Biochemistry, 2017). We show that the p.W748S leads to depletion of POLG amount and assembled mtDNA replisome, as well as mtDNA loss by the viral infection. As new data, we now also demonstrated lowered release of mtDNA/mtRNA fragments in the cytosol to activate the viral sensors. **The new data is now included in Figure 1g.***

3) The metabolomic data shows many changes after a viral infection in MIRAS mice that are hard to interpret relative to MIRAS phenotypes or the authors model of innate immunity.

*Authors response: Thank you for the comment. We have now further clarified these findings. **As the most relevant findings in the metabolomic data of MIRAS we report acute infection-induced impact on nucleotide metabolism causing decreased steady-state pools of pyrimidines (UMP, dUDP, thymine, thymidine, deoxycytidine, deoxyribose) required for the cellular RNA and DNA synthesis; and 2) altered methyl cycle and transsulfuration pathway driving e.g. glutathione synthesis that is required for nucleotide synthesis and redox regulation, (Figure 4b-d).** As explained above, these challenge mtDNA replication and aggravate loss of mtDNA,*

Referees' comments:

as demonstrated by the enhanced mtDNA depletion in MIRAS mouse brain at 4 dpi of TBEV (Figure 4e).

As our new finding, we also report acute viral-induced loss of GABA-ergic neurons. As now added in the discussion, GABA and nucleotide pools are linked via GABA transaminase, ABAT. It has a dual role in GABA synthesis and also in nucleotide salvage pathway (Besse et al. Cell Metab, 2015). We hypothesize that prioritization of ABAT to either GABA or nucleotide metabolism in GABAergic neurons would challenge viability – either by nucleotide depletion or by loss of inhibitory signaling. Single-cell metabolomics in GABA-ergic neurons is an interesting future direction.

Reference list:

- 1 Hakonen, A. H. et al. Mitochondrial DNA polymerase W748S mutation: a common cause of autosomal recessive ataxia with ancient European origin. *American journal of human genetics* 77, 430-441 (2005). <https://doi.org/10.1086/444548>
- 2 Al-Zubeidi, D. et al. Fatal human herpesvirus 6-associated encephalitis in two boys with underlying POLG mitochondrial disorders. *Pediatr Neurol* 51, 448-452 (2014). <https://doi.org/10.1016/j.pediatrneurol.2014.04.006>
- 3 Gaudó, P. et al. Infectious stress triggers a POLG-related mitochondrial disease. *Neurogenetics* 21, 19-27 (2020). <https://doi.org/10.1007/s10048-019-00593-2>
- 4 Hayhurst, H. et al. Dissecting the neuronal vulnerability underpinning Alpers' syndrome: a clinical and neuropathological study. *Brain Pathol* 29, 97-113 (2019). <https://doi.org/10.1111/bpa.12640>
- 5 Ahlqvist, K. J. et al. Somatic progenitor cell vulnerability to mitochondrial DNA mutagenesis underlies progeroid phenotypes in Polg mutator mice. *Cell Metab* 15, 100-109 (2012). <https://doi.org/10.1016/j.cmet.2011.11.012>
- 6 Ylikallio, E., Tyynismaa, H., Tsutsui, H., Ide, T. & Suomalainen, A. High mitochondrial DNA copy number has detrimental effects in mice. *Human Molecular Genetics* 19, 2695-2705 (2010). <https://doi.org/10.1093/hmg/ddq163>
- 7 Hämäläinen, R. H. et al. Defects in mtDNA replication challenge nuclear genome stability through nucleotide depletion and provide a unifying mechanism for mouse progerias. *Nature Metabolism* 1, 958-965 (2019). <https://doi.org/10.1038/s42255-019-0120-1>
- 8 West, A. P. et al. Mitochondrial DNA stress primes the antiviral innate immune response. *Nature* 520, 553-557 (2015). <https://doi.org/10.1038/nature14156>
- 9 Saffran, H. A., Pare, J. M., Corcoran, J. A., Weller, S. K. & Smiley, J. R. Herpes simplex virus eliminates host mitochondrial DNA. *EMBO Rep* 8, 188-193 (2007). <https://doi.org/10.1038/sj.embor.7400878>
- 10 Berry, N. et al. Herpes Simplex Virus Type 1 Infection Disturbs the Mitochondrial Network, Leading to Type I Interferon Production through the RNA Polymerase III/RIG-I Pathway. *mBio* 12, e02557-02521 (2021). <https://doi.org/doi:10.1128/mBio.02557-21>
- 11 Tigano, M., Vargas, D. C., Tremblay-Belzile, S., Fu, Y. & Sfeir, A. Nuclear sensing of breaks in mitochondrial DNA enhances immune surveillance. *Nature* (2021). <https://doi.org/10.1038/s41586-021-03269-w>
- 12 de Regt, A. K. et al. A conserved isoleucine in the binding pocket of RIG-I controls immune tolerance to mitochondrial RNA. *Nucleic Acids Res* (2023). <https://doi.org/10.1093/nar/gkad835>
- 13 Doke, T. et al. NAD⁺ precursor supplementation prevents mtRNA/RIG-I-dependent inflammation during kidney injury. *Nature Metabolism* 5, 414-430 (2023). <https://doi.org/10.1038/s42255-023-00761-7>
- 14 Euro, L. et al. Atomistic Molecular Dynamics Simulations of Mitochondrial DNA Polymerase γ : Novel Mechanisms of Function and Pathogenesis. *Biochemistry* 56, 1227-1238 (2017). <https://doi.org/10.1021/acs.biochem.6b00934>
- 15 Besse, A. et al. The GABA transaminase, ABAT, is essential for mitochondrial nucleoside metabolism. *Cell Metab* 21, 417-427 (2015). <https://doi.org/10.1016/j.cmet.2015.02.008>

Reviewer Reports on the Second Revision:

Referees' comments:

Referee #2 (Remarks to the Author):

The authors have satisfied all of my remaining concerns. This is a tour-de-force study and illuminates very important new concepts for the mitochondrial disease field with regard to the role of viral infection and inflammation in disease pathogenesis.

Referee #3 (Remarks to the Author):

The major claim of the paper is that POLG1 mutations cause MIRAS by somehow impairing an early anti-viral response, which then leads to an exacerbated inflammatory response to infection, which then results in disease.

The authors' model is somewhat confusing because it simultaneously proposes that the MIRAS patients are BOTH immune compromised and ALSO suffering from excessive inflammation. Immunosuppression and inflammation can certainly co-exist, but such a scenario is also very challenging to explain in a convincing manner. Indeed, my central criticism of the original and revised versions of the paper was that the mechanistic connections proposed to exist between POLG1 mutations and MIRAS remained very poorly substantiated by the data. I think this is still the case. My specific criticisms, which are largely unchanged from before, are as follows

1. No plausible mechanism connecting POLG1 mutations to defective immune responses is provided.

Figure 1 shows that MIRAS (POLG1) mutant cells exhibit a modest defect in the innate immune response to both dsDNA and dsRNA. It remains unclear why this is the case and no true mechanism connecting the function of POLG1 to this defective response is provided. The new data provided by the authors suggests that MIRAS cells exhibit lower levels of mitochondrial DNA and RNA in the cytosol after dsDNA/dsRNA stimulation. But it remains unclear HOW this explains the lower response to exogenous dsDNA/dsRNA stimulation (wouldn't the exogenously provided dsDNA/dsRNA itself be stimulatory independent of any released mtDNA/RNA?). And it also remains unclear WHETHER the lower levels of cytosolic mtDNA/RNA explain the defective response to dsDNA/dsRNA stimulation. Some other (metabolic?) defect in the cells arising from POLG1 mutation might explain the defective response. Lastly, it remains unexplained why responses to BOTH dsDNA and dsRNA are defective. dsDNA is detected by cGAS, whereas dsRNA is detected by MDA5/RIG-I. If the RIG-I signaling was impaired then this would not be predicted to affect the response to dsDNA (The PolIII pathway only contributes in a very minor way in limited cell types in response to certain stimuli, and the cytosolic response to dsDNA depends entirely on cGAS, not at all on RIG-I). It seems that some common component of both dsDNA and dsRNA signaling must be affected. But which one? The authors do not explain. It is possible that there is no defect in cytosolic dsRNA/DNA signaling pathways per se, but instead a generic defect in the metabolic state of the cell, or some other defect, that "non-specifically" affects the ability of cells to mount a response. Overall the paper is very unsatisfying on all these counts.

2. No plausible mechanism for how the defective antiviral response leads to an exacerbated inflammatory response is provided, and it is not even clear these two phenotypes are causally connected.

The authors show that MIRAS cells exhibit slightly elevated inflammatory responses at late

timepoints, but why or how this occurs is not shown. Interestingly, the elevated inflammatory response is seen after dsDNA/dsRNA stimulation, so cannot be a result of enhanced viral replication (in this case, there is no virus). It is not shown that the excessive inflammatory response arises as a result of the defective antiviral response (both phenotypes exist, but they might be unrelated). Overall, the excessive inflammatory response is both a modest and confusing phenotype, leading me to wonder if it might be very indirect or non-specific.

3. Most importantly: neither the defective antiviral response nor the exacerbated inflammatory response is shown to underlie the disease pathology of MIRAS.

The strength of the paper is (a) showing that both human MIRAS cells and the POLG1 mutant mouse are susceptible to virus infection; and (b) that infection of the POLG1 mutant mice leads to MIRAS-associated disease phenotypes. I think this is a significant step forward. But the underlying mechanism remains unclear. Is the lack of virus control responsible (i.e., does damage arise due to enhanced virus replication)? Or, is the exacerbated inflammatory response responsible? Or is something else going on (e.g., metabolic alterations, defective stress responses, etc.)? We really have no idea. In fact, so much seems to be altered in the mice, including metabolic alterations (as shown by the authors in Figure 4), it is difficult to discern what is causing what. In the end, the manuscript amounts to a comprehensive and detailed descriptive analysis of the MIRAS phenotype, and the generation of a new mouse model of the disease, both of which are certainly of value, but in the absence of almost any mechanistic insight, I believe the manuscript falls short of what I believe is generally expected of papers in Nature.

Minor point:

line 88-89: the authors state that there is NO difference in POLG1 transcript or protein levels, immediately after just stating that there was a defect in POLG1 protein levels (line 87). This is confusing.

Referee #4:

The authors rigorously addressed my comments to the extent possible. I find the work important and support publication without further experimentation.

PS UW:

Please note the format related points raised by our editorial assistants as you revised the manuscript.

1. Please submit a revised title within 75 characters (including spaces) that is free of any punctuation marks like colons, exclamation marks, full stops or speech marks.
2. Please reduce subheadings to 40 characters (with spaces) or less.
3. Please provide a supplementary information guide.
4. Flagging that there are potential third-party rights issues in the figures - please check sources or if permissions are needed for the mice, cells, and petri plates illustrations in the figures.
5. Figures 3, 4 & 6 are too tall in height, please reduce them to 17 cm or less.
6. Please ensure that the text size in all figures is at least 5 pt Arial.

Author Rebuttals to Second Revision:

Response to reviewers:

Referee #2 (Remarks to the Author):

The authors have satisfied all of my remaining concerns. This is a tour-de-force study and illuminates very important new concepts for the mitochondrial disease field with regard to the role of viral infection and inflammation in disease pathogenesis.

Authors: Thank you, we truly appreciate your comment.

Referee #3 (Remarks to the Author):

The major claim of the paper is that POLG1 mutations cause MIRAS by somehow impairing an early anti-viral response, which then leads to an exacerbated inflammatory response to infection, which then results in disease.

The authors' model is somewhat confusing because it simultaneously proposes that the MIRAS patients are BOTH immune compromised and ALSO suffering from excessive inflammation. Immunosuppression and inflammation can certainly co-exist, but such a scenario is also very challenging to explain in a convincing manner. Indeed, my central criticism of the original and revised versions of the paper was that the mechanistic connections proposed to exist between POLG1 mutations and MIRAS remained very poorly substantiated by the data.

Authors: This comment is somewhat unclear to us. POLG1 mutations, and especially the one handled here (p.W748S) are causative for MIRAS, as was published by us already in 2005 (Hakonen et al) and a Norwegian group (Winterthun et al. 2005) and replicated by a high number of publications ever since – and also introduced in detail in the manuscript. The POLG1-mutation p.W748S (founder allele) is globally the second-most common mutation underlying MIRAS and POLG-diseases in children and adults and the most common in Northern Europe. There is no doubt that POLG1 mutations underlie MIRAS.

I think this is still the case. My specific criticisms, which are largely unchanged from before, are as follows

1. No plausible mechanism connecting POLG1 mutations to defective immune responses is provided.

Figure 1 shows that MIRAS (POLG1) mutant cells exhibit a modest defect in the innate immune response to both dsDNA and dsRNA. It remains unclear why this is the case and no true mechanism connecting the function of POLG1 to this defective response is provided. The new data provided by the authors suggests that MIRAS cells exhibit lower

levels of mitochondrial DNA and RNA in the cytosol after dsDNA/dsRNA stimulation. But it remains unclear HOW this explains the lower response to exogenous dsDNA/dsRNA stimulation (wouldn't the exogenously provided dsDNA/dsRNA itself be stimulatory independent of any released mtDNA/RNA?).

Authors: This is an interesting question which actually applies to all publications since the landmark publication of West et al, Nature 2015, describing that mitochondrial nucleic acids have a role in eliciting innate immunity. These reports studied models with experimental mitochondrial dysfunctions and found an overactivated early IFN-I response as a consequence of PAMP mimetics (dsDNA, poly:IC). Our results are the first to show that in a human disease the mechanism can be the opposite: less released mt-nucleic acids and slowed-down viral sensor activation. Why, then, poly:IC is not enough? First of all, mtDNA is a plasmid and can produce a large amount of mtRNA which makes it a significant amplifier of the viral response if released to the cytoplasm, when exogenous nucleic acids enter (see e.g. Tadepalle & Shadel, Mol Cell 2021). Secondly, RIG-I can bind different RNAs and these have different consequences. It does bind poly I:C, which results in a conformational change exposing a repressor domain and activating RIG-I. However, when 5'ppp-uncapped ssRNAs, such as some mtDNA transcripts, bind RIG-I, they disrupt the inhibitory interaction between RIG-I repressor domain and CARDs – caspase activation recruitment domains - and activate the viral sensor (See e.g. Saito & Gale J Exp Med 2008; Tan et al. 2018). Our results indicate that mitochondrial nucleic acid release is indeed required for the rapid early-stage innate immunity, because without it PAMPs induction is slower. IFN-I induction does still occur, but slower.

*The exploration of this interesting topic is out of scope of this article. **We have however, added a sentence to the discussion on this interesting point highlighted by the reviewer.** “A recent report showed that mtDNA breaks activated RIG-I via mtRNA release (Tigano et al, 2021). These findings highlight the importance of mtRNA release for innate immunity, which aligns well with our findings in MIRAS. Indeed, we show that even in the presence of poly I:C, mimicking viral RNA, MIRAS cells trigger a lowered IFN-I response. These data suggest that mitochondrial nucleic acid release is necessary for full activation of the early-stage antiviral response and that mtDNA replisome is an active component of innate immune responses in vivo.”*

And it also remains unclear WHETHER the lower levels of cytosolic mtDNA/RNA explain the defective response to dsDNA/dsRNA stimulation. Some other (metabolic?) defect in the cells arising from POLG1 mutation might explain the defective response.

Authors: IFN-I response is contributed by mitochondrial nucleic acid release as explained above. This is a well-respected fact in the field and backed up by a number of excellent publications. We find lowered release and slowed down IFN-I response, with increased viral replication in early stage – a logical consequence of insufficient antiviral response. This overactivates inflammatory response and causes tissue-specific damage in patient materials and in vivo in mice. Inflammatory pathways are extensively activated. To sum up, our data show that the globally spread founder mutation in POLG is causing immune-mechanistic sensitivity to viral infections that triggers and exacerbates the disease in mouse and human materials.

Lastly, it remains unexplained why responses to BOTH dsDNA and dsRNA are defective. dsDNA is detected by cGAS, whereas dsRNA is detected by MDA5/RIG-I. If the RIG-I signaling was impaired then this would not be predicted to affect the response to dsDNA (The PolIII pathway only contributes in a very minor way in limited cell types in response to certain stimuli, and the cytosolic response to dsDNA depends entirely on cGAS, not at all on RIG-I). It seems that some common component of both dsDNA and dsRNA signaling must be affected. But which one? The authors do not explain. It is possible that there is no defect in cytosolic dsRNA/DNA signaling pathways per se, but instead a generic defect in the metabolic state of the cell, or some other defect, that “non-specifically” affects the ability of cells to mount a response. Overall the paper is very unsatisfying on all these counts.

Authors: It is interesting indeed that we find both to be lowered. An increasing amount of publications report crosstalk of cytoplasmic DNA and RNA sensing machineries, and these may vary in different cell types. The field is in its early stages in medicine and little is known of the brain. In the course of (all) diseases, a high number of metabolic pathway do change and can contribute to disease progression. However, to understand diseases as complex as e.g. MIRAS, the phenotypes ranging from epilepsy to ataxia and parkinson's disease, even when caused by a single mutation, these mechanisms have to be clarified in different stages of diseases to find mechanisms and therapy targets. We show that a viral disease can trigger the manifestation of a brain and liver disease, a common combination in mitochondrial diseases. In general, our data indicate that extrinsic factors, such as viral infections, are major modulators of mitochondrial diseases in the nervous system and liver.

2. No plausible mechanism for how the defective antiviral response leads to an exacerbated inflammatory response is provided, and it is not even clear these two phenotypes are causally connected.

The authors show that MIRAS cells exhibit slightly elevated inflammatory responses at late timepoints, but why or how this occurs is not shown. Interestingly, the elevated inflammatory response is seen after dsDNA/dsRNA stimulation, so cannot be a result of enhanced viral replication (in this case, there is no virus). It is not shown that the excessive inflammatory response arises as a result of the defective antiviral response (both phenotypes exist, but they might be unrelated). Overall, the excessive inflammatory response is both a modest and confusing phenotype, leading me to wonder if it might be very indirect or non-specific.

Authors: As the comment is the same as above, we refer the Reviewer to our previous responses.

3. Most importantly: neither the defective antiviral response nor the exacerbated inflammatory response is shown to underlie the disease pathology of MIRAS.

Authors: Our MIRAS-knock-in mouse data indicate that viral infection triggers dramatically a brain and liver disease (subacute GABA-neuron degradation and hepatic failure, similar to MIRAS teenagers) as its rapid consequence with wide-spread activation of innate immunity aberrations. The data indicate that MIRAS-allele is sensitizing the animals to viral-triggered disease manifestations. We are excited as our study is highly likely to motivate clinical studies to target innate immunity pathways upon the first manifestation of the disease (healthy teenager manifesting with first epileptic seizure, with prognostic knowledge of a recent infection) as well as primary and secondary diseases of POLG-symptom spectrum. Such studies require, however, multicentre collaborations and years of work, and obviously are not in the scope of this article. Our results are highly important because POLG disease spectrum expands to a wide field of neurology, to diseases that are always devastating, rapidly progressive with no previous knowledge of molecular mechanism or targeted therapies.

The strength of the paper is (a) showing that both human MIRAS cells and the POLG1 mutant mouse are susceptible to virus infection; and (b) that infection of the POLG1 mutant mice leads to MIRAS-associated disease phenotypes. I think this is a significant step forward.

Authors: Thank you, we do agree.

But the underlying mechanism remains unclear. Is the lack of virus control responsible (i.e., does damage arise due to enhanced virus replication)? Or, is the exacerbated inflammatory response responsible?

Authors: We show that virus replicates faster, which is followed by overactivated, delayed inflammation and damage. The inflammation is a known cause of tissue damage, including MIRAS-like symptoms e.g. in HSV1 encephalitis and is likely to be the cause of tissue damage by activation of necroptosis, as we report.

Or is something else going on (e.g., metabolic alterations, defective stress responses, etc.)? We really have no idea. In fact, so much seems to be altered in the mice, including metabolic alterations (as shown by the authors in Figure 4), it is difficult to discern what is causing what. In the end, the manuscript amounts to a comprehensive and detailed descriptive analysis of the MIRAS phenotype, and the generation of a new mouse model of the disease, both of which are certainly of value, but in the absence of almost any mechanistic insight, I believe the manuscript falls short of what I believe is generally expected of papers in Nature.

Authors: As explained in detail above, because of our mechanistic data of viral infection sensitivity of different MIRAS models, different tissues, acute onset damage in vivo in viral exposure etc, the authors have to cordially disagree with the reviewer of "having no idea" of what is going on.

Minor point:

line 88-89: the authors state that there is NO difference in POLG1 transcript or protein levels, immediately after just stating that there was a defect in POLG1 protein levels (line 87). This is confusing.

Authors: thank you, this has been corrected.

Referee #4 (Remarks to the Author):

The authors rigorously addressed my comments to the extent possible. I find the work important and support publication without further experimentation.

Authors: Thank you, we highly appreciate this comment.